# Revisiting Matrix-Based Inversion of SMPS and HTDMA Data

Markus D Petters[1]

[1]NC State University, Department of Marine, Earth, and Atmospheric Sciences, Raleigh, NC, 27695-8208

**Correspondence:** Markus D Petters (mdpetter@ncsu.edu)

**Abstract.** Tikhonov regularization is a tool for reducing noise amplification during data inversion. This work introduces *RegularizationTools.jl*, a general-purpose software package to apply Tikhonov regularization to data. The package implements well-established numerical algorithms and is suitable for systems of up to ~1000 equations. Included is an abstraction to systematically categorize specific inversion configurations and their associated hyperparameters. A generic interface translates arbitrary linear forward models defined by a computer function into the corresponding design matrix. This obviates the need to explicitly write out and discretize the Fredholm integral equation, thus facilitating fast prototyping of new regularization schemes associated with measurement techniques. Example applications include the inversion involving data from scanning mobility particle sizers (SMPS) and humidified tandem differential mobility analyzers (HTDMA). Inversion of SMPS size distributions reported in this work builds upon the freely-available software *DifferentialMobilityAnalyzers.jl*. The speed of inversion is improved by a factor of ~200, now requiring between 2 and 5 ms per SMPS scan when using 120 size bins. Previously reported occasional failure to converge to a valid solution is reduced by switching from the L-curve method to generalized cross-validation as the metric to search for the optimal regularization parameter. Higher-order inversions resulting in smooth, denoised reconstructions of size distributions are now included in *DifferentialMobilityAnalyzers.jl*. This work also demonstrates that an SMPS-style matrix-based inversion can be applied to find the growth factor frequency distribution from raw HTDMA data, while also accounting for multiply-charged particles. The outcome of the aerosol-related inversion methods is showcased by inverting multi-week SMPS and HTDMA datasets from ground-based observations, including SMPS data obtained at Bodega Bay Marine Laboratory during the Calwater 2/ACAPEX campaign, and co-located SMPS and HTDMA data collected at the U.S. Department of Energy observatory located at the Southern Great Plains site in Oklahoma, U.S.A. Results show that the proposed approaches are suitable for unsupervised, nonparametric inversion of large-scale datasets as well as inversion in real-time during data acquisition on low-cost reduced-instruction-set architectures used in single-board computers. The included software implementation of Tikhonov regularization is freely-available, general, and domain-independent, and thus can be applied to many other inverse problems arising in atmospheric measurement techniques and beyond.

## 1 Introduction

Atmospheric aerosol play an important role in shaping the microphysics of clouds and the Earth's climate (Farmer et al., 2015; Kreidenweis et al., 2019). To predict the impact of aerosol on the Earth system, the distributions of particle size, chemical composition, hygroscopicity, and morphology must be known. The distribution of these properties across a population of

particles formally defines the mixing state of the aerosol (Riemer et al., 2019). Accurate measurements of these distributions are critical for formulating models that link aerosol, cloud, and climate properties.

Differential mobility analyzers (DMAs) select particles as a function of their size, charge, and an applied voltage. DMAs and tandem DMAs are widely used to measure the distributions of size and distributions of aerosol physicochemical properties (Park et al., 2008). For examples, a single DMA can be used to measure the aerosol size distribution by scanning voltage (Wang and Flagan, 1990). Humidified tandem DMAs (HTDMAs) can be used to measure the growth factor or hygroscopicity frequency distribution (Gysel et al., 2009). DMA-particle mass analyzer measurements can be used to resolve particle density distributions (Rawat et al., 2016; Sipkens et al., 2020). Tandem DMAs are important because they are one of only a handful techniques that can specifically characterize aspects of the aerosol mixing state (Riemer et al., 2019). Unfortunately, particles carrying multiple charges and different sizes transmit through the DMA at a single voltage, which creates artifacts in the raw instrument response that must be removed during post-processing of the data.

Humidified tandem DMAs select a single particle mobility diameter, pass this quasi-monodisperse aerosol through a humidification system, and then measure the humidified mobility response function using a second DMA operated in stepping or scanning mode (Rader and McMurry, 1986; Suda and Petters, 2013; Dawson et al., 2016). The humidified mobility response function is influenced by the particle size distribution, aerosol charge distribution, and growth factor frequency distribution function of the upstream aerosol. Gysel et al. (2009) show that the inversion from the humidified mobility response function to the growth factor frequency distribution is an ill-posed problem.

The inverse solution of ill-posed problems is characterized by strong sensitivity to noise superimposed on the data. Regularization methods are needed to relate an observed instrument response to the underlying physical property of the system under investigation. A common inverse method is $L_2$-regularization, developed independently by Phillips (1962), Twomey (1963), and Tikhonov (1963). Some examples of $L_2$-regularization involving atmospheric measurement techniques include inversion to find aerosol microphysical properties from measurements of optical properties (Dubovik and King, 2000; Müller et al., 2019), retrieve trace gas concentrations from remote sensors (Borsdorff et al., 2014), or estimate fluxes from a combination of measurements and atmospheric transport models (Krakauer et al., 2004). Application of $L_2$-regularization for problems involving DMAs include the reconstruction of the particle size distribution downstream of a single DMA (Wolfenbarger and Seinfeld, 1990; Kandlikar and Ramachandran, 1999; Talukdar and Swihart, 2003; Petters, 2018) and inversion to find size–mass distributions from coupled DMA-particle mass analyzer measurements (Rawat et al., 2016; Sipkens et al., 2020).

To date, $L_2$-regularization has not been applied to the inversion of HTDMA data. However, multiple other approaches have been used to estimate the growth factor frequency distribution from the humidified mobility response function. Stolzenburg and McMurry (1988) introduce the TDMAfit method. TDMAfit assumes a multi-mode normally-distributed hygroscopic growth factor frequency distribution. Parameters of the growth factor frequency distribution are varied such that the error between the modeled and observed humidified mobility response-functions are minimized. Cubison et al. (2005) apply the optimal estimation method (OEM) to derive the growth factor frequency distribution. This method uses an estimate of the covariance matrix, the measurements, and the forward model to retrieve the growth factor frequency distribution. The advantage of the optimal estimation method over TDMAfit is that it is nonparametric, i.e., it makes no prior assumption about the functional form of the

growth factor frequency distribution. However, the method sometimes produces oscillatory and negative solutions. Gysel et al. (2009) introduce TDMAinv, a piecewise linear version of TDMAfit. The piecewise method is also nonparametric. Constrained minimization is applied to find the growth factor frequency distribution; this avoids the negative solutions encountered in the optimal estimation method. Gysel et al. (2009) briefly discuss the role of multiple charges in the inversion and state that "the measured humidified mobility response function is a superposition of contributions from different dry sizes [...] and appropriate data inversion is hardly possible. Unfortunately an SMPS-style multicharge correction cannot be applied because the relative contributions from singly and multiply charged particles to every data point of the MDF cannot be distinguished." (In the direct quote, SMPS denotes scanning mobility particle sizer (Wang and Flagan, 1990) and MDF denotes mobility distribution function). Nevertheless, Shen et al. (2021) compute the contribution of multiply charged particles to the humidified mobility response function assuming that the larger multiply charged particles express the mean growth factor. However, they state that the correction of growth factor frequency distribution for multiply charged particles "is too complicated" (Shen et al., 2021) due to the need for multidimensional integration. Finally, Oxford et al. (2020) introduced a forward model named TAO that corrects for the contribution of multiply charged particle to the signal when interpreting volatility tandem DMA measurement.

This work revisits the challenge to perform an SMPS-style inversion of the humidified mobility distribution to retrieve the growth factor frequency distribution while also accounting for multiply charged particles. $L_2$-regularization is used to find the inverse. The remainder of the work is structured as follows: Section 2 describes the theory of $L_2$-regularization and the numerical solution of the equations. The software package *RegularizationTools.jl* is introduced, which is a general domain-independent implementation of $L_2$-regularization. Forward models for transfer through the single DMA and tandem DMA are formulated using the formalism developed in Petters (2018) and cast into matrix form using abstractions introduced in *RegularizationTools.jl*. Section 3 uses synthetic data to demonstrate that $L_2$-regularization can be used to invert the humidified mobility distribution function to find the growth factor frequency distribution. Section 4 uses real-world data to showcase improvements for size distribution inversion and the newly introduced tandem DMA inversion that were added to the freely-available software package *DifferentialMobilityAnalyzers.jl* (Petters, 2018). Finally, Section 5 summarizes the improvements, advantages, and limitations of the methodologies introduced in this work.

## 2 Theory

Section 2.1 and 2.2 uses the following linear algebra notation. Capital bold-roman letters denote matrices ($\mathbf{A}$), lowercase roman letters denote vectors ($\mathrm{x}$) and lowercase italic symbols denote scalars ($n$). $\mathbf{A}^{\mathrm{T}}$ denotes the matrix transpose, and $\mathbf{A}^{+} = (\mathbf{A}^{\mathrm{T}}\mathbf{A})^{-1}\mathbf{A}^{\mathrm{T}}$ is the matrix pseudo-inverse. Section 2.3 uses additional notation described there.

## 2.1  $L_2$ Regularization

### 2.1.1  Theory

The formalisms closely follow the description in Hansen (2000). Consider a system of equations

$$b = \mathbf{A}x + \epsilon \tag{1}$$

where b is the measured response, $\mathbf{A}$ is the design matrix (which may or may not be square), x is the true quantity of interest, and $\epsilon$ is the random error. The regular least-squares solution computed using the pseudo inverse via $x = \mathbf{A}^+ b$ is often dominated by contributions from the error and the thus-obtained estimate for x is useless. Regularization addresses this issue by solving the minimization problem

$$x_\lambda = \arg\min \left\{ \|\mathbf{A}x - b\|_2^2 + \lambda^2 \|\mathbf{L}(x - x_0)\|_2^2 \right\} \tag{2}$$

where $x_\lambda$ is the regularized estimate of x, $\|\cdot\|_2$ is the Euclidean norm, $\mathbf{L}$ is a filter matrix, $\lambda$ is the regularization parameter, and $x_0$ is a vector of an *a-priori* estimate of the solution. The *a-priori* estimate can be taken to be $x_0 = 0$ if no *a-priori* information is known. The filter matrix is often taken to be the identity matrix $\mathbf{I}$ or a derivative operator. Common choices are the first and second derivative operator defined as the uppper bidiagonal $(-1, 1)$ and the upper tridiagonal $(1, -2, 1)$ matrix, respectively. For $\lambda = 0$, the solution is equivalent to $x_\lambda = \mathbf{A}^+ b$. In the limit $\lim_{\lambda \to \infty} x_\lambda = x_0$. Thus the regularization parameter "interpolates" between the noisy ordinary least squares solution and the *a-priori* estimate $x_0$.

The analytical solution for Eq. (2) is the regularized normal equation

$$x_\lambda = \left( \mathbf{A}^T \mathbf{A} + \lambda^2 \mathbf{L}^T \mathbf{L} \right)^{-1} \left( \mathbf{A}^T b + \lambda^2 \mathbf{L}^T \mathbf{L} x_0 \right) \tag{3}$$

which is derived by taking the derivative of the right hand side of Eq. (2) with respect to x, setting it to zero, and solving for x. Equation (3) is in standard form if $\mathbf{L} = \mathbf{I}$. The optimal regularization parameter can be obtained using a variety of techniques, including the L-curve method (Hansen, 2000) and generalized cross-validation (GCV, Golub et al., 1979). Both methods use metrics that penalize solutions with large variance (amplified noise) or large bias.

The L-curve method involves a plot of $\log \|\mathbf{A}x_\lambda - b\|_2^2$ vs. $\log \|\mathbf{L}(x_\lambda - x_0)\|_2^2$ . The optimal $\lambda$ occurs at the corner of the resulting L-curve, which can be found algorithmically. However, automating the L-curve method can be more challenging than other automated methods, as further discussed below.

The generalized cross-validation estimator presents a mathematical shortcut to compute the leave-one out cross-validation estimate, which removes one point from the data, creates a model, computes the error between the model and data point not included in the data, and then averages the result over all permutations. It is given by

$$V(\lambda) = \frac{n \|(\mathbf{I} - \mathbf{A}_\lambda) b\|_2^2}{tr(\mathbf{I} - \mathbf{A}_\lambda)^2} \tag{4}$$

where $\mathbf{A}_\lambda b = \mathbf{A} x_\lambda$, $\mathbf{A}_\lambda$ is the influence matrix, $tr$ is the matrix trace, and $n$ is the size of b. The optimal $\lambda_{opt}$ coincides with the global minimum of $V(\lambda)$. Equation (4) requires that the system is in standard form. For systems in non-standard form,

conversion to standard form is required before computing $V(\lambda)$. In many cases $\lambda_{opt}$ found by the L-curve and generalized cross-validation are similar, and the retrieved solutions $x_\lambda$ are nearly indistinguishable. Differences between these two estimates are related to the computational speed-to-converge and robustness, i.e., that the system converges to the optimal solution.

### 2.1.2 Algorithms

Equation (3) can be solved straightforwardly using any software that supports linear algebra operations. This brute force approach, however, is slow. Efficient algorithms to solve Eqs. (3) and (4) have been developed. The algorithms used here are briefly described. If $\mathbf{L} \neq \mathbf{I}$, Eq. (3) is transformed to standard form using the generalized singular value decomposition of $\mathbf{A}$ and $\mathbf{L}$ as derived by Eldén (1982) and summarized by Hansen (1998). Equation (3) is solved using Cholesky factorization when possible since it is the computationally fastest approach (Lira et al., 2016). If Cholesky factorization fails, one of the fallback solvers selected by the linear algebra package of the programming language is used. Equation (4) is solved using the singular value decomposition of $\mathbf{A}$ and the iterative algorithm described in Bates et al. (1986). The optimal $\lambda_{opt}$ for generalized cross-validation is found by minimizing $V(\lambda)$ on a bounded interval using Brent's method (Mogensen and Riseth, 2018). The optimal $\lambda_{opt}$ for the L-curve method is found by maximizing Eq. (18) in Hansen (2000) on a bounded interval using Brent's method.

### 2.1.3 Classification of Methods

The inverse problem can be solved using specific methods. Here, method refers to the content of the filter matrix $\mathbf{L}$, whether an *a-priori* estimate is used, and whether constraints are imposed on the solution. Methods are encoded through the following expression

$$L_k x_0 D_\epsilon B_{[lb,ub]}(alg) \tag{5}$$

where $L_k$ denotes the order of the filter matrix $\mathbf{L}$, $x_0$ denotes whether an *a-priori* estimate is used, $D_\epsilon$ denotes whether data-based constraints are used (explained further below), and $B_{[lb,ub]}$ denotes whether a lower bound (lb) or upper bound (ub) is imposed on the solution (explained further below). The argument ("alg") denotes constraints on the search algorithms, e.g., L_curve or gcv and/or the bounded interval over which $\lambda$ is varied. The expression is composable. For example, the method $L_2$ denotes inversion using the second order derivative without an *a-priori* estimate, data-based constraints, and lower/upper bound constraints. The method $L_0 B_{[0,1]}(alg = L\_curve, \lambda_1 = 0.01, \lambda_2 = 100)$ denotes inversion with $\mathbf{L} = \mathbf{I}$, imposing that all $x_\lambda \in [0,1]$, the use of the L-curve method, and $\lambda_{opt} \in [0.01, 100]$. If alg is unspecified, defaults of $alg = gcv, \lambda_1 = 0.001, \lambda_2 = 1000$ are implied. This approach of method encoding provides a convenient classification system to enumerate the set of available methods as well as to specify the method in a high-level application interface for software function calls. There are eight combinations by which to compose methods via Eq. (5), $L$, $Lx_0$, $Lx_0 B$, $Lx_0 D$, $Lx_0 BD$, $LB$, $LBD$, and $LD$. Combined with the three most common filter matrices $L_0 = \mathbf{I}$, $L_1 =$ upper bidiagonal$(-1,1)$ and, $L_2 =$ upper tridiagonal$(1,-2,1)$ this results in 24 unique methods.

*Data-based constraints:* Huckle and Sedlacek (2012) proposed a two-step data-based regularization where the filter matrix is modified according to

$$\mathbf{L} = \mathbf{L}_k \mathbf{D}_{\hat{x}}^{-1} \tag{6}$$

where $\mathbf{L}_k$ is one of the finite difference approximations of a derivative, $\mathbf{D}_{\hat{x}} = diag(|\hat{x_1}|, \ldots, |\hat{x_n}|)$, $\hat{x}$ is the reconstruction of $x$ using $\mathbf{L}_k$. In the case that $|\hat{x_i}| < \epsilon$ those elements are set to be equal $\epsilon$, where $0 < \epsilon << 1$.. The method $L_1 D_{1e-2}$ represents a filter matrix with a first-order derivative operator applied to Eq. (6) with $\epsilon = 1e - 2$. Exponential notation is used because subscripts are difficult to superscript.

*Lower/upper bound constraints:* The retrieved $\mathrm{x}_\lambda$ from the regularized normal equation can have oscillatory and/or nonphysical solutions. An alternative approach is to treat Eq. (2) as a constrained minimization such that the solution is subject to the optional constraint $\mathrm{x}_{lb} < \mathrm{x} < \mathrm{x}_{ub}$. Here, the following procedure is implemented for the bounded search: first, the optimal $\lambda_{opt}$ is found using the regularized normal equations. The thus-obtained solution $\mathrm{x}_\lambda$ is truncated at the upper and lower bounds and then passed as an initial condition to a least-squares numerical solver. The Ceres solver (Agarwal et al.) is used with the Dogleg method and QR solver as implemented in the freely-available *LeastSquaresOptim.jl*[1] library. The net result is an optimized solution that is within the specified upper and lower bounds. The upper and lower bounds are vectors of the same size as $\mathrm{x}$.

### 2.1.4 Software Implementation

$L_2$-regularization, as described in the previous sections, is implemented in a freely-available software package *Regularization-Tools.jl* that is written by the author and provided as a supplement to this work. The implementation is in the Julia programming language (Bezanson et al., 2017). The package has a similar name and some overlap with the package Regularization Tools by Hansen (2007). However, the packages differ in software architecture, programming language, and scope. *Regularization-Tools.jl* provides a simple high-level interface to compute $\mathrm{x}_\lambda$ using a single function call, for example

$$\mathrm{x}\lambda = \mathrm{invert}(\mathrm{A}, \mathrm{b}, \mathrm{L_k x_0 B}(\mathrm{k}, \mathrm{x_0}, \mathrm{lb}, \mathrm{ub}); \mathrm{alg} =: \mathrm{L\_curve}) \tag{7}$$

where A is the design matrix, b is the observation vector, $\mathrm{L_k x_0 B}$ is a parameterized algebraic data type that encodes the specific method. The hyper parameter k specifies the order, $\mathrm{x_0}$ specifies a vector of the *a priori* estimate, lb and ub specifies vectors of the lower and upper bounds. Other methods can be specified according to Eq. (5). Examples are provided in the documentation of the package.

### 2.2 Computing the Design Matrix

The design matrix can be obtained from a forward model

$$\mathrm{y} = F(\mathrm{x}, \mathrm{c}) \tag{8}$$

---

[1] *https://github.com/matthieugomez/LeastSquaresOptim.jl*

where y is a vector representing the error-free observations, x is the vector of true inputs, c is a vector of controlling parameters, and $F$ is the linear forward model function that maps over x to compute y subject to the constraint of c. The matrix of the linear transformation $y = \mathbf{A}x$ is then given by

$$\mathbf{A} = [F(e_1)\ F(e_2)\ \ldots\ F(e_n)] \tag{9}$$

where $e_1, \ldots, e_n$ is the standard basis. *RegularizationTools.jl* also provides an abstract generic interface that simplifies computation of the design matrix from arbitrary forward models of linear processes. Examples demonstrating how to use this generic interface are provided in the documentation of the package. The examples include the solution for transit through the tandem DMA described further below, the solution of the Fredholm integral equation of the first kind given by Baart (1982), the optical convolution that underlies size distribution retrieval from scattering and absorption properties (Müller et al., 2019), and the 2D Gaussian blur function encountered in image processing.

## 2.3 Design Matrices For Differential Mobility Analyzers

Differential mobility analyzers consist of two electrodes held at a constant- or time-varying electric potential. Cylindrical (Knutson and Whitby, 1975) and radial (Zhang et al., 1995; Russell et al., 1996) electrode geometries are the most common. Charged particles in a flow between the electrodes are deflected to an exit slit and measured by a suitable detector, usually a condensation particle counter. The fraction of particles carrying $k$ charges is described by a statistical distribution that is created by the charge conditioner used upstream of the DMA. The functions governing the transfer through bipolar charge conditioners, single DMAs, and tandem DMAs are well understood (Knutson and Whitby, 1975; Rader and McMurry, 1986; Reineking and Porstendörfer, 1986; Wang and Flagan, 1990; Stolzenburg and McMurry, 2008; Jiang et al., 2014).

The DMA selects particles by electrical mobility. The relationship between mobility and mobility diameter is well known and well defined. The relationship is given, for example, in Eq. (2) in Petters (2018). This work also makes use of the "apparent +1 mobility diameter". It is defined as the conversion from mobility to diameter assuming singly charged particles using the mobility grid scanned by either DMA 1 or DMA 2. The apparent +1 mobility diameter represents the natural diameter axis of a DMA response function, i.e. a plot of the raw detector response versus the nominal DMA setpoint diameter. It is an equivalent measure of mobility. The apparent +1 mobility diameter is ambiguous. Larger particles carrying more than one charge may have the same apparent +1 mobility diameter as smaller particles carrying fewer charges. The "apparent growth factor" is defined as the apparent +1 mobility diameter in scanned by DMA 2 divided by the nominal selected dry diameter in DMA.

The traditional mathematical formulation of transfer through the DMA is summarized in Stolzenburg and McMurry (2008) and references therein. Briefly, the integrated response downstream of the DMA operated at voltage $V_1$ is given by a single integral that includes a summation over all selected charges. The size distribution is measured by varying voltage $V_1$, which produces the raw response function defined as the integrated response downstream of the DMA as a function of upstream voltage. The size distribution is found by inversion. The basic mathematical problem associated with inverting the response function to find the size distribution is summarized by Kandlikar and Ramachandran (1999). The integral is discretized by quadrature to find the design matrix that maps the size distribution to the response function. $L_2$ regularization is one of several

methods to reconstruct the size distribution from the response function (Voutilainen et al., 2001; Kandlikar and Ramachandran, 1999).

The integrated response downstream of a tandem DMA that is operated at voltages $V_1$ and $V_2$ requires evaluating integrals of the upstream particle size distribution over size and the grown particle size distribution over size. The integration must be repeated for each charge state. Scanning over a range of voltages $V_2$ results in the raw TDMA response function. For the forward calculation, the objective is to find a design matrix that maps the growth factor frequency distribution to the raw TDMA response function.

Petters (2018) introduced a computational approach to model transfer through the DMA. The main idea of the approach is to provide a domain specific language comprising a set of simple building blocks that can be used to algebraically express the response functions intuitively through a form of pseudo code. The main advantage of this approach is that the expressions simultaneously encode the theory governing the transfer through the DMA and the algorithmic solution to compute the response function. The resulting expressions are concise. They are easily identified within actual source code when working through the examples provided with the package documentation. This makes the code easily modifiable by non-experts to change existing terms or add new convolution terms without the need to develop algorithms.

A disadvantage of the computational approach over the traditional mathematical approach is that computation lacks standardization of notation. This can blur the line between general pseudo code and language specific syntax. Some of the applied computing concepts may be less widely known when compared to standard mathematical approaches. Nevertheless, the author believes that the advantages of the computational approach outweigh the drawbacks. Therefore, this work builds upon the expressions reported in Petters (2018). Updates and clarifications to the earlier work are noted where appropriate.

The computational language includes a standardized representation of aerosol size distributions, operators to construct expressions, and functions to evaluate the expressions. Size distributions are represented as a histogram and internally stored in the form of the *SizeDistribution* composite data type. Composite data types combine multiple arrays into a single symbol for ease of use, thus facilitating faster experimental design and analysis. The size distribution data type *SizeDistribution* includes vectors of the selected mobility bins considered by the DMA, +1 mobility diameter bin edges and +1 mobility diameter bin midpoints computed from the mobility grid, number concentration, log-normalized spectral density, and logarithmic bin widths. *SizeDistributions* are denoted in blackboard bold font (e.g., $\mathbb{n}$, $\mathbb{r}$, etc.). *SizeDistributions* are the building block of composable algebraic expressions through operators that evaluate to transformed *SizeDistributions*. For example, $\mathbb{n}_1 + \mathbb{n}_2$ is the superposition of two size distributions and $a * \mathbb{n}$ is the uniform scaling of the concentration fields by factor $a$, $\mathbf{A} * \mathbb{n}$ is matrix multiplication of $\mathbf{A}$ and concentration fields of the size distribution, and $a \cdot \mathbb{n}$ is the uniform scaling of the diameter field of the size distribution by factor $a$, and $T \cdot \mathbb{n}$ is the elementwise scaling of the diameter field by factor $T$. (Note that Petters (2018) used $T \cdot \mathbb{n}$ as the elementwise scaling. The extra dot has been dropped to stay consistent with the current software implementation).

Generic functions are used to evaluate expressions. The function $\sum(f, m)$ evaluates the function $f(x)$ for $x = [1, \ldots, m]$ and sums the results. If $f(x)$ evaluates to a vector, the sum is the sum of the vectors. The function $\mathrm{map}(f, x)$ applies $f(x)$ to each element of vector $x$ and returns a vector of results in the same order. The function $\mathrm{foldl}(f, x)$ applies the bivariate function $f(a, x)$ to each element of $x$ and accumulates the result, where $a$ represents the accumulated value. If no initial value is pro-

250 vided, as is the case in this manuscript, foldl applies the function to the first two elements of the list to compute the first $a$ . For example $\mathrm{foldl}(-, [1, 2, 3])$ evaluates the function $-(a, x)$ and yields $1 - 2 - 3 = -4$. The function $\mathrm{mapfoldl}(f, g, x)$ combines map and foldl. It applies function $f$ to each element in $x$ such that $y = f(x)$ and then reduces the result using the bivariate function function $g(a, y)$ where $a$ represents the accumulated value. For example, $\mathrm{mapfoldl}(\mathrm{sqrt}, -, [4, 16, 64])$ evaluates to $\mathrm{foldl}(-, [2, 4, 8]) = 2 - 4 - 8 = -10$. The function $vcat(x, y)$ concatenates arrays x and y along the first dimension in Julia.

However, other programming languages may concatenate along a different dimension as the definition of horizontal and vertical is arbitrary. Passing vcat to foldl (or mapfoldl) will result in a concatenated array. Anonymous functions are used as arguments to reducing functions. Anonymous functions are denoted as $x \rightarrow expression$, where $x$ is the argument consumed in the evaluation of the $expression$. These functions are generic and represent widely used computing concepts. They are implemented in most modern programming languages. DMA geometry, dimensions, and configuration are abstracted into composite types

$\Lambda$ (configuration comprising flow rates, power supply polarity, and thermodynamic state) and $\delta$ (DMA domain defined by a mobility/size grid). Each DMA is fully described by a pair $\Lambda, \delta$. Subscripts and superscripts are used to distinguish between different configurations in chained DMA setups, e.g. $\delta_1$ and $\delta_2$ denoting the first and second DMA, respectively. Application of size distribution expressions to transfer functions constructs a concise model of the transmitted DMA mobility distribution, denoted as the DMA response function. Implementation of the language is distributed through a freely-available and indepen-

dently documented package *DifferentialMobilityAnalyzers.jl,* written in the Julia language. Expressions in the text are provided in general mathematical form for readability.

Petters (2018) gives a simple expression that model transfer through the DMA. The function $T_{size}^{\Lambda, \delta}(k, z^s)$ evaluates to a vector representing the fraction of particles carrying $k$ charges that exit DMA$^{\Lambda, \delta}$ as a function of mobility

$$T_{size}^{\Lambda, \delta}(k, z^s) = \Omega(Z, z^s/k, k). * T_c(k, D_{p,1}). * T_l(D_{p,1}) \tag{10}$$

where $z^s$ is the centroid mobility selected by the DMA (determined by the voltage and DMA geometry), $Z$ is a vector of particle mobilities, $\Omega$ is the diffusing DMA transfer function (Stolzenburg and McMurry, 2008), $T_c$ is the charge frequency distribution (Wiedensohler, 1988), and $T_l$ is the diameter-dependent penetration efficiency (Reineking and Porstendörfer, 1986). The diameter $D_{p,1} = D_p(z, k = 1)$, where $z$ is an element of $Z$. The function $\Omega$ has been updated from Petters (2018). The version in Petters (2018) computed the shape of the transfer function for the mobility diameter corresponding to singly charged particles and then applied the same shape of the transfer function and diffusional loss to the multiply charged particles. The functional $\Omega$

depends on three arguments $\Omega(Z, z^*, k)$ and implicitly on the DMA configuration $\Lambda$ (i.e., Eq 13 in Stolzenburg and McMurry, 2008). The output is a vector along the mobility grid $Z$. The maximum transmission occurs at $Z/z^* = 1$. The last argument denotes the number of charges. It is used to compute the mobility diameter from $z^*$ and in turn the diffusion coefficient which is required to account for diffusional broadening of the transfer function. The output of $T_{size}^{\Lambda, \delta}(k, z^s)$ is the transmission of parti-

cles through the DMA in terms of the true particle mobility diameter. This is achieved by passing $z^s/k$ as argument to $\Omega$, which corresponds to the centroid mobility setting for the DMA to transmit particles with $k$ charges under the assumption that they carry only a single charge. The net result is that $D_{p,1} = D_p(z, k = 1)$, where $z$ is an element of $Z$, becomes equal to the true mobility diameter axis. As a consequence the charge fraction $T_c(k, D_{p,1})$ and penetration efficiency $T_l(D_{p,1})$ are evaluated at

the correct diameter. The function $T_{size}^{\Lambda,\delta}(1, z^s)$ evaluates to a vector of the same length as $Z$. Performing an elementwise sum over all $T_{size}^{\Lambda,\delta}(k, z^s)$ (where the sum is over all charges $k$) produces the net transmission probability function. Multiplication of the transmission probability function with the input distribution results in the mobility distribution transmitted by the DMA. Examples for $T_{size}^{\Lambda,\delta}(1, z^s) * \mathbb{n}$, $T_{size}^{\Lambda,\delta}(2, z^s) * \mathbb{n}$, and $T_{size}^{\Lambda,\delta}(3, z^s) * \mathbb{n}$ are shown in Figure 2, right panel in Petters (2018). Note that Eq. (10) can be evaluated using arbitrarily discretized $Z$ vectors.

Petters (2018) also gives an expression that evaluates to the convolution matrix for passage through a single DMA that is valid in the context of size distribution measurement system, e.g. SMPS. Since the expression includes a summation over all charges, the information on particle physical diameter of multiply charged particles is lost.

$$\mathbf{A} = \mathrm{mapfoldl}\{z^s \rightarrow \Sigma[k \rightarrow T_{size}^{\Lambda,\delta}(k, z^s), m]^\mathrm{T}, \mathrm{vcat}, Z_s\} \tag{11}$$

where, $m$ is the upper number of multiply charged particles, $^\mathrm{T}$ is the transpose operator, and $Z_s$ is a vector of centroid mobilities scanned by the DMA. The matrix is square if $Z_s = Z$ in Eq. 11. However, this is not a necessary restriction. Eq. (11) evaluates to the same as Eq. (8) in Petters (2018), but the notation is revised to be more general by removing the Julia-specific splatting construct and replacing it with more widely used generic functions.

To help with parsing the expression, $T_{size}^{\Lambda,\delta}(k, z^s)$ evaluates to a vector of transmission for $k$ charges and set point centroid mobility $z^s$ as a function of the entire mobility grid (e.g. 120 bins discretized between mobility $z_1$ and $z_2$). The function $\Sigma[k \rightarrow T_{size}^{\Lambda,\delta}(k, z^s), m]$ superimposes the vectors for all charges. Mapping $z^s \rightarrow \Sigma[k \rightarrow T_{size}^{\Lambda,\delta}(k, z^s), m]$ over the centroid mobility grid $Z_s$ produces an array of vectors, each corresponding to the transmission for a single size bin. Transposing the vectors and reducing the collection through concatenation produces the design matrix that links the mobility size distribution to the response function, i.e.

$$\mathbb{r} = \mathbf{A}\mathbb{n} + \epsilon \tag{12}$$

where $\mathbb{r}$ is the response distribution, $\mathbb{n}$ is the true mobility size distribution, and $\epsilon$ is a vector denoting the random error that may be superimposed as a result of measurement uncertainties. By design $\mathbb{n}$ and $\mathbb{r}$ are *SizeDistribution* objects, which represent the distribution as a histogram in both spectral density units (dN/dlnD) and concentration per bin units. The latter is the raw response function, where each element corresponds to the integrated response downstream of DMA 1 for a set upstream voltage (or corresponding $z^s$ or apparent +1 mobility diameter but not true physical diameter for multiply charged particles). Note, however, that the response function is not a true particle size distribution in the scientific sense since information about multiply charged particles is lost. The representation of $\mathbb{r}$ as *SizeDistribution* object is to allow response functions to used in the expression-based framework used here.

The mobility distribution exiting the humidity conditioner and before entering DMA 2 in the humidified tandem DMA is evaluated using the expression

$$\mathbb{M}_k^{\delta_1} = g_0 \cdot \left[ T_{size}^{\Lambda_1,\delta_1}(k, z^s) * \mathbb{n} \right] \tag{13}$$

where, $g_0 = D_{wet}/D_{dry}$ is the diameter growth factor, $D_{dry}$ is the selected diameter by DMA 1, $D_{wet}$ is the diameter after the humidifier, $T_{size}^{\Lambda_1,\delta_1}(k, z^s)$ is as in Eq. (10), and $\mathbb{n}$ is the mobility size distribution upstream of DMA 1. Subscripts are used

to differentiate DMA 1 and 2 which possibly have different geometries, flow rates, thermodynamic state, and mobility grids, e.g. $\Lambda_1$, $\Lambda_2$ and $\delta_1$, $\delta_2$. To help parse Eq. (13), the product $T_{size}^{\Lambda_1, \delta_1}(k, z^s) * \mathbb{n}$ evaluates to the transmitted mobility distributions of particles carrying $k$ charges at the set-point mobility $z^s$ in DMA 1. The size distribution is grown by the growth factor $g_0$, which is achieved by applying the $\cdot$ operator to the product $T_{size}^{\Lambda, \delta}(k, z^s) * \mathbb{n}$. Equation (13) assumes that $g_0$ applies to all particle sizes.

The total humidified apparent +1 mobility diameter distribution $\mathbb{m}_t^{\delta_2}$ exiting DMA 2 is given by

$$\mathbb{m}_t^{\delta_2} = \sum_{k=1}^{m} \left( \mathbf{O}_k * \mathbb{M}_k^{\delta_1} \right) \tag{14}$$

where, $m$ is upper number of charges on the multiply charged particles and

$$\mathbf{O}_k = \text{mapfoldl}\{z^s \rightarrow [\Omega^{\Lambda_2, \delta_2}(Z, z^s/k, k). * T_l^{\Lambda_2, \delta_2}(D_{p,1})]^T, \text{vcat}, Z_{s,2}\} \tag{15}$$

is the convolution matrix for transport through DMA 2 and particles carrying $k$ charges. In Eq. (15), $Z_{s,2}$ is a vector of centroid mobilities scanned by DMA 2. Note that the choice of $Z$ inside $\Omega$ is up to the user. Sensible choices are $Z = Z_{s,1}$ or $Z = Z_{s,2}$ the implications of which are further discussed later. Equations (14) and (15) have been modified from those in Petters (2018) in the following manner. The convolution matrix $\mathbf{O}_k$ is computed individually for each charge. The version in Petters (2018) computed the matrix corresponding to singly charged particles and then applied the same matrix to multiply charged particles. Since $\mathbf{O}_k$ is now charge resolved, it is moved into the summation in Eq. (14). Computation of $\mathbf{O}_k$ through Eq. (15) has been revised to be more general by removing a Julia-language specific construct. $\mathbf{O}_1$ computed by Eq. (15) produces the same matrix as in Petters (2018). The resulting $m_t^{\delta_2}$ size distribution represents the apparent +1 mobility diameter scanned by DMA 2. Eqs. (13)-(15) relax an approximation made in a similar treatment in Petters (2018). There it was assumed that particles that the apparent +1 mobility diameter (and thus apparent growth factor) for particles carrying multiple charges is the same as for single charged particles. This is incorrect. Particles carrying more than a single charge alias at a smaller particle size (Gysel et al., 2009; Shen et al., 2021). The effect is due to the size dependence of the slip-flow correction factor and captured by the revised charge-resolved convolution matrices $\mathbf{O}_k$.

If the aerosol is externally mixed, the humidified apparent growth factordistribution function exiting DMA 2 is given by

$$\mathbb{m}_t^{\delta_2} = \int_0^\infty P_g * \left[ \sum_{k=1}^{m} \left( \mathbf{O}_k * \mathbb{M}_k^{\delta_1} \right) \right] dg_0 \tag{16}$$

where $P_g$ is the growth factor probability density function and the diameters resulting from the intermediate calculation $\sum_{k=1}^{m} \left( \mathbf{O}_k * \mathbb{M}_k^{\delta_1} \right)$ are normalized by $D_{dry}$. $\mathbb{m}_t^{\delta_2}$ in Eq. (16) is the forward model through the tandem DMA. Using the notation in section 2.2,

$$F(\mathrm{x}, \mathrm{c}) = \int_0^\infty P_g * \left[ \sum_{k=1}^{m} \left( \mathbf{O}_k * \mathbb{M}_k^{\delta_1} \right) \right] dg_0 \tag{17}$$

where x is the true $P_g$ and the vector c of constraining parameters comprises the DMA setups $\Lambda_1, \Lambda_2, \delta_1, \delta_2$ and upstream size distribution $\mathbb{n}$. Computer code that creates a forward model for tandem DMAs has been added to the *DifferentialMobiltyAnalyzers.jl* package and is annotated in the documentation of the package.

For purposes of the forward model, the mobility grid for DMA 1 is discretized at a resolution of $i$ bins by specifying the $Z$ vector in Eq. (10). If the $Z$ vector does not match that of the aerosol size distribution $\mathtt{n}$, the size distribution bins are

interpolated onto the diameter bins corresponding to the $Z$ bins. Transmission through DMA 1 is computed for a specified $z^s$ (the dry mobility) and $g_0$ (the growth factor) via Eq. (13). The resulting $\mathbb{M}_k^{\delta_1}$ lie on the same $Z$ grid with $i$ bins. Any mismatches between the apparent growth factor and the underlying $Z$ grid are resolved via interpolation implicit in the $\cdot$ operator. ($f \cdot \mathtt{n}$ is the uniform scaling of the diameter field of the size distribution by factor $f$. If the resulting diameters are off the original diameter grid, the result in interpolated onto the grid defined within $\mathtt{m}$).

The mobility grid for DMA 2 is represented by the vector $Z_{s,2}$ in Eq. (15) and discretized at a resolution of $j$ bins over a custom mobility range. If the vector $Z$ inside the square bracket of Eq. (15), $[\Omega^{\Lambda_2,\delta_2}(Z, z^s/k, k). * T_l^{\Lambda_2,\delta_2}(D_{p,1})]$ equals that of DMA 1, the product $\mathbf{O}_k * \mathbb{M}_k^{\delta_1}$ will map the $i$ bins from DMA 1 to the $j$ bins in DMA 2. Alternatively, if the $Z$ vector inside the square bracket of Eq. (15) is taken to be equal to $Z_{s,2}$, the matrices $\mathbf{O}_k$ are square and of dimension $j \times j$. In that case, the transmitted and grown distribution from DMA 1 ($i$ bins along the mobility axis of DMA 1) is interpolated onto the mobility

grid of DMA 2 prior to evaluating $\mathbf{O}_k * \mathbb{M}_k^{\delta_1}$. The advantage of this approach is that for $j < i$ the matrices $\mathbf{O}_k$ are smaller and subsequent calculations are faster.

The forward model, defined by Eq. (14) can be evaluated for arbitrary $g_0$ values. Thus the growth factor probability distribution $P_g$ in Eq. (17) can be discretized into $n$ arbitrary growth factor bins. A natural choice is to accept growth factor values that coincide with the mobility grid of DMA 2, i.e. the bins align with $g = D_{p,1}/D_d$, where $D_d$ is the nominal diameter selected by

DMA 1 and $D_{p,1} = D_p(z, k = 1)$ and $z$ is an element of $Z_{s,2}$. However, this is not required for evaluating Eq. (17). Equation (17) is cast into matrix form such that the humidified mobility distribution function is given by

$$\mathtt{m}_t^{\delta_2} = \mathbf{B} P_g + \epsilon \tag{18}$$

where the matrix $\mathbf{B}$ is understood to be computed for a specific input aerosol size distribution, and $\epsilon$ is a vector that denotes the random error that may be superimposed as a result of measurement uncertainties. If the grids used to represent the growth

factor distribution and that of DMA 2 do not align, interpolation is used to map the growth factor bins from the growth factor distribution onto those corresponding to the DMA 2 grid. The choice of $i, j, n$, the ranges of mobility grids for DMA 1, DMA 2, and the range of the growth grid for $P_g$, is only constrained by computing resources and a physically reasonable representation of the problem domain. Reasonable choices are $i = 120$, $j = n = 60$, where the range in apparent growth factor spans $0.8 < g$ $< 5.0$. The size of $\mathbf{B}$ is $j \times n$. Uncertainties in the size distribution propagate into $\mathbf{B}$. The main influence of the error will be the

relative fraction of +1, +2, and +3 charged particles. Assuming a random error of $\pm 20\%$ in concentration, the overall effect on $\mathtt{m}_t^{\delta_2}$ is expected to be small.

Note that interpolation is widely used in this framework. Interpolation may affect how errors propagate through the model. Interpolation in Eq. (13) is unavoidable. However, interpolation can be minimized by working with non-square $\mathbf{O}_k$ and matching the grid of $P_g$ to that of DMA 2. Informal tests working with different binning schemes suggests that the influence of

interpolation choices on the final result is smaller than typical experimental errors.

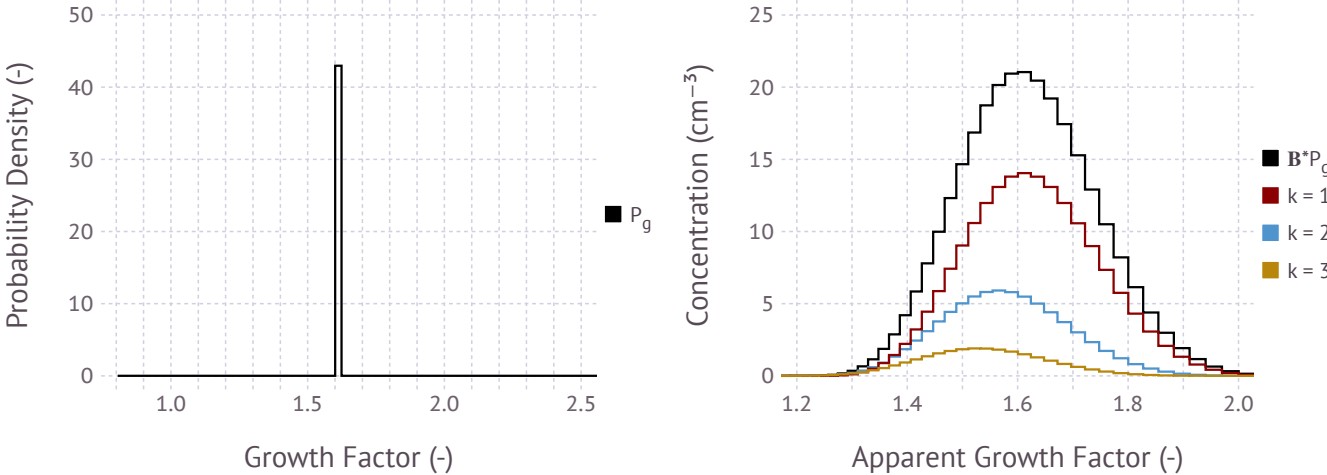

**Figure 1.** *Left:* Input growth factor probability density $dF/dg$ assuming that all particles have a single growth factor ~1.6. The area under the curve evaluates to unity. *Right:* Modeled apparent mobility distribution function calculated using Eq. (15) and partial distributions for individual charges $k = +1, +2, +3$ computed via $\mathbf{O}_k * \mathbb{M}_k^{\delta_1}$. The example is free of measurement error, i.e., $\epsilon = 0$. The black trace is what would be observed by a hypothetical measurement with a condensation particle counter.

Figure 1 shows an example application of Eq. (18) for an input growth factor frequency distribution where all particles are assumed to have the same growth factor ~1.6. The frequency distribution is evaluated along a discrete growth factor grid with 120 bins between $0.8 < g < 5$. Note that the size grid (or apparent growth factor grid) must be extended to large sizes to capture the growth of multiply-charged particles computed via Eq. (13). The assumed input size distribution is bimodal with mode diameters of 60 and 140 nm, geometric standard deviations of 1.4 and 1.6, and number concentrations of 1300 and 2000 cm$^{-3}$ in modes 1 and 2, respectively. The assumed sheath-to-sample flow ratios are 5:1 in both DMAs. The product $\mathbf{B}P_g$ is the raw response that would be measured by a condensation particle counter at the exit of the instrument. Contribution of $+1$, $+2$, and $+3$ charged particles to the total can be computed via $\mathbf{O}_k * \mathbb{M}_k^{\delta_1}$. Although the nominal growth factor is the same for all sizes, the apparent mode of the growth factor decreases with increasing particle charge (see also Gysel et al., 2009; Shen et al., 2021). Therefore the axis is denoted as the apparent growth factor. Summing the partial distributions results in $\mathbf{B}P_g$, demonstrating that the matrix equation correctly maps $P_g$ to the response, including multiply charged particles. Figure 2 shows the relationship between four illustrative growth factor frequency distributions and the modeled apparent mobility distribution functions. The apparent mobility distribution function represents the raw particle concentration that would be measured by a detector as a function of apparent +1 mobility diameter. The diameter axis is normalized by the dry diameter selected by DMA 1. The selected examples comprise a testbed to evaluate the feasibility of an SMPS-style matrix-based inversion to recover $P_g$. The *Populations* example consists of an external mixture with compositions corresponding to four unique growth factors. The *Bimodal* example is the superposition of two Gaussian distributions with 70% of particles in the less hygroscopic mode. The

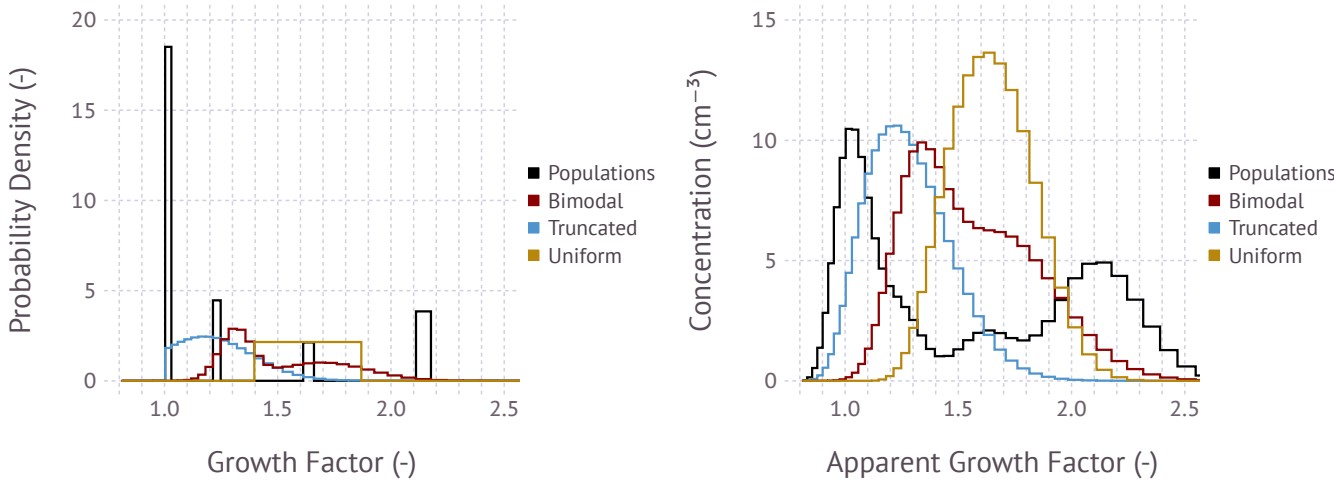

**Figure 2.** *Left:* Illustrative input growth factor probability density distributions. The area under the curve evaluates to unity. *Right:* Corresponding modeled apparent mobility distribution function calculated using Eq. (15). The example is free of measurement error, i.e. $\epsilon = 0$.

*Truncated* example is a Gaussian distribution truncated at $g = 1.0$. The *Uniform* example is a uniform distribution over a fixed interval. All frequency distributions integrate to unity, thus accounting for 100% of the particle population. The dry diameter and assumed input size distribution to compute the matrix $\mathbf{B}$ is the same as in Fig. 1. However, unlike in Fig. 1, the frequency distribution and matrix are evaluated along a coarser discrete growth factor grid with 60 bins between $0.8 < g < 5$. Note that the growth factor bin width is not constant, with wider bins at larger growth factors. This is due to the evaluation of the humidified size distribution along a geometrically stepped mobility grid. As will be shown next, 60-bin resolution is a suitable compromise between speed, accuracy, and resolution when computing the matrix-based inversion to infer $P_g$ from noise-perturbed apparent growth factor frequency distributions.

## 3  Matrix Inversion of the Humidified Mobility Distribution Function using Synthetic Data

Simulated examples are used to test if Eq. (18) is invertible. Figure 3 shows an example simulation for the *Bimodal* growth factor distribution test case. The humidified apparent growth factor distributions are calculated using Eq. (18). The noise-free example corresponds to $\epsilon = 0$ and represents the idealized measurement. Poisson counting statistics are simulated by converting concentration to the expected number of counts for a typical particle counter flow rate and bin integration time. Counts in each bin are computed by drawing a pseudo-random number from a Poisson distribution and converting the result back to concentration. Lower flow rates and shorter integration times increase the noise-perturbation of the apparent growth factor distribution. The apparent growth factor distribution is then inverted using the $L_0 D_{1e-3} B_{[0,1]}$ and $L_2 x_0 B_{[0,1]}$ method. Here $B_{[0,1]}$ is shorthand for setting all lower bounds equal to zero and all upper bounds equal to one. The *a-priori* estimate $\mathrm{x}_0$ is taken

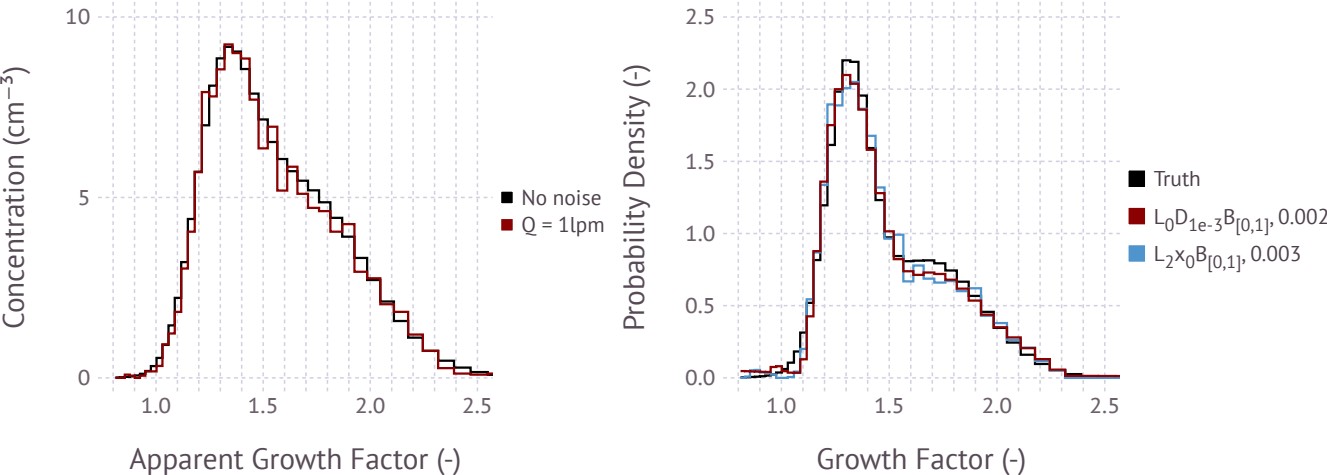

**Figure 3.** *Left:* Humidified apparent growth factor distribution function for the *Bimodal* example comprising superposition of two Gaussian distributions with 70% of particles in the less hygroscopic mode. The distributions are calculated $m_t^{\delta 2} = \mathbf{B} P_{gf} + \epsilon$. "No noise" corresponds to $\epsilon = 0$. "$Q = 1lpm$" corresponds to simulated Poisson noise equivalent for a condensation particle counter measuring at a flow rate of 1 L min$^{-1}$ and bin integration time of 2 seconds per bin. *Right:* Inverted growth factor probability density distribution using the $L_0 D_{1e-3} B_{[0,1]}$ and $L_2 x_0 B_{[0,1]}$ method. The area under the curve evaluates to unity. The *a-priori* estimate $x_0$ is the normalized apparent growth factor distribution. Values in the legend (0.002 and 0.003) correspond to the root mean square error between the true input (Truth) and the regularized solution evaluated in frequency space.

to be the normalized apparent growth factor distribution derived from the measured response function, where the normalization ensures that the sum over all bins is unity. Note that the inversion is performed treating the growth factor distribution in units of frequency instead of frequency density. This choice enables the upper bound constraint of unity. Since the true noise-free input growth factor frequency distribution is known, the fidelity of the inversion can be evaluated by computing the root mean square error between the noise-free solution and the regularized solution. Evaluating the root mean square error in frequency rather

than frequency density space results in more comparable values when contrasting narrow and broad probability distribution functions. The figure shows that both inversion methods produce a root mean square error between 0.002 and 0.003. Values less than 0.01 are typical for the reconstruction of *Bimodal* distributions at this bin resolution (see supporting information). Visual evaluation of the agreement between the reconstruction and the input suggests that either method is suitable for inversion.

Figure 4 is similar to Figure 3, showing an example simulation for an aerosol with uniform composition, i.e., all particles

have the same growth factor. Although the $L_2 x_0 B_{[0,1]}$ approach correctly infers the most probable growth factor, the predicted distribution is incorrect. Multiple modes to the left and right of the main mode are observed. The $L_2 x_0$ method produces an oscillatory solution with negative values (not shown). The small modes are the residual of this oscillatory solution that is truncated by the enforced [0,1] bound and the inability of the least-squares solver to converge on a better solution. A large root mean square error of 0.107results. In contrast, the data-constrained method $L_0 D_{1e-3} B_{[0,1]}$ leads to better reconstruction of the

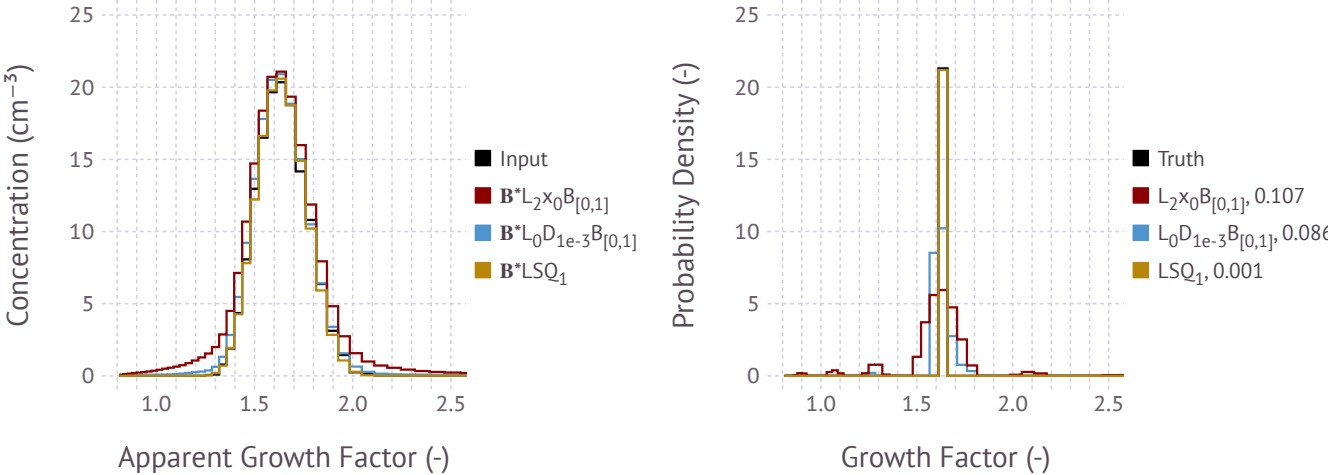

**Figure 4.** *Left:* Humidified apparent growth factor distribution function assuming that all particles have a single growth factor ~1.6. The black histogram corresponds to the input to the inversion, which is the noise-perturbed apparent growth factor distribution with simulated Poisson noise equivalent for a condensation particle counter measuring at a flow rate of 1 L min$^{-1}$ and bin integration time of 2 s per bin. Color lines depict the predicted apparent growth factor distributions based on the corresponding inversion shown in the right panel. *Right:* Inverted growth factor frequency distribution from the noise-perturbed spectrum. The true growth factor frequency distribution (black line) is obscured behind the gold and blue lines and is as in Fig. 1, left panel. Colors correspond to the inverted size distribution using the $L_0 D_{1e-3} B_{[0,1]}$, $L_2 x_0 B_{[0,1]}$, and $LSQ_1$ methods. The *a-priori* estimate $x_0$ is the normalized apparent growth factor distribution. Values in the legend (0.107, ...) correspond to the root mean square error between the true noise-free solution and the proposed solution.

true input. The main advantage of the $L_0 D_{1e-3} B_{[0,1]}$ inversion method over $L_2 x_0 B_{[0,1]}$ is that it is better able to reconstruct inputs with sharp edges.

The total number of composable regularization methods according to Eq. (5) is 24. Half of these methods do not include lower and upper bounds and these are not suitable for tandem DMA inversion due to the negative and oscillatory solutions for narrow inputs. The remaining 12 methods have been systematically tested using Monte-Carlo analysis described in detail
in the supporting information. Briefly, 60000 inversions were performed on synthetic data similar to the examples shown in Figs. 3 and 4. The total number concentration, dry diameter, number of bins, and random seeds were varied and the root mean square error was evaluated for each simulation. Results compiled in Fig. S1 show that all of the methods perform equally well for the *Bimodal, Uniform,* and *Truncated Normal* examples shown in Fig. 3. Method $L_0 D_{1e-3} B_{[0,1]}$ outperforms the other methods for grids with < 60 growth factor bins between $0.8 < g < 5$ and test cases with either one (e.g., Fig. 4) or two
discrete populations. However, even $L_0 D_{1e-3} B_{[0,1]}$ can lead to results similar to the example $L_2 x_0 B_{[0,1]}$ shown in Fig. 4 for some random seeds. Higher resolution grids generally lead to poor performance for discrete populations even for method $L_0 D_{1e-3} B_{[0,1]}$.

An alternative approach to fit single component data is to perform a nonlinear least squares fit to match the apparent growth factor distribution using the forward model while restricting the number of compositions to either one or two. This corresponds to a two- or four-parameter fit. Results from this procedure are either one or two growth factors and one or two fractions. The corresponding methods are denoted as $LSQ_1$ and $LSQ_2$, respectively. In the example shown in Fig. 4, $LSQ_1$ has the smallest root mean square error and is the best method to reconstruct the true growth factor. The $LSQ_1$ method is most suitable to infer the growth factor for laboratory measurements when it is known that the aerosol is internally mixed and only a single growth factor is expected.

Which method, however, should be selected when inverting real-world data and the number of components is unknown? Since the true solution is also unknown, the root mean square error between the truth and reconstruction is unavailable. It is, however, possible to compute the residual between the measured apparent growth factor distribution and the predicted apparent growth factor distribution from different reconstructions. A large residual can be used to flag truncated oscillatory solutions such as $L_2x_0B_{[0,1]}$ for narrow/single composition cases. Similarly, the residual is high if the true input is a broad growth factor frequency distribution that is attempted to be fitted using $LSQ_1$. For example, the red spectrum in the left graph of Fig. 4 shows poor agreement with the input and results in a much larger residual than $LSQ_1$ (values not shown). Therefore, a proposed unsupervised inversion scheme is to compute the solution of $LSQ_1$, $LSQ_2$ and $L_0D_{1e-3}B_{[0,1]}$ and then select the solution with the lowest residual relative to the apparent growth factor distribution.

Note, however, that the low residuals between the apparent growth factor distribution and the model do not automatically ensure that the algorithm a good or adequate solution. Additional tests should be performed to validate the physical plausibility of the solution. For example, the retrieved growth factors should be physically plausible at the applied relative humidity. The mode of the apparent growth factor distribution and the mode of the inverted growth factor distribution should be similar. A histogram of the root mean square error between can be plotted for a large data set. Visual inspection of fits for large root mean square error can be used to derive a threshold above which reconstructions are automatically rejected. The integrated probability density function of the reconstructions should be near unity. Deviations from unity may occur due to concentration errors between the size distribution measurement and the growth factor distribution measurement, unaccounted transmission losses, and errors from the inversion. Reconstructions deviating significantly from unity should be flagged and rejected.

A limitation of the above approach is that the forward model (and thus matrix $\mathbf{B}$) assumes that the larger multiply charged particles have the same growth factor frequency distribution as the smaller singly charged particles. This limitation can in principle be eliminated by specifying a 2D probability frequency distribution that also depends on dry diameter. Constructing an appropriate forward model that adds another integration dimension to Eq. (17) is straightforward. An inversion that solves for the 2D frequency distribution, similar to those performed elsewhere (Rawat et al., 2016; Sipkens et al., 2020), is feasible using the algorithms in *RegularizationTools.jl* and has been attempted by the author. In practice, however, this approach proved impractical. For example, using 10 dry diameters and a 30-bin size resolution results in a large inversion matrix. Adding an integration dimension to the forward model and recomputing this matrix for each scan significantly slows the inversion. Furthermore, interpolation is needed to estimate the growth factor frequency distribution for the multiply charged particles. The physical size of the multiply charged particles depends on their charge. For example, +2 charged particles are approximately

1.5 times larger than +1 charged particles. The diameter of the multiply charged particles will therefore not necessarily coincide with any of the 10 dry diameters selected for direct measurement. This introduces additional uncertainty due to assumptions that need to be made in the interpolation scheme. Errors from scans with low non-zero concentration at the edge of the size distribution propagate back into the inversion at other dry sizes. Finally, only a single size distribution can be used to compute the matrix $\mathbf{B}$. Collecting data for 10 dry sizes can take 20 min or longer, during which the aerosol size distribution may change, thus invalidating the use of a single inversion matrix. In situation where the temporal evolution of the size distribution is predictable, e.g. environmental chamber measurements, Kalman smoothing might be used to predict the in-between states (Ozon et al., 2021b, a). Although no exhaustive analysis was performed, the compounding errors during a 2D inversion seem to outweigh the benefits of relaxing the assumption that +2 and +3 charged particles have the same growth factor frequency distribution as the +1 charged particles.

## 4 Inversion of Real-World Data

### 4.1 Data Sources

#### 4.1.1 Bodega Bay Marine Laboratory

Aerosol size distribution data to contrast inversion schemes were obtained from measurements taken at Bodega Bay Marine Laboratory (39°18′25″N 123°3′58″W) between 16 January 2015 and 8 March 2015 as part of the Calwater 2/ACAPEX campaign. A subset of the data have been published by Atwood et al. (2019). Sample flow was brought into a mobile laboratory using an inlet, dried to $10 \pm 2\%$ relative humidity using a Nafion membrane drier, and brought to charge equilibrium using an X-ray source (TSI 3088, TSI Inc., Shoreview, MN, U.S.A.) prior to entering a cylindrical DMA column (TSI 3081). The DMA was configured to measure the size distribution in scanning mobility particle sizer mode. Voltage was scanned exponentially from 10 kV to 10 V over 300 s. A condensation particle counter (TSI 3771, flow rate = 1 L min$^{-1}$) and a cloud condensation nuclei counter (DMT Model 100, Droplet Measurement Technologies, Boulder, CO, U.S.A., flow rate = 0.3 L min$^{-1}$) were used to measure particle concentration downstream of the DMA. The sheath-to-sample flow rate in the DMA was 5:1.3 L min$^{-1}$. Raw DMA response distributions comprising CPC concentration vs. apparent +1 mobility diameter were constructed along a 120-bin, geometrically-stepped mobility grid. Response distributions are denoted as $\mathbb{r}$. The apparent +1 mobility diameter is computed from the centroid mobility selected by the DMA assuming that all particles are singly charged. The dynamic diameter range for this setup is from 12 to 550 nm. The inversion matrix $\mathbf{A}$ is computed using Eq. (11) for the diffusionally-broadened transfer function (Stolzenburg and McMurry, 2008) and transmission loss correction through the DMA (Reineking and Porstendörfer, 1986). Inclusion of these terms results in a more ill-posed inverse problem due to increasing overlap between the kernels (Kandlikar and Ramachandran, 1999). The DMA response functions were inverted using the $L_0 x_0 B_{[0,\infty]}$ and $L_2 B_{[0,\infty]}$ methods. The *a-priori* estimate for $L_0 x_0 B_{[0,\infty]}$ was taken to be $\mathbb{x}_0 = \mathbf{S}^{-1}\mathbb{r}$, where $\mathbf{S}$ is obtained by summing the rows of $\mathbf{A}$ and placing the results on the diagonal of $\mathbf{S}$ (Talukdar and Swihart, 2003). The method $L_0 x_0 B_{[0,\infty]}$ with $\mathbb{x}_0 = \mathbf{S}^{-1}\mathbb{r}$ is essentially equivalent to the method used by Petters (2018), where it was shown that the thus-inverted

spectra are similar to those output by the inversion algorithm employed by the commercial TSI Aerosol Instrument Manager software suite. Small differences between $L_0 x_0 B_{[0,\infty]}$ employed here and the approach of Petters (2018) include the use of generalized cross-validation instead of the L-curve method to search for the optimal regularization parameter and the method to eliminate negative values after inversion. Petters (2018) truncated negative values instead of using a least squares numerical solver as described in section 2.1.3.

### 4.1.2 Southern Great Plains Site

Aerosol size distribution and humidified tandem DMA data to illustrate the tandem DMA inversion schemes were taken from measurements made by the U.S. Department of Energy (DOE) Atmospheric Radiation Measurement (ARM) program. The Southern Great Plains (SGP) site is located in Lamont, OK, U.S.A. (Lat: 36.604937, Long:$-97.485561$) and placed in a rural continental setting that is surrounded by agricultural activity as well as oil and gas production. The aerosol evolution at the site is influenced by frequent new particle formation events (Hodshire et al., 2016; Chen et al., 2018; Marinescu et al., 2019). Number concentrations fluctuate in response to the nitrate and organic aerosol cycle on short time scales and synoptic weather variability on longer time scales. During winter months, the inorganic aerosol composition at the site is dominated by nitrate aerosols (Jefferson et al., 2017; Mahish et al., 2018) and hygroscopicity derived from scattering measurements is largest during those months (Jefferson et al., 2017).

The instruments/measurements are part of the Aerosol Observing System (AOS, Uin and Smith, 2020). The instruments are operated by DOE personnel and data are distributed through a publicly accessible archive. Size distributions were measured with a scanning mobility particle sizer (TSI Model 3936, Kuang, 2016). Data in the archive are already inverted and reported at 5-min intervals. Humidified DMA response functions were measured using a humidified tandem DMA (Model 3100, Brechtel Manufacturing, Inc., Hayward, CA, U.S.A.). The first and second DMA are operated at 5:0.63 L min$^{-1}$ and 5:1 L min$^{-1}$ sheath-to-sample flow ratio, respectively (Janek Uin, personal communication). The instrument measures the humidified mobility distribution function at 85% relative humidity for 50, 100, 150, 200, and 250 nm dry diameter particles. Typical data density results in 228 scans per day, with equal coverage for the five dry sizes. Pre-processing that is already applied to the archived data accounts for conversion between mobility and apparent mobility diameter, the size-dependent detector counting efficiency, and number count smearing during the scan resulting from insufficient particle counter response time. When divided by the dry diameter, the archived data correspond to the apparent growth factor distribution evaluated by the forward model in Eq. (17).

The matrix $\mathbf{B}$ was evaluated for each scan using the flow rates given above, the dimensions of the DMAs given in Lopez-Yglesias et al. (2014), and the aerosol size distribution measured by the co-located SMPS with the timestamp closest to the scan of the humidified tandem DMA. Typical time differences between the two instruments' scan times are between 1 and 3 min. The humidified size distribution was interpolated onto a discrete growth factor grid with 60 bins between $0.8 < g < 5.0$ to match the matrix $\mathbf{B}$. The data were then inverted using the $L_0 D_{1e-3} B_{[0,1]}$ method. The method $L_0 D_{1e-3} B_{[0,1]}$ was further constrained such that growth factors $< 1$ are disallowed. This is achieved by setting the upper bound to zero for bins with $gf < 1$. Growth factors less than unity can occur due to particle restructuring upon humidification (Mikhailov et al., 2004; Shingler et al., 2016) or evaporation during transit through the humidifier and second DMA. Both effects are assumed to be less important for

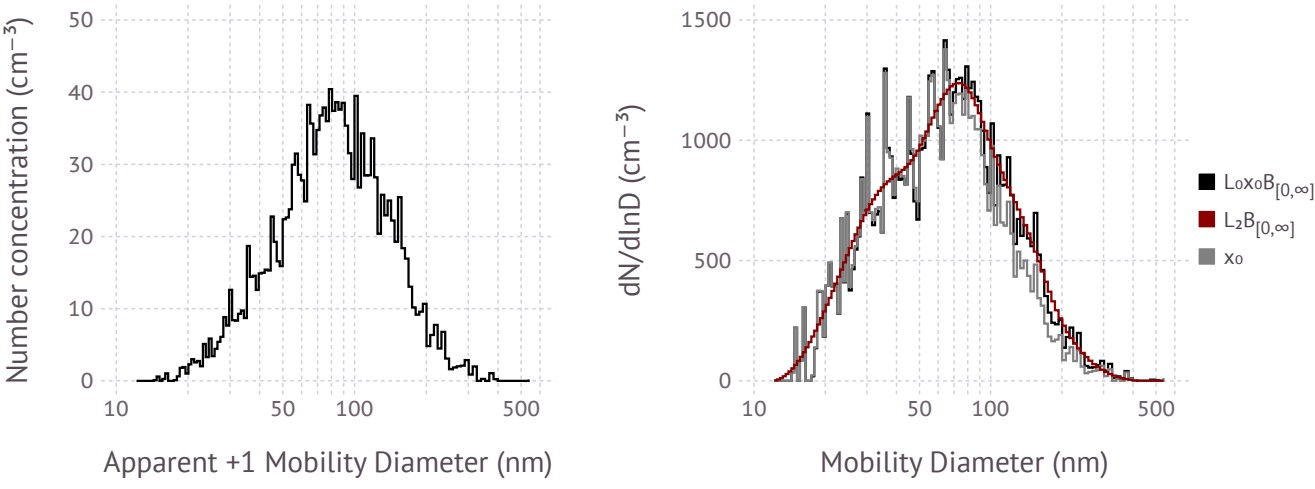

**Figure 5.** *Left:* Raw DMA response function for a single scan on 5 March 2015 at 10:40 UTC at Bodega Bay Marine Laboratory. *Right:* Inverted size distribution using the $L_0 x_0 B_{[0,\infty]}$ and $L_2 B_{[0,\infty]}$ method and the *a-priori* estimate of the solution $x_0 = \mathbf{S}^{-1} \mathbb{r}$.

ambient aerosol compared to the desire to constrain the inversion. In addition, the efficacy of the $LSQ_1$ and $LSQ_2$ methods for inverting the data was tested. For each scan, the root mean square error between the measured apparent growth factor distribution and the predicted growth factor distribution was evaluated for all three inversion approaches ($L_0 D_{1e-3} B_{[0,1]}$, $LSQ_1$, and $LSQ_2$). The method that resulted in the smallest residual was taken to be the inverted growth factor frequency distribution.

## 4.2 Results

### 4.2.1 Inversion of Size Distribution Data (Bodega Bay Marine Laboratory Site)

Figure 5 shows a real-world example size distribution response function gridded into 120 size bins. The total particle concentration is $\approx 2000$ cm$^{-3}$. The ragged structure is typically explained by random noise due to Poisson counting statistics. However, in this example the noise level is larger than Poisson counting statistics alone, which is thought to be due to the processing of raw data internal to the specific CPC model that was used to collect the data. At this diameter resolution, and with inclusion of the diffusion and loss terms in the forward model, the unregularized matrix inverse is entirely dominated by amplified random noise and is useless. The $L_0 x_0 B_{[0,\infty]}$ method converges to the solution with slight amplification of the random noise presented in the raw response function. The random noise is carried over into the *a-priori* estimate $x_0 = \mathbf{S}^{-1} \mathbb{r}$, which roughly represents the noise visible in the reconstructed solution. Nevertheless, $L_0 x_0 B_{[0,\infty]}$ is highly robust and unlikely to go astray, because $x_0$ is an excellent approximation of the solution at diameters less than 100 nm where singly charged particles dominate and is a good initial estimate for larger particles. Second order inversion using $L_2 B_{[0,\infty]}$ produces a smooth, denoised solution due to

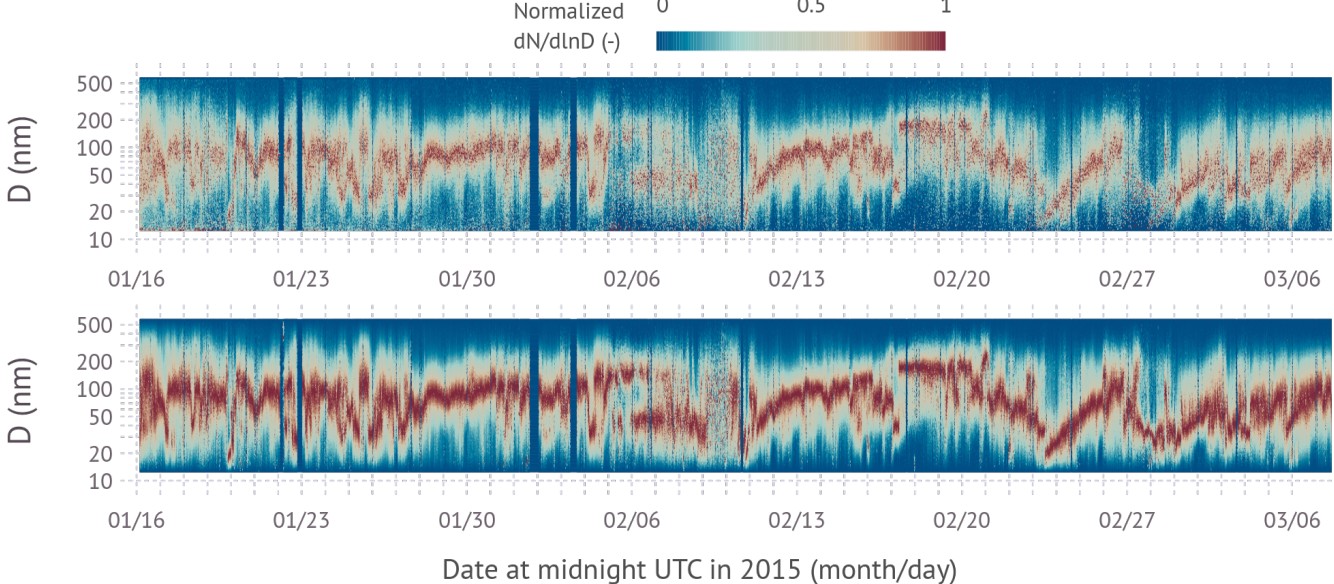

**Figure 6.** Time evolution of the normalized particle size distributions collected between 16 January and 7 March at Bodega Bay Marine Laboratory. The normalization is for each size distribution such that the maximum of the spectral density equals to unity. The red color visualizes the time evolution of the mode diameter of the dominant mode. Top panel: inverted using $L_0x_0B_{[0,\infty]}$; bottom panel: inverted using $L_2B_{[0,\infty]}$.

application of the derivative operator in the regularization filter matrix. The solution converges even though no *a-priori* estimate is used, i.e., $x_0 = 0$. Inclusion of an *a-priori* in the form of $L_2x_0B_{[0,\infty]}$ is possible. However, noise in the *a-priori* propagates into the solution, thus negating the intended benefit of the second order Tikhonov matrix. The algorithms specified in Section 2.1.2 significantly speed up the inversion relative to previous versions of the software (Petters, 2018). Wall-clock times on an i7-8559U CPU for the inversion of a single spectrum are 5 and 2 ms for $L_0x_0B_{[0,\infty]}$ and $L_2B_{[0,\infty]}$, respectively. This contrasts to 500 to 1000 ms required by the brute force algorithm – approximately equivalent to $L_0x_0B_{[0,\infty]}$ – used previously. Finding the global minimum of $V(\lambda)$ to identify the optimal regularization parameter also eliminates the occasional failure to converge when the L-curve algorithm is used (Petters, 2018). Either $L_0x_0B_{[0,\infty]}$ or $L_2B_{[0,\infty]}$, combined with generalized cross-validation, are suitable to be used in routine unsupervised inversion of size distribution data.

Figure 6 shows the time evolution of the normalized particle size distributions over a seven week period. The normalization is to highlight changes in the mode diameter(s). In general, the aerosol at the site is dominated by continental rural background conditions and the land-sea breeze circulation (Atwood et al., 2019). The timeseries is punctuated by aerosol transported from the California Central Valley to the site through the Petaluma Gap (Martin et al., 2017). Periods of low particle concentration occurred during the passage of an atmospheric river on $7-9$ February 2015 and a marine inflow event on $27-28$ February 2015. The atmospheric river brought heavy precipitation and marine airmasses from the southwest direction, while the marine inflow

event brought strong winds and precipitation free maritime air from the northwest direction. Several periods of prolonged modal growth were observed starting, e.g., 11 February 2015, 24 February 2015, and 1 March 2015. Figure 6 demonstrates the influence of inversion noise on visualizing the dynamic evolution of the size distribution. The denoised $L_2B_{[0,\infty]}$ solution significantly improves visualization of modes without the need to reduce the size resolution in the inversion. The signal is especially improved during low concentration periods during the atmospheric river passage and marine inflow event.

### 4.2.2   Inversion Humidified Tandem DMA Data (DOE ARM SGP Site)

Figure 7 shows real-world examples of growth factor frequency distributions for five dry sizes. Also shown for context is the evolution of the normalized aerosol number size distribution. Figure 7 shows dynamic evolution of the size distribution with sudden changes in mode diameter, several apparent new particle formation events, and several prolonged modal growth events. The distribution of the methods selected for best inversion was $LSQ_1$ (~5% of spectra), $LSQ_2$ (~50% of spectra), and $L_0D_{1e-3}B_{[0,1]}$ (~45% of spectra). In Fig. 7, the $LSQ_2$ inverted frequency distributions show a clean bimodal structure (2 colors per scan), while the $L_0D_{1e-3}B_{[0,1]}$ spectra appear more smeared. The 250 nm dry diameter data show a dominant contribution of more hygroscopic particles with $gf$ ~$1.5 - 1.6$ and a small contribution of less hygroscopic particles with $gf$ ~$1.05 - 1.2$. Similar trends are observed for 200, 150, and 100 nm particles. However, the hygroscopicity of the dominant mode decreases with decreasing diameter. The fraction of cases where a broad hygroscopicity frequency distribution is observed is larger than for the 250 nm particles. Notably, time periods with broad growth factor frequency distributions are observed at multiple sizes. For example, the period of $9 - 11$ February 2020 shows a broad frequency distribution at 100, 150, 200, and 250 nm dry diameter. Occasionally temporal trends in the hygroscopicity of the less hygroscopic mode are observed. For example, the growth factor of the less hygroscopic mode systematically increases on 20 February 2020 for 150, 200, and 250 nm particles, indicative of a chemical transformation of some, but not all, of the particles. The 50 nm dry particle hygroscopicity frequency distributions are also predominantly bimodal. However, the overall growth factor is significantly smaller, with most $gf < 1.2$.

## 5   Discussion, Summary, and Conclusions

*RegularizationTools.jl* is a general purpose software package to invert data using $L_2$-regularization. It is included as a supplement to this work and published as free software through the GNU General Public License. The package implements well-established numerical algorithms (Golub et al., 1979; Eldén, 1982; Bates et al., 1986; Hansen, 1998, 2000; Mogensen and Riseth, 2018) and filter matrices (Huckle and Sedlacek, 2012). Systems with up to ~1000 equations can be inverted. The upper limit is determined by the need to compute the generalized singular value decomposition of the design matrix and filter matrix, which has at minimum $O(n^2)$ time complexity. The time to compute the generalized singular value decomposition exceeds several 10s of seconds for systems exceeding 1000 equations. Iterative methods to support inversion of large-scale systems have been formulated (e.g. Lampe et al., 2012), but these are currently not implemented.

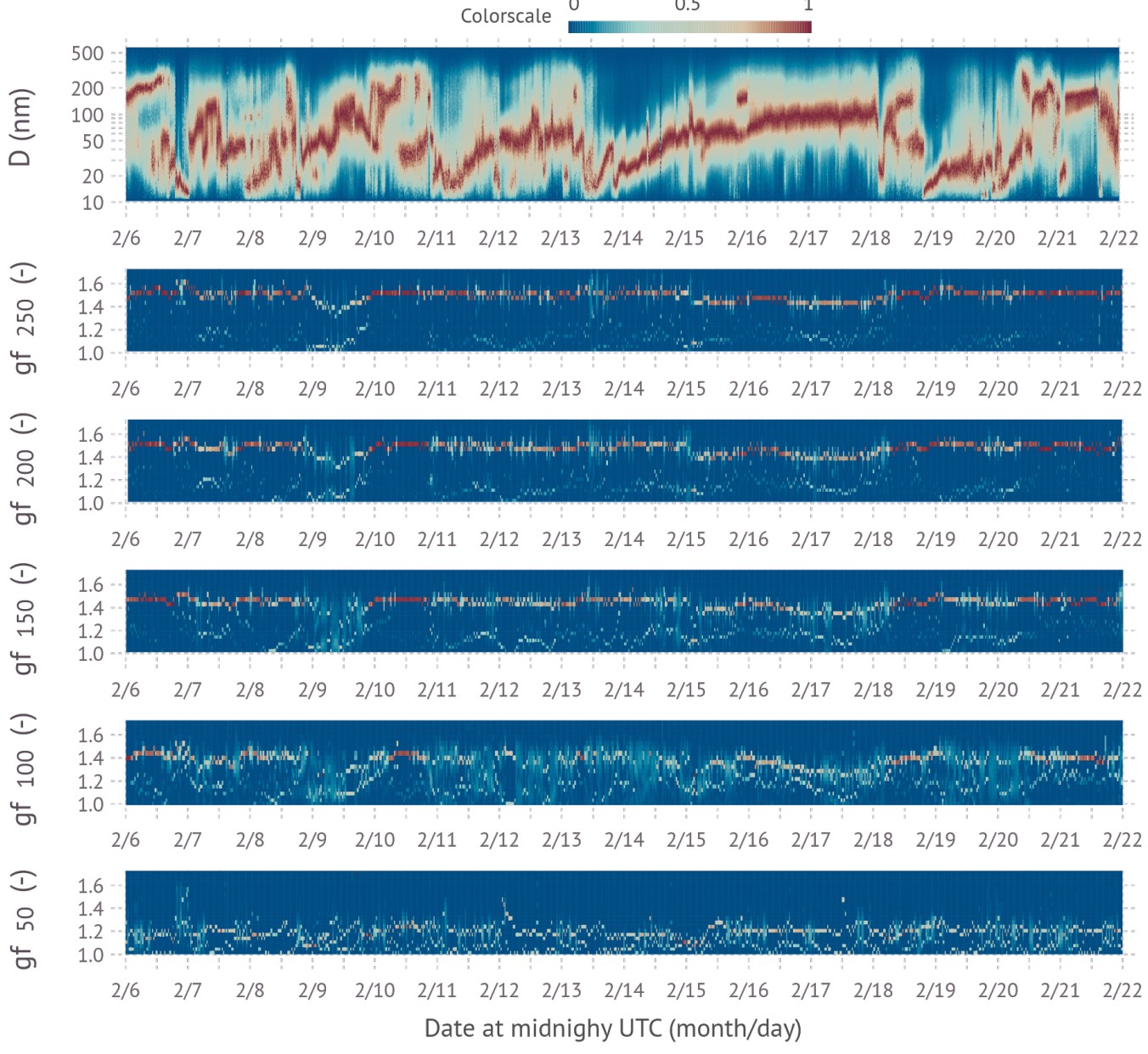

**Figure 7.** *Top:* Time evolution of the normalized particle size distributions collected between 6 February and 22 February at the Southern Great Plains research site. The normalization is for each size distribution such that the maximum of the spectral density equals to unity. *Second-Top to Bottom:* Inverted growth factor frequency distributions at 85% relative humidity for 250, 200, 150, 100, and 50 nm particles, respectively.

The software package can be used to simplify the prototyping of a wide variety of inverse problems that arise in science and engineering applications. Although the package does not add any novel regularization methods, it provides a systematic method to categorize inversion methods via the expression in Eq. (5). A total of 24 basic permutations can be combined with a set of hyperparameters to attempt the inversion of ill-posed problems. Hyperparameters include boundary constraints, values for *a-priori* estimates, and the lower bound $\epsilon$ for the Huckle and Sedlacek (2012) two-pass inversion approach. Users can define custom filter matrices and thus are able to further extend the number of methods. Equation (7) provides an example of a simplified interface that allows testing of different permutations with a simple function call. Furthermore, a generic interface is provided to translate arbitrary linear forward models defined by a computer function into the corresponding matrix of linear transformation. This obviates the need to explicitly write out the Fredholm integral equation and discretize it using the quadrature or the Galerkin method. For example, the forward model for transfer of a growth factor frequency distribution through the tandem DMA in Eq. (17) represents a triple integral and also contains a sum term for the multiple charges. Explicit discretization of this model would be tedious compared to the method employed here. As demonstrated in the documentation of the package, the generic interface can readily be used to solve other common inversion problems. Only a few lines of new code are needed to reproduce the essential core of the algorithm used in the unsupervised inversion of Lidar data (Müller et al., 2019), which involves the retrieval of a size distribution from multi-wavelength scattering and absorption data (see package documentation for code).

$L_2$-regularization is one of several techniques that is suitable to invert size distribution data (e.g. Voutilainen et al., 2001; Kandlikar and Ramachandran, 1999). The technique has been used previously for size-distribution inversion (e.g. Wolfenbarger and Seinfeld, 1990; Talukdar and Swihart, 2003; Petters, 2018). An advantage of this method is that data can be inverted when the number data of channels becomes large (Talukdar and Swihart, 2003). In contrast, Bayesian inversion schemes, which are not further discussed here, are suitable for uncertainty quantification (Voutilainen et al., 2001). To the author's knowledge the package *DifferentialMobilityAnalyzers.jl* is the only publicly available free-software for size distribution inversion from DMA data. This work extends the capabilities of that package. The $L_0 x_0 B_{[0,\infty]}$ and $L_2 B_{[0,\infty]}$ methods can be used with generalized cross-validation to perform fast unsupervised inversion of size distribution data. Convergence issues resulting from the use of the L-curve method used previously (Petters, 2018) are resolved by switching to the generalized cross-validation approach to find the optimal regularization parameter. Higher-order inversions resulting in smooth, denoised solutions are now available. It is expected that such denoised spectra will benefit unsupervised machine-learning approaches that seek to extract features from such datasets (e.g. Joutsensaari et al., 2018; Atwood et al., 2019), although this hypothesis has not been tested by the author. Revision of the numerical algorithms improves the speed of inversion by a factor ~200. The millisecond inversion speed for a single scan permits rapid inversion of large datasets and facilitates inversion in real-time during data acquisition on low-cost and low computational power hardware platforms. For example, the inversion has been tested on ARM Cortex A72/A53 64 bit reduced-instruction-set architecture used by the ROCKPro64 single board computer. The Julia language provides tier 1 support for this architecture. Julia binaries are available; *DifferentialMobilitityAnalyzers.jl* and *RegularizationTools.jl* compile and run without any modification. Inversion speeds on the order of several 10s of milliseconds are fast enough

on this inexpensive but relatively low-powered platform to permit embedding the inversion into the data acquisition and display software and running the inversion before each display update.

To the author's knowledge this is the first time $L_2$-regularization has been applied to the inversion of tandem DMA data. Inversion of simulated data shows that the an SMPS-style matrix-based inversion is possible, while also accounting for multiply charged particles. Application of solution constraints fixes the issue of oscillatory and negative solutions that were encountered with the matrix-based optimal estimation method used by Cubison et al. (2005). The 12 methods that include boundary constraints were systematically tested against five test cases. All of the methods performed similarly well when inverting frequency distributions. However, poor results were obtained when inverting narrow distributions or data produced by single compositions. The method $L_0 D_{1e-3} B_{[0,1]}$ is often, but not always, able to invert these data. For narrow distributions a nonlinear least squares fit with either one or two growth factors, termed $LSQ_1$ and $LSQ_2$, can fill this gap. Ambient data can be inverted by applying all three methods and then selecting the inversion with the smallest root mean square error between the data and the prediction. In contrast to previous inversion routines (Stolzenburg and McMurry, 1988; Cubison et al., 2005; Gysel et al., 2009), explicit knowledge of the aerosol size distribution is needed. These data can either be obtained using a co-located scanning mobility particle sizer, or by configuring the tandem DMA to also measure the size distribution every few scans. The resulting algorithm is unsupervised and nonparametric, i.e., it can be fully automated and does not require any *a-priori* assumption about the functional form of the growth factor frequency distribution. The speed of the inversion algorithm is much slower than for size distribution inversion for several reasons. For each scan, the matrix **B** must be recomputed to account for changes in the size distribution. This requires recomputing the generalized singular value decomposition for **B** and **L**, which is slow. Furthermore, three inversions are computed for each scan. The $LSQ_1$ and $LSQ_2$ methods use a gradient descent algorithm together with the forward model, which is slower than the matrix inverse. Nevertheless, a single day's worth of data can be inverted on a regular personal computer within a few minutes.

Application of the inversion to a 16-day dataset demonstrates that the thus-obtained growth factor frequency distribution data can reveal significant details about the mixing state of the aerosol. The inverted dataset is suitable as input to carry out common analyses made with growth factor frequency distributions. Examples include the characterization of the evolution of aerosol mixing state as a function of time, characterization of changes in growth factor with dry diameter and its relationship to chemical composition or characterization of the growth factor at the mode diameter of particles during modal growth events (Park et al., 2008; Wu et al., 2013; Jung and Kawamura, 2014). Additional examples include the decomposition of the hygroscopicity frequency distributions into distinct growth factor classes (Swietlicki et al., 2008), evaluation of the temporal trends of spectral concentration for hygroscopicity-resolved data (Royalty et al., 2017), evaluation of the accuracy of aerosol mass spectrometer measured (organic) mass concentration through hygroscopicity constraints (Jimenez et al., 2016), and inclusion of growth factor frequency distributions to account for mixing state in aerosol hygroscopicity to cloud condensation nuclei closure (Mahish et al., 2018).

*Code and data availability.*

1. SGP SMPS Data: Atmospheric Radiation Measurement (ARM) user facility. 2016, updated hourly. Scanning mobility particle sizer (AOSSMPS). 2020-01-01 to 2020-09-27, Southern Great Plains (SGP) Lamont, OK (Extended and Co-located with C1) (E13). Compiled by C. Kuang, C. Salwen, M. Boyer and A. Singh. ARM Data Center. Data set accessed 2020-09-29 at http://dx.doi.org/10.5439/1095583.

2. SGP HTDMA Data: Atmospheric Radiation Measurement (ARM) user facility. 2017, updated hourly. Humidified Tandem Differential Mobility Analyzer (AOSHTDMA). 2020-01-01 to 2020-02-22, Southern Great Plains (SGP) Lamont, OK (Extended and Co-located with C1) (E13). Compiled by J. Uin, C. Salwen and G. Senum. ARM Data Center. Data set accessed 2020-09-25 at http://dx.doi.org/10.5439/1095581.

3. Bodega Bay Preprocessed Data: Petters, Markus D., Rothfuss, Nicholas E., Taylor, Hans, Kreidenweis, Sonia M., DeMott, Paul J., and Atwood, Samuel A. (2019). Size-resolved cloud condensation nuclei data collected during the CalWater 2015 field campaign (Version v1.0) [Data set]. Zenodo. http://doi.org/10.5281/zenodo.2605668.

4. DifferentialMobilityAnalyzers.jl: A general purpose software package implementing the "Language to Simplify Computation of Differential Mobility Analyzer Response Functions" is available using the GPL-v3 license and is hosted is on GitHub. Version 2.5.6 was used in this work. The version of the software will be permanently archived with a doi upon acceptance of the manuscript for publication.
   Documentation: https://mdpetters.github.io/DifferentialMobilityAnalyzers.jl/stable/
   Source Code: https://github.com/mdpetters/DifferentialMobilityAnalyzers.jl

5. RegularizationTools.jl: A general purpose software package implementing Phillips-Twomey-Tikhonov Regularization. The package is available using the GPL-v3 license and source code and documentationa are hosted on GitHub. Version 0.4.1 was used in this work. The version of the software will be permanently archived with a doi upon acceptance of the manuscript for publication.
   Documentation: https://mdpetters.github.io/RegularizationTools.jl/stable/
   Source Code: https://github.com/mdpetters/RegularizationTools.jl

6. Source Code to reproduce the figures and derived datasets, and copies of the derived dartasets are temporarily hosted on GitHub and will be permanently archived on Zenodo with a doi upon final acceptance of the manuscript for publication.
   Source Code: https://github.com/mdpetters/softwarePackageSimplify2021

*Competing interests.* The author declares no competing interests.

*Acknowledgements.* This work was supported by the United States Department of Energy, Office of Science, Biological and Environment Research, grant number DE-SC 0021074 and NASA grant number 80NSSC19K0694. Data from the SGP site were obtained from the Atmospheric Radiation Measurement (ARM) Program sponsored by the U.S. Department of Energy, Office of Science, Office of Biological and Environmental Research, Climate and Environmental Sciences Division. I thank Janek Uin for providing additional information about the data. Size distribution data at Bodega Bay Marine Laboratory were collected with support from the National Science Foundation grant AGS-1450690. I thank Nicholas Rothfuss, Sam Atwood, and Hans Taylor for help operating the SMPS at Bodega Bay Marine Laboratory.

I thank Kimberly Prather, Sonia Kreidenweis, and Paul DeMott for logistical support during the field campaign. I thank Sarah Petters for helpful discussions. I thank Mark Stolzenburg for exceptionally helpful referee comments.

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
