# Peer review of "Revisiting Matrix-Based Inversion of SMPS and HTDMA Data"

_Atmospheric Measurement Techniques, 2021_

## Referee Comment (RC2)

**Referee Comments for Manuscript AMT-2021-51 "A Software Package to Simplify Tikhonov Regularization with Examples for Matrix-Based Inversion of SMPS and HTDMA Data"** Markus D Petters

Disclaimer:  Other than just some broad principles, this reviewer is not familiar with regularization techniques or the Julia syntax and is therefore ill-equipped to properly review the technical nature of that aspect of this work.  Attention is generally focused on other aspects of this paper.  Also, the lack of full comprehensive documentation of all the notation used in the equations presented here has frequently hampered a thorough understanding of these equations.  However, it is still possible to discern the general meaning of most equations.  Equation (10) is a good example of this.  The definition of the map() function and the interpretation of the right arrow ($\rightarrow$) are not given in the text here.  At least the map() function is defined in the Petters (2018) reference.  It appears the arrow notation is part of notation for a series or sequence.

This manuscript addresses the important issue of automating the processing of tandem DMA data.  The idea of inverting data with regularization is sound.  However, there are problems with the forward model of calculating system response from a known input distribution.  If these issues can be properly addressed, the resulting software package should prove of great utility.

**Major Comments**

There appears to be a problem with proper accounting of diffusional losses and broadening of the transfer function, $\Omega$, for DMA 2.  Eq. (10) in the form of $\mathbf{A}$ characterizes the transfer through DMA 1 while the equation for $\mathbf{O}$ (line 230) characterizes transfer through DMA 2.  As noted in Petters (2018), these two expressions are analogous except for the inclusion of $T_c$ in the former and the limitation of the summation to $k=1$ in the latter.  In the DMA, a particle is sized according to its apparent mobility diameter whereas diffusional losses as well as broadening of the transfer function are dependent on the true mobility diameter via particle diffusivity.  Given one of these diameters, the particle charge is required to calculate the other and ultimately $T_{size}^{\Lambda,\delta}$ .  Thus, it is important to sum over all charge states individually to calculate the diffusing transfer through a DMA.  As this is not done for the second DMA, the given expression cannot be properly accounting for transfer of multiply charged particles.

The interpretation of Eq. (11) and its components would be greatly facilitated by an explicit indication of the independent parameters of distribution for the input size distribution $\mathrm{n}^{cn}$.  Also, the precise form of $\mathrm{n}^{cn}$ (*e.g.* $dN/dD_\mathrm{p}$, $dN/d\mathrm{ln}D_\mathrm{p}$, or $dN/d\mathrm{log}D_\mathrm{p}$) is important.  The most obvious set of independent particle parameters would be (true) mobility diameter, $D_1$, and charge, $k$.  However, it appears that $\mathrm{n}^{cn}$ is distributed according to apparent mobility diameter,

$D_k$, and $k$ in order to have the balance of the equation work out. The apparent mobility diameter is then pre-multiplied by the effective, or apparent, growth factor, $gf_k(z^s, gf_0)$, and then by the ratio of true to apparent mobility diameters, $D_1/D_k$. However, this ratio is being evaluated at the DMA 1 centroid mobility, $z^s$, but applied to the $Z$ grid after growth. Since this ratio is a function of size, this does not work out. Also, this means that the input distribution to the **O** operator characterizing DMA 2 transfer is in terms of true mobility diameter, in contrast to the $\mathrm{n}^{cn}$ input to DMA 1 and **A**. All of this switching back and forth between true and apparent mobility diameter seems overly complicated.

**Minor Comments and Corrections**

line 158: Insert a space between "as" and "$x$".

line 186: The description of a DMA here is a bit too brief, saying nothing about the flow. Try "Charged particles in a flow between the electrodes are deflected to an exit slit ..."

lines 188-189: "The functions ... and tandem DMAs  are well understood ..."

line 200: "$T \cdot \mathrm{n}$" should be "$T. \cdot \mathrm{n}$" according to Petters (2018). Presumably $T$ is a vector, but this differs from the notation conventions given in lines 87-88.

Eq. (10): Here $T_{size}^{\Lambda, \delta}(k, z^s)$ alone characterizes transfer through the DMA. Evidently the balance of this expression puts this into the required form for later matrix manipulation. Some additional explanation of how this matrix is created from $T_{size}^{\Lambda, \delta}(k, z^s)$ would be useful here. And though perhaps only parentheses may be used in programming, the readability of this equation would be greatly improved by alternating "( )" with "[ ]" and "{ }".

lines 213, 230: Though $z^s$ is defined in lines 223-224, what is $z_k^s$ in the indicated lines? Since $z^s$ is used in Eq. (10), it might be more conveniently defined in line 211 (rather than line 224) along with $Z$ as "... $Z$ is a vector of centroid mobilities, $z^s$, scanned by the DMA ..."

lines 216-217: "The size distribution after passage through the DMA is given by $\mathrm{r} = \mathbf{A}\mathrm{n} + \varepsilon$, where $\mathrm{r}$ is the response function ... ." The size distribution exiting the DMA and the response of the detector are not the same thing. The former is usually given as $dN/d\log D_p$ while the latter, as in the case of a CPC, is given by $N_{CPC}$, a simple number concentration. Also, there is the matter of the detector efficiency as well as the transport efficiency between the DMA and the detector, unless the latter has been subsumed into the DMA transport efficiency. As **A** is to later serve as the operator corresponding to transfer through DMA 1 in a tandem DMA setup, $\mathbf{A}\mathrm{n}$ must represent a size distribution, not a response function.

Eq. (11) and following: The double character notation for growth factor as "$gf$" is atypical as far as normal mathematical notation is concerned. It is too easily interpreted as $g$ times $f$, rather than as a single parameter. And in this draft of the manuscript there is actually extra space between the two letters, increasing the likelihood of the wrong interpretation. However, it is seen that this space is eliminated in Petters (2018) so presumably it can and will be eliminated in the final typeset form. If not for this preexisting work and a strong preference to remain consistent with that, it would be better to change this to a single character form such as simply "$g$". Also, the reason for the choice of the $cn$ superscript on $n$ for the input distribution is quite obscure. Does that stand for something?

line 230: Given the length and complexity of the expression for the operator $\mathbf{O}$, it would be better placed on a line by itself and numbered.

Discretization: As noted (lines 461-462), the forward model for the TDMA represents a triple integral. The parameters of integration may be denoted as $D_i$ and $D_o$, the mobility diameters before and after growth, and $gf_0$, the size-independent growth factor. Though the discretization of these parameters is automated in the software, some discussion of the constraints on this discretization should be included here. For instance, is there a restriction between the number of particle diameter bins and the number of measurement bins? Eq. (15) and the statement (line 248) "The size of $\mathbf{A}_2$ is $n^2$, ..." would imply that the number of $gf$ bins must be equal to the number of measurement bins. Is this a necessary condition and, if so, why?

Figs. 1-4: Frequency vs. Growth Factor: Growth factor $gf$ and its frequency distribution $P_{gf}$ are naturally continuous functions, though the former is (artificially) discretized for the purposes of inversion. Just as the size distribution, $n$, is explicitly written as $dN/dD_p$ or $dN/d\ln D_p$ with total integral $N$, the growth factor frequency distribution is also a derivative, $dF/dgf$ or $dF/d\ln gf$, with total integral $F=1$. However, in the indicated plots, the frequency is plotted as for a parameter with truly discrete values such that the sum of the heights, rather than the areas, of the bars is equal to 1. That is, the height of each bar is given by $(dF/d\ln gf)\cdot\Delta\ln gf$ where $\Delta\ln gf$ is the width of the bar. If the growth factor is discretized such that $\Delta\ln gf$ is constant, then what is plotted is simply a uniformly scaled version of the more traditional $dF/d\ln gf$ plot, though this would normally be versus $\ln gf$. As plotted, the area under these curves is not equal to 1.

Number Concentration vs. Apparent Growth Factor: In these plots, the Apparent Growth Factor is evidently given by

$$gf_{app} = D_1\left(z_2^s\right)\big/D_1\left(z_1^s\right).$$

The "Concentration" parameter is apparently the first-order inverted number distribution function given by

$$dN_{app}/d\ln D_{p2} = (dN_{app}/d\ln Z_{p2})(d\ln Z_p/d\ln D_{p2}) = (N_{CPC}/\beta_2)(d\ln Z_p/d\ln D_{p2})$$

where $\beta_2 = Q_{aerosol}/Q_{sheath}$ for DMA 2.  This is also seen to be a scaled version of the apparent growth factor frequency distribution as

$$dN_{app}/d\ln D_{p2} = N_{t,2} \cdot (dF_{app}/d\ln gf_{app})$$

where $N_{t,2}$ is the total concentration exiting DMA 2.  If this is to be compared to the Frequency vs. Growth Factor plot, this would need to be multiplied by $\Delta \ln gf_{app} = \Delta \ln D_1(z_2^s)$.

For the two plots to be directly comparable, $\Delta \ln D_1(z_2^s)$ would have to be a constant.

line 315-316:  "… the residual is high  if the true input is a broad growth factor frequency distribution …"

lines 332-333:  "Errors from scans with low non-zero concentration at the edge of the size distribution propagate back into the inversion at other dry sizes."

line 345:  "… a cylindrical DMA column (TSI 3080)."  Model "3080" does not specify the actual DMA column.  Assuming it is the TSI long DMA, this should be specified as either "TSI 3080L" for the whole system or "TSI 3081" for just the column.

lines 386-387:  "… with the timestamp closest to the  scan …"  Eliminate "a".

line 419:  "… a marine inflow event on March 27–28 2015."  Use a date format consistent with the other dates, *i.e*. 27-28 March 2015.  However, this date is beyond the limits of the plot in Fig. 6.

line 422:  "… 9 February 2015, …".  Shouldn't this be 11 February 2015?

Lines 505-513:  "The inverted dataset … closure(Mahish et al., 2018)."  This is a very long run-on sentence.  It needs to be broken up into several sentences.

"Best fit" vs "good fit":  Though regularization produces what might be considered a best fit solution to the inversion problem, this does not necessarily imply it is a good fit.  It would be best to calculate a fit parameter such as the chi square of the normalized residuals over the degrees of freedom.  For a good fit, this should be near 1.  That is, the residuals are on the order of what is predicted by Poisson statistics.  Values an order of magnitude or more greater than that would suggest some sort of problem either with the dataset or the model.

---

## Author Comment (AC1)

*Response to Reviewer Comments*

*July 1, 2021*

*Author Statement*

I thank the referee for their time to review this manuscript and their constructive critiques. Below are itemized responses to the referees' comments. In response to the comments, Section 2.3 of the manuscript was significantly expanded. The complete revised section is mentioned in several responses. To avoid repetition, the revised section is reproduced at the end of the document.

*Response to Reviewer #1*

**Overview**
This manuscript presents a software package to invert aerosol size distributions from measurements, in particular from scanning mobility particle sizers (SMPSs), using the Tikhonov regularization approach.

This manuscript sits at the intersection of producing open-source, scientific code and presenting new scientific ideas. The reviewer admits this is an awkward position, as existing dissemination methods are not amendable to publishing well-maintained software, which is a critical component to modern, reproducible analysis. That being said, the scientific contributions of the underlying code are not hugely significant. Inversion of aerosol distributions using Tikhonov regularization is well-established. In fact, the author already notes one other instance of open-source software designed - at least in part - for this task (Hansen). (Other codes undoubtedly exist, though, admittedly most of these codes are closed source or not immediately available to the user, with very few exceptions, as the author notes.) Otherwise, this code does little to innovate on existing methods and is somewhat behind in terms of state-of-the-art, such as not presenting any form of uncertainty quantification - see Kandlikar and Ramachandran (1999); Voutilainen et al., (2001); and Voutilainen, Kolehmainen, Stratmann, & Kaipio (2001). The use of a GSVD to speed computation is insightful but is still based on existing literature. The code does also extend existing analysis tools to the Julia programming language, but it is this reviewer's opinion that this contributes little in terms of a novel scientific contribution.

Of note, the author could focus on the less-investigated HTDMA problem and the specific challenges that arise for that application (e.g., present the underlying integral equation for that scenario), which the authors note in the conclusion is one of the more novel aspects of this manuscript.

Altogether, this reviewer thinks the manuscript could be reoriented more towards novel scientific components, including more of a focus on HTDMA. As such, the author SHOULD be given the chance to respond to comments and refine the manuscript. MAJOR REVISIONS are recommended.
* * *
[1]*I do not disagree with the referee about the scientific novelty of the regularization code. The manuscript makes no claim in this regard. It also is not the purpose of the paper. I agree*     [1] *Response*

*that the most novel part of the work is the new HTDMA inversion. I have changed the title to "Revisiting Matrix-Based Inversion of SMPS and HTDMA Data" to reflect that. Uncertainty quantification described in the papers brought up by the referee is now included in the discussion.*

*That said, I will defend the work largely "as is" as a significant contribution in Atmospheric Measurement Techniques. Here is why. At issue is neither the mathematical novelty of Tikhonov regularization nor it's application to size distribution data. As mentioned by the referee, and in the draft manuscript, this has been demonstrated in the literature long before. The issue is about accessibility and extensibility of these techniques to the measurement community that is not trained in inverse problem solving. This is still true for size distribution inversion. Quoting from an anonymous referee of the preceding 2018 manuscript:*

> In computational work, graduate students and senior scientists tend to "reinvent the wheel." This wastes time and introduces errors. (In contrast, we happily use purchased instruments to make measurements sometimes with "black box" codes that contain errors that are exceedingly difficult to discover.)

*To perform size distribution inversion one can either use inflexible closed-source code, somehow be lucky enough to be handed down code from established laboratories and use them to ones own advantage, or write ones own. As stated by the referee quoted (it echos my opinion), the latter option is unrealistic for researchers that do not seek a career in inverse techniques, but simply want to make good measurements or develop instruments. This work reports on critical improvements to the open DifferentialMobilityAnalyzers.jl package that addresses this problem. It significantly improves inversion speed and extending the capabilities to higher order inversions. Those improvements are based on implementing high-performance algorithms. Reporting these improvements in the literature is valuable in its own right.*

*As discussed by Gysel et al. [2009], HTDMA inversions are complex to develop and not yet applied universally to data. The novelty and purpose of this work is to describe a methodology to tame the complexity developing inversion schemes and to provide a means to apply inversion to data for practitioners. Taming complexity is proposed to be achieved by three new ideas. First, the code systematically classifies regularization input assumptions and creates a simple interface for practitioners for trying out methods. I am not aware that such an interface is available anywhere else. Second, the work introduces a means to create design matrices from arbitrary forward models, although the details were only described in the supplementary material. Third, this work further extends the formalisms first introduced in the Petters 2018 paper, to show how it can be applied to HTDMA inversion. Breaking the problem into three independent parts should help prototyping and adapting future inversion approaches. The HTDMA inversion reported here is scientifically novel by addressing the limitations of oscillatory solutions reported in Cubison et al. [2005] and including multi-charge correction in the matrix. The conceptual and practical framework on how to approach the forward problem is novel.*

**Specific Comments**

² Referee

² The focus on the programming aspects also often distracts from the underlying science. For example, presenting the underlying mathematics in a programming language- and program-specific representation without the more standard mathematical forms (e.g., the underlying integral equations) makes the manuscript harder to follow. It seems that in an attempt to tread a line between a scientific manuscript and code documentation, the manuscript does not accomplish either task particularly well. In this respect, the manuscript may be better structured by clearly presenting the underlying scientific principles in a more standard mathematical notation, moving coding references out of the body of the manuscript. The alternative - presenting the manuscript as a form of documentation for a program - is better structured with specific coding examples in the text. However, this latter route is less amendable to a research article in AMT. In this case, another platform (a technical note in a journal or an article in a computational journal) may be more suitable. As a hybrid, the SI could be formally formed into documentation for the code that refers to the scientific principles in the base article without cluttering the body with code snippets and representations. Regardless, clarifications should be made before further review.

³ Response

³*I thank the referee for this comment, even though I disagree with it. Before responding in detail, I would like to list the revisions to section 2.3 made in response to the comment.*

1. **Included explicit references to the standard integral equations.**

2. **Clarified the purpose of the notation as a formal representation of the problem (they are not code snippets or code documentation; there is a separate detailed code documentation that is a supplement to this work).**

3. **Cleaned up the notation to limit it to standard computational concepts, i.e. eliminated parts that could be interpreted as programming language specific constructs.**

4. **Significantly expanded the text to aid parsing of the expressions.**

*Rationale: The expressions for the forward model presented in the manuscript are unconventional and were perceived by the referee as "code snippets". This is incorrect. What they really represent is a domain specific language comprising a set of simple building blocks that can be used to algebraically express the response functions intuitively through a form of pseudo code. The expressions evaluate to a deterministic answer and represent just a different form of mathematics. **The main advantage of this approach is that the expressions simultaneously encode the theory governing the transfer through the DMA and the algorithm to compute the solution.** The resulting expressions are concise. They are easily identified within actual source code. This makes the code easily modifiable by non-experts to change existing terms or add new convolution terms without the need to develop algorithms.*

*I want to elaborate on the computational viewpoint I have taken. The expressions evaluate in the same way than mathematical functions. The applied concepts are borrowed from the functional programming community and makes use of broadly understood concepts such as*

*lambda functions, generic functions, pure functions, higher order functions, function composition, and domain specific algebras. The expressions are a valid format to represent the mathematics. I have expanded the text to more carefully define each of the building blocks. I also recognize that these concepts are less widely used in the atmospheric community than the standard mathematical form. The expressions themselves can at first glance be more difficult to parse than the seemingly simple and familiar integral equations. Nevertheless, referee Mark Stolzenburg was able to follow the work (and call out two hidden assumptions) from the admittedly not-so-well written initial draft section. The assumptions had a very small effect on the final result, but of course it is important to address them when striving for correctness (which is done in the revised manuscript). This proves my point(s) above. It is trivial to recite the integral equations from one of the many preceding papers. Yet these equations do not fully communicate the model. The assumptions I made that were identified by Mark Stolzenburg would likely have never been detected in review, because the mathematical form is completely detached from the algorithmic solution. However, I understand the value of these equations and I now refer the reader to those works.*

*A disadvantage of the computational approach over the traditional mathematical approach is that algorithmic descriptions lack standardization of notation. This can blur the line between the pseudo code notation and language specific syntax. The reviewer brought to light that I had used some julia language specific constructs which I had introduced in the 2018 work. This is not ideal, because the expressions are really general and programming language independent. I therefore eliminated language specific constructs and only use generic functions that fall into the domain of general computing concepts. This results in more general expressions that are interpretable in most modern programming languages/syntax frameworks.*

*I firmly believe that the advantages of the computational approach outweigh their drawbacks. This work is in part an experiment on how to conceptionally model DMA transfer in the computational domain. It may in the end remain an obscure approach, and one that is not the preferred one by the referee or the majority of the field, but this is not a justification to hide it in a supplement or a computational journal. The work addresses atmospheric measurement techniques using computational concepts, not computational concepts themselves. Publication of this work is only adding to the list of available approaches; it does not force anyone adopt either the notation or approach.*

[4]**Please see revised section 2.3 at the end of this document.** [4] **Revision**

[5] In the abstract, the authors note that the inversion speed is improved by ~200 times down to 2-5 ms. Is the implication to work towards online inversion of the measurements? If not, there is a fair degree of flexibility in terms of inversion speed, such that speed may not be the only or the best metric gauging improvement. Can the authors comment? If the hope is for online inversion, can the authors comment on the interface with the instrument, which would be a substantial component of the overall process. [5] Referee

[6]*Improving performance in terms of speed is desirable as long as the inversion step presents some form of bottle neck for a particular application. Two example applications for inversion discussed in this manuscript involve either inversion of large data sets using different assumptions or inversion in real time during data acquisition. The quoted times are approaching the speed where the inversion bottle-neck disappears, although that will depend on the specific circumstance. As mentioned in the manuscript data acquisition and inversion on inexpensive reduced-instruction-set architecture is now possible. The interface to instrumentation depends on the user. We use julia as language to write all data acquisition software. The inversion is then just a function call to the software package(s) given as a supplement in this manuscript. The author shares the data acquisition software for scanning mobility particle sizers via GitHub (https://github.com/mdpetters/smpsDAQ) that is widely used in our laboratory. However, it is currently not well-documented. We mostly run the software on x86 and we are currently experimenting with running it on ARM v8 systems. Translating the approach to Python or other languages should be fairly easy.*

[6] *Response*

[7]**Since there is no peer-reviewed publication of the SMPS software, and since the response to the referee comment is publicly available, we do not discuss this further in the manuscript.**

[7] **Revision**

[8]Related to the above, this code is 200 times faster compared to what? A previous version of this code? It is worth noting that Tikhonov regularization for these distributions is a relatively straightforward problem, solving a simple linear system. As such, the speed improvements are likely linked to the external libraries that solve the linear system, something which the authors do imply later in the work. However, this does limit the novelty of using those methods for a different problem

[8] Referee

[9]*Yes, 200 times faster compared to a previous non-optimized regularized inverse. The speed improvement is due to the application of factorization techniques and implementing the numerical algorithms described in Section 2.1.2 instead of relying on the naive matrix inverse used in Petters (2018). The implemented algorithms are general. Virtually all languages, including julia, outsource basic linear algebra computations (e.g. the QR factorization) to highly performant external libraries (LAPACK, OpenBLAS, MKL) and the inversion speeds of these libraries are fairly similar. The reason that RegularizationTools.jl is distributed as a separate package is that it can be applied to any inversion problem, not just the DMA examples highlighted here. Examples for the generality of the approach are given in Section 2.1.4.*

[9] *Response*

[10]Line 93: What is the dimension/size of the different quantities defined here? Based on the subsequent discussion, it seems that A is assumed to be square (same reconstruction and measurement discretization). The A matrix is not required to be square, but this reviewer thinks it does make the GSVD simpler to compute (a

[10] Referee

non-square matrix may require special treatment) and should be stated clearly.

[11]*The matrix **A** does not need to be square. All algorithms are implemented to allow for non-square problems. An example for a non-square problem is given in the documentation to RegularizationTools.jl, which is a formal supplement to this paper as stated in the "Code and data availability." section. The relevant example for a non-square problem here:* [https://mdpetters.github.io/RegularizationTools.jl/stable/manual/#Creating-a-Design-Matrix](https://mdpetters.github.io/RegularizationTools.jl/stable/manual/#Creating-a-Design-Matrix) *under "Example 2". It is now stated that the A matrix need not to be square. The description for matrix A2 has been updated to describe the discretization, where it also mentions that the matrix does not need to be square. It was given as square due to the specific discretization scheme used to generate the figures in the draft.*

[11] *Response*

[12]**..., A is the design matrix (which may or may not be square), $x$ is the true quantity of interest, and ffl is the random error.**

[12] **Revision**

[13]**..., the matrix $A_2$ is understood to be computed for a specific input aerosol size distribution, and $\epsilon$ is a vector that denotes the random error that may be superimposed as a result of measurement uncertainties. The size of $A_2$ is $j \times n$.**

[13] **Revision**

[14]Line 108: Consider explicitly noting that automating the L-curve method, while feasible, is often more challenging than other automated methods and can be affected by noise and type of solver (which the authors admittedly imply later when they state that the L-curve algorithm used previously occasionally failed).

[14] Referee

[15]*Done.*

[15] *Response*

[16]**The optimal $\lambda$ occurs at the corner of the L-curve, which can be found algorithmically. However, automating the L-curve method can be more challenging than other automated methods, as further discussed below.**

[16] **Revision**

[17]Line 112: Clarify "standard form". What is the standard form? How would one compute this standard form? Under what conditions does one not use the standard form?

[17] Referee

[18]*Equation (3) is in standard form if $\mathbf{L} = \mathbf{I}$. (Stated a few lines above). The text around line 112 has been slightly reworded to make this clear.*

[18] *Response*

[19]**If L $\neq$ I, Eq. (3) is transformed to standard form using the generalized singular value decomposition of A and L as derived by Eldén (1982) and summarized by Hansen (1998).**

[19] **Revision**

[20]Line 208: Is Petters (2018) the best reference for this? The underlying equations

[20] Referee

for the discrete transfer function of the SMPS have been stated more formally many times before this work. If there is something specific in Petters (2018) about which the authors can be more explicit? There are also multiple ways to discretize the problem, which could be a route to a more specific representation from Petters. Further, why not present this in a more standard form, such as the transfer function given by Stolzenburg (2018), rather than a programming language-specific representation?

[21]Through Section 2.3: Similar to above, why not present all of the physics in terms of its underlying integration equations rather than language-specific concatenation and mapping operators or convolution "*" operators? For the discrete version, why not state these in terms of matrices instead? Interestingly, there are multiple ways to discretize the problem (e.g., finite element bases), which is also not detailed here. The HTDMA problem is based on a double convolution with three components to the underlying integral equation/kernel: 1) the transfer function of the first DMA, 2) a kernel describing the humidification process, and 3) the transfer function of the second DMA. This feature is not clear from the current reading.

[21] Referee

[22]*Combined response to 20 and 21. Section 2.3 has been significantly revised based on comments by Mark Stolzenburg and comments above. The method used for matrix generation is discussed in Section 2.2. The method is equivalent to the quadrature method, as discussed in the supplemental documentation (https://mdpetters.github.io/RegularizationTools.jl/stable/manual/#Creating-a-Design-Matrix).*

[22] *Response*

[23]**See revised section 2.3 at the end of this document.**

[23] **Revision**

[24]Line 268+: The current manuscript structure makes it challenging to ascertain the role of the 30 (or other) bins for the growth factor in the overall procedure. This reviewer would expect that the growth factor would contribute to another matrix that bridges the mobility distribution output by the first DMA to the mobility distribution input to the second DMA. In this respect, since the other components of the problem have a constant number of bins (at least this reviewer gathered as such), would it not make more sense to have a matrix with the same dimension/number of bins as the larger problem? Further, depending on whether one is inferring these quantities or not, this matrix could be combined with one or more of the DMA transfer function kernels and thus be pre-computed, with little effect on the overall computational effort. If one is inferring the growth factor, the structure of the problem deviates somewhat from the more general aerosol inversion problem, a fact that should be clarified. Namely, there will be at least two integrations (over the mobility distributions for each DMA) with an intermediary quantity that is being inferred. Then, there is also the question of the uncertainties in the input size distribution, which is measured independently, also inferred, or assumed. Overall, these definitions could be clarified.

[24] Referee

[25]*The discretization/structure of the HTDMA inversion problem is now explained in more detail. Uncertainty in the size distribution will propagate into* **A**$_2$*. This uncertainty is now mentioned in the manuscript.*

[26]**For purposes of the forward model, the mobility grid for DMA 1 is discretized at a resolution of *i* bins. Transmission through DMA is computed for a specified $z^s$ (the dry mobility), $g_0$ (the growth factor), and an input size distribution, which results in a vector *i* concentrations along this grid. If the input size distribution does not match the mobility grid the grids are merged through interpolation. The mobility grid for DMA 2 is discretized at a resolution of *j* bins. The transmitted and grown distribution from DMA 1 (*i* bins along the mobility axis of DMA 1) is interpolated onto the mobility grid of DMA 2. The outer integral in Eq. (17) is discretized into *n* bins that models $P_g$. If the output mobility of grid of DMA 2 does not match, the grids are merged through interpolation. The choice of *i*, *j*, *n*, the ranges of mobility grids for DMA 1, DMA 2, and the range of $P_g$ is only constrained by computing resources and a physically reasonable representation of the problem domain. Reasonable choices are $i = 120$, $j = n = 30$. The forward model is used to cast Eq. (17) into matrix form such that the humidified mobility distribution function is given by**

$$\mathrm{m}_t^{\delta_2} = \mathbf{A}_2 P_g + \epsilon \tag{18}$$

**where the subscript 2 specifies transmission through DMA 2, the matrix $\mathbf{A}_2$ is understood to be computed for a specific input aerosol size distribution, and $\epsilon$ is a vector that denotes the random error that may be superimposed as a result of measurement uncertainties. The size of $\mathbf{A}_2$ is $j \times n$. Uncertainties in the size distribution propagate into $\mathbf{A}_2$. The main influence of the error will be the relative fractions of +1, +2, and +3 charged particles. Assuming a random error of $\pm 20\%$ in concentration, the overall effect on the $\mathrm{m}_t^{\delta_2}$ is expected to small.**

[27]Line 276: The use of Poisson noise could be used to appropriately weight the data. Why was this not considered (i.e., use weighted least-squares instead of naïve least-squares)? One limitation is that measurements that span multiple orders of magnitude will result in numerical instabilities, such that a baseline amount of background noise may be required. Can the authors comment?

[28]*I have not tried weighted least-squares. It's plausible that it helps. However, Poisson noise may not be the only source of error in the measurement. (For examples, false counts from leaks in the line, fluctuations of RH in sample flow, flow rate fluctuations, electronic noise, etc. may all contribute to the error). It's not clear how to estimate the total error from data. Since L2-regularization works well for the problem there is no need to explore this approach.*

[29]Line 280: How often would this a priori information be known? In the experimental section to follow, there is a short phrase about this being computed using the inverse of the S matrix. Would it be worth noting this here? Also, what is S? This information does not seem to be immediately available.    [30]*For the HTDMA*

[29] Referee

[30] Response

*problem (line 280), the "a-priori estimate $x_0$ is taken to be the normalized apparent growth factor distribution, where the normalization ensures that the sum over all bins is unity." This information is derived from the measured data, so it is always available. This is now mentioned in the text. The* **S** *matrix is used to compute the a-priori guess for size distribution inversion (line 355). It is explained there how it is derived at the location where it is first introduced: "..., where S is obtained by summing the rows of* **A** *and placing the results on the diagonal of* **S** *(Talukdar and Swihart, 2003)."*

[31]**The *a-priori* estimate $x_0$ is taken to be the normalized apparent growth factor distribution derived from the measured response function, where the normalization ensures that the sum over all bins is unity**

[31] **Revision**

[32]Line 280: Continuing from above, do the choices for x_o make sense given the chosen Tikhonov matrix? For example, a first-differences Tikhonov matrix encodes information about the expected slope of the solution. Using (x - x_o) implies regularization of the slope of the residual with respect to an a priori estimate. Can the author comment?

[32] Referee

[33]*With the exception of the of $L_0D_{1e-3}B_{[0,1]}$ method, which is creates a Tikhonov matrix that is less sensitive to sharp edges, all of the methods worked similarly well when tested against simulated test data. For example the inversion using $L_2x_0B_{[0,1]}$ and $L_2B_{[0,1]}$ produces reconstructions of similar quality (see supplementary information). So empirically, inclusion of this particular a-priori $x_0$ does not make a difference when smoothing with derivative operators $L_1$ and $L_2$. I also experimented with $L_2x_0B_{[0,\infty]}$ for size distribution inversion (now discussed in the paper) and found that it works, though without the smoothing benefits. This is because for large regularization parameters, the solution converges toward the initial guess, regardless of the choice of* L. *Letting a-priori information through the filter may thus negate the benefit of smoothing.*

[33] *Response*

[34]**Second order inversion using $L_2B_{[0,\infty]}$ produces a smooth, denoised solution due to application of the derivative operator in the regularization filter matrix. The solution converges even though no *a-priori* estimate is used, i.e., $x_0 = 0$. Inclusion of an *a-priori* in the form of $L_2x_0B_{[0,\infty]}$ is possible. However, noise in the *a-priori* propagates into the solution, thus negating the intended benefit of the second order Tikhonov matrix.**

[34] **Revision**

[35]With respect to, "Higher resolution grids generally lead to poor performance even for method LoD1e−3B[0,1]." This is *slightly* surprising. Given the way Tikhonov prior operates, one may expect the extra grid points to be filled using the prior (a little like interpolating between lower resolution points, but not quite the same). Could this be an indication of limitations in the error metric used (there are most points at which the error is being calculated such that one is not comparing the same quantity)? Alternatively, the regularization parameter would change depending on the reconstruction grid. Was the regularization parameter re-optimized each time?

[35] Referee

[36]*The regularization parameter is optimized for each inversion. The effect is not due to the definition of the error metric. The effect of higher-resolution grids leading to poor performance is limited to the case with a single bin/sharp edges and using the two-step data-based regularization. The figure below shows and example of this for 120 bins and 30 bins and the same input distribution.*

[36] Response

[Figure]

*The two-step regularization technique first performs a reconstruction based on $L_0$ and the uses this to build a revised Tikhonov matrix. More bins generally lead to the same spread in the first reconstruction. Narrowing the solution down further is not possible based on that input. The text now clarifies that this only applies to the discrete resolution cases.*

[37]**Higher resolution grids generally lead to poor performance for discrete populations even for method $L_0 D_{1e-3} B_{[0,1]}$.**

[37] **Revision**

[38]Paragraph around Line 315: Is the unweighted residual really the best metric? How about measurement noise (one may have more confidence in some measure-

[38] Referee

ments than others)?  Should this be accommodated in terms of calculating this residual?

[39] *Please see a detailed response to referee Mark Stolzenburg for detailed discussion about why the RMSE was selected as residual (the last comment in my response to his comments). Weighing the RMSE by the measurement error is possible, but not desirable. Specifically, that would mean that bins with low or zero counts would effectively be excluded from the error estimation. However, there are cases where the model produces false oscillatory solutions (predicted counts) when measured (or expected counts) are zero. Filtering these in the error metric would bias the results.*

[39] *Response*

[40] Line 355: Small values in the A matrix do not matter as much as where they are located.  Small diagonals or nearly all-zero rows/columns are the real issues. Consider clarifying.  There is the issue of numerical noise (scattered small values) in the kernel, which does little but slow down the inversion. Was this dealt with?

[40] Referee

[41] *Thank you for pointing this out.  The language is revised (see below).  No attempt was made to filter numerical noise in the kernel.*

[41] *Response*

[42] **Inclusion of these terms results in a more ill-posed inverse problem due to increasing overlap between the kernels [Kandlikar and Ramachandran, 1999].**

[42] **Revision**

[43] Figure 5: The real-world noise in Fig. 5 does not seem to match noise in other number concentrations reported in the theoretical components of the work. Can the authors comment on this difference and/or update the earlier scenarios to be more representative?

[43] Referee

[44] *This is an excellent observation.  There are a couple of differences between the earlier scenarios and this real-world example.  The example is for size distribution measurement, while the previous scenarios are for growth factor measurements. However, I verified that it is true that random error in Fig. 5 is larger than what one predict from Poisson counting statistics alone. As mentioned earlier, there are other factors that may increase random noise in the data. In this particular case, the additional noise is related to the internal electronics of the specific CPC model.*

[44] *Response*

[Figure]

*The Figure shows a voltage scan from a DMA (TSI long column, 9:1 flow ratio, 120 s voltage scan) acquired with the same CPC model (TSI 3771/3772) as in the paper. Each bin corresponds to 1 second data. The two data streams are the digitized pulse output acquired using an external pulse counter card and the output from the serial port. (For the distributions in the paper, only serial port data were available). The serial port output is much noisier than the pulse count and the issue is present for all units of that particular model series. The pulse counts are more consistent with Poisson statistics. It is not entirely clear to me why the CPC serial output is so poor for this model. It seems to be related to the on-board processing of raw counts, which appears to be too slow. We identified this issue in 2016 and now always acquire both serial port and pulse data when available. Since this issue is related to a specific model and data acquisition mode there is no need to update the hypothetical HTDMA scenarios. Obviously noisier data is more difficult to invert. The observation that the noise exceeds the Poisson noise in the example is now is now mentioned in the text.*

[45]**The ragged structure is typically explained by random noise due to Poisson counting statistics. However, in this example the noise level is larger than Poisson statistics alone, which is thought to be due to the processing of raw data internal to the specific CPC model that was used to collect the data.**

[45] **Revision**

[46]For the temporally-evolving measurements, recent work by Ozon et al.

[46] Referee

(https://acp.copernicus.org/preprints/acp-2021-99/) presents an improvement to this existing technique and is closer to state-of-the-art. Can the authors comment and cite appropriately?

[47]*Thank you for the comment. The possibility is now mentioned. It is not clear though how this would work in ambient settings where conditions can change rapidly and unpredictably due to emissions or wind-direction changes. The possibility is mentioned in the revised manuscript.*

[47] *Response*

[48]**In situation where the temporal evolution of the size distribution is predictable, e.g. environmental chamber measurements, Kalman smoothing might be used to predict the in-between states [Ozon et al., 2021b,a].**

[48] **Revision**

[49]Line 460: Code would never involve writing out the Fredholm integral equations, making this statement a bit confusing. Further, scientific manuscripts supporting such code probably should state the underlying Fredholm integral equations. As before, program-specific language makes the scientific components of the article harder to follow.

[49] Referee

[50]*Please see my response to the earlier comment about the motivation for this approach.*

[50] *Response*

*Revised Section 2.3*

**Design Matrices For Differential Mobility Analyzers**

Differential mobility analyzers consist of two electrodes held at a constant- or time-varying electric potential. Cylindrical [Knutson and Whitby, 1975] and radial [Zhang et al., 1995, Russell et al., 1996] electrode geometries are the most common. Charged particles in a flow between the electrodes are deflected to an exit slit and measured by a suitable detector, usually a condensation particle counter. The fraction of particles carrying $k$ charges is described by a statistical distribution that is created by the charge conditioner used upstream of the DMA. The functions governing the transfer through bipolar charge conditioners, single DMAs, and tandem DMAs are well understood [Knutson and Whitby, 1975, Rader and McMurry, 1986, Reineking and Porstendörfer, 1986, Wang and Flagan, 1990, Stolzenburg and McMurry, 2008, Jiang et al., 2014].

The traditional mathematical formulation of transfer through the DMA is summarized in Stolzenburg and McMurry [2008] and references therein. Briefly, the integrated response downstream of the DMA operated at voltage $V_1$ is given by a single integral that includes a summation over all selected charges. The size distribution is measured by varying voltage $V_1$, which produces the raw response function defined as integrated response downstream of the DMA as a function of upstream voltage. The size distribution is found by inversion. The basic mathematical problem associated with inverting the response function to find the size distribution is summarized by Kandlikar and Ramachandran [1999]. The integral is discretized by quadrature to find the design matrix that maps the size distribution to the response function. $L_2$ regularization is one of several methods to reconstruct the size distribution from the response function [Voutilainen et al., 2001, Kandlikar and Ramachandran, 1999].

The integrated response downstream of a tandem DMA that is operated at voltages $V_1$ and $V_2$ is given by a double integral and the summation of all selected charges. The integrals are over the upstream size distribution and the aerosol conditioner function, which here is the growth factor frequency distribution. 
[revised manuscript text omitted]
), $g_0$ (the growth factor), and an input size distribution, which results in a vector $i$ concentrations along this grid. If the input size distribution does not match the mobility grid the grids are merged through interpolation. The mobility grid for DMA 2 is discretized at a resolution of $j$ bins. The transmitted and grown distribution from DMA 1 ($i$ bins along the mobility axis of DMA 1) is interpolated onto the mobility grid of DMA 2. The outer integral in Eq. (17) is discretized into $n$ bins that models $P_g$. If the output mobility of grid of DMA 2 does not match, the grids are merged through interpolation. The choice of $i$, $j$, $n$, the ranges of mobility grids for DMA 1, DMA 2, and the range of $P_g$ is only constrained by computing resources and a physically

reasonable representation of the problem domain. Reasonable choices are $i = 120$, $j = n = 30$. The forward model is used to cast Eq. (17) into matrix form such that the humidified mobility distribution function is given by

$$\mathrm{m}_t^{\delta_2} = \mathbf{A}_2 P_g + \epsilon \tag{18}$$

where the subscript 2 specifies transmission through DMA 2, the matrix $\mathbf{A}_2$ is understood to be computed for a specific input aerosol size distribution, and $\epsilon$ is a vector that denotes the random error that may be superimposed as a result of measurement uncertainties. The size of $\mathbf{A}_2$ is $j \times n$. Uncertainties in the size distribution propagate into $\mathbf{A}_2$. The main influence of the error will be the relative fraction of +1, +2, and +3 charged particles. Assuming a random error of $\pm 20\%$ in concentration, the overall effect on the $\mathrm{m}_t^{\delta_2}$ is expected to small.

---

## Author Comment (AC2)

**Response to Reviewer Comments**

*July 1, 2021*

**Author Statement**

I thank Dr. Stolzenburg for his time to review this manuscript and his constructive critiques. Below are itemized responses to the referees' comments. In response to the comments, Section 2.3 of the manuscript was significantly expanded. The complete revised section is mentioned in several responses. To avoid repetition, the revised section is reproduced at the end of the document.

**Response to Reviewer #2 (Mark Stolzenburg)**

**Overview**

[1] Disclaimer: Other than just some broad principles, this reviewer is not familiar with regularization techniques or the Julia syntax and is therefore ill-equipped to properly review the technical nature of that aspect of this work. Attention is generally focused on other aspects of this paper. Also, the lack of full comprehensive documentation of all the notation used in the equations presented here has frequently hampered a thorough understanding of these equations. However, it is still possible to discern the general meaning of most equations. Equation (10) is a good example of this. The definition of the map() function and the interpretation of the right arrow ($\rightarrow$) are not given in the text here. At least the map() function is defined in the Petters (2018) reference. It appears the arrow notation is part of notation for a series or sequence.

[1] Referee

[2] *I apologize for the missing definitions in the draft. These are now included in the revised version.*

[2] *Response*

[3] This manuscript addresses the important issue of automating the processing of tandem DMA data. The idea of inverting data with regularization is sound. However, there are problems with the forward model of calculating system response from a known input distribution. If these issues can be properly addressed, the resulting software package should prove of great utility.

[3] Referee

[4] *I thank Dr. Stolzenburg for his detailed and helpful review comments below. The issue raised regarding the forward model is addressed via a revision of the text and equations. The comments highlighted two assumptions that had no impact on the result, but were important to revise to be as correct as possible.*

[4] *Response*

**Major Comments**

[5] There appears to be a problem with proper accounting of diffusional losses and broadening of the transfer function, $\Omega$, for DMA2. Eq. (10) in the form of **A** characterizes the transfer through DMA1 while the equation for **O** (line 230) characterizes transfer through DMA2. As noted in Petters [2018], these two expressions are analogous except for the inclusion of $T_c$ in the former and the limitation of the summation to $k = 1$ in the latter. In the DMA, a particle is sized according to its apparent mobility diameter whereas diffusional losses as well as broadening of the transfer function are dependent on the true mobility diameter via particle diffusivity. Given one of these diameters, the particle charge is required to calculate the other and ultimately $T_{size}^{\Lambda,\delta}$. Thus, it is important to sum over all charge states individually to calculate the diffusing transfer through a DMA. As this is not done for the second DMA, the given expression cannot be properly accounting for transfer of multiply charged particles.

[5] Referee

[6]*Thank you for the raising this issue. In fact, the issue affects both the matrix **A** and the matrix **O**. The version in Petters (2018) and the draft manuscript compute the shape of the transfer function and losses for the mobility diameter corresponding to singly charged particles and then apply the same shape of the transfer function and diffusional loss to the multiply charged particles. The error that is introduced by this assumption/simplification is generally small since the fraction of multiply charged particles is small for sizes when diffusional broadening becomes important, and because the change in the shape of the transfer function/diffusional loss rate between the sizes is small. Nevertheless, there is no need to make this simplification. The formalism is now updated to properly account for the transfer of multiply charged particles.*

[6] Response

[7]**Section 2.3 of the manuscript has been revised to include this effect. Since this section includes changes in response to multiple other comments, please see the changed section at the end of this document for details.**

[7] Revision

[8] The interpretation of Eq. (11) and its components would be greatly facilitated by an explicit indication of the independent parameters of distribution for the input size distribution $\mathfrak{n}^{cn}$. Also, the precise form of $\mathfrak{n}^{cn}$ (e.g. $dN/dDp$, $dN/dlnDp$, or $dN/dlogDp$) is important. The most obvious set of independent particle parameters would be (true) mobility diameter, $D_1$, and charge, $k$. However, it appears that $\mathfrak{n}^{cn}$ is distributed according to apparent mobility diameter, $D_k$, and $k$ in order to have the balance of the equation work out. The apparent mobility diameter is then premultiplied by the effective, or apparent, growth factor, $gf_k(z^s, gf_0)$, and then by the ratio of true to apparent mobility diameters, $D_1/D_k$. However, this ratio is being evaluated at the DMA1 centroid mobility, $z^s$, but applied to the $Z$ grid after growth. Since this ratio is a function of size, this does not work out. Also, this means that the input distribution to the operator characterizing DMA2 transfer is in terms of true mobility diameter, in contrast to the $\mathfrak{n}^{cn}$ input to DMA1 and A. All of this switching back and forth between true and apparent mobility diameter

[8] Referee

seems overly complicated.

⁹*The precise form* $n^{cn}$ *is now clarified in the text just above the equation in question. It is a histogram in dN/dlnD units. The "cn" has been dropped based on a later comment. The distribution* $n$ *is along the actual mobility diameter. The referee is correct that the way it was formulated in the draft was confusing due to multiple switches between true and apparent mobility diameter. The referee is also correct that the ratio was being evaluated at the DMA1 centroid mobility,* $z^s$, *but applied to the Z grid after growth. The assumption was that particles within a charge grouping all behave the same. I changed the equation and the code to make it more intuitive and more correct, i.e. when projecting the physically grown diameter back to mobility space, the correction is applied for each point in the Z grid. The impact on the calculation due to the change is almost imperceptibly. Text has been added to help parsing the equation.*

$$\mathbb{r} = \mathbf{A}\mathbb{n} + \epsilon \tag{12}$$

where $\mathbb{r}$ is the response distribution, $\mathbb{n}$ is the true mobility size distribution, and $\epsilon$ is a vector denoting the random error that may be superimposed as a result of measurement uncertainties. Note that by design $\mathbb{n}$ and $\mathbb{r}$ are *SizeDistribution* objects, which represented the distribution as a histogram simultaneously as spectral density units (dN/dlnD) and concentration per bin units. The latter is the raw response function defined as integrated response downstream of the DMA as a function of upstream voltage (or corresponding $z^s$ or corresponding apparent +1 mobility diameter).

The mobility distribution exiting DMA 2 in the humidified tandem DMA is evaluated using the expressions

$$\mathbb{M}_k^{\delta_1} = \Pi_k^{\Lambda,\delta} \cdot \left\{ g_0 \cdot \left[ T_{size}^{\Lambda,\delta}(k,z^s) * \mathbb{n} \right] \right\} \tag{13}$$

In Eq. (13), $\mathbb{M}_k^{\delta_1}$ evaluates to the apparent +1 mobility distribution particles that exit the DMA$^{\Lambda,\delta}$ at the nominal setpoint-diameter defined by mobility $z^s$ (or z-star) in DMA 1 and particle charge $k$. Subscripts are used to differentiate DMA 1 and 2 which possibly have different geometries, flow rates, and grids, e.g. $\Lambda_1$, $\Lambda_2$ and $\delta_1$, $\delta_2$. $\Pi_k^{\Lambda,\delta}$ is the projection of particles having physical diameter $D$ and carrying $k$ charges onto the apparent +1 mobility grid. It is a function that converts each diameter/charge pair to mobility and interprets the result as apparent +1 mobility diameter. $g_0 = D_{wet}/D_{dry}$ is the true diameter growth factor, $D_{dry}$ is the selected diameter by DMA 1, $D_{wet}$ is the diameter after the humidifier, $T_{size}^{\Lambda,\delta}(k,z^s)$ is as in Eq. (10), and $\mathbb{n}$ is the mobility size distribution upstream of DMA 1.

To help parse Eq. (13), the product $T_{size}^{\Lambda,\delta}(k,z^s) * \mathbb{n}$ evaluates to the transmitted mobility distributions of particles carrying $k$ charges at the set-point mobility $z^s$ in DMA 1. The size distribution is grown by the growth factor $g_0$. The resulting size distribution is shifted to the apparent +1 mobility diameter using $\Pi_k^{\Lambda,\delta}$. Equation (13) differs from that in Petters [2018] where it was assumed that particles of all charges grow by the same amount. This is incorrect. Particles carrying

**more than a single charge alias at a smaller particle size [Gysel et al., 2009, Shen et al., 2021]. The effect is due to the size dependence of the slip-flow correction factor and captured through the function $\Pi_k^{\Lambda,\delta}$. Equation (13) assumes that $g_0$ applies to all particle sizes.**

**Minor Comments and Corrections**

[11] line 158: Insert a space between "as" and "x".                                              [11] Referee

[12]*Done.*                                                                                      [12] *Response*

[13] line 186: The description of a DMA here is a bit too brief, saying nothing about            [13] Referee
the flow. Try "Charged particles in a flow between the electrodes are deflected to
an exit slit..."

[14]*Thank you for the suggestion. Done.*                                                        [14] *Response*

[15] lines 188-189: "The functions ... and tandem DMAs  are well understood ..."    [15] Referee

[16]*Done.*                                                                                      [16] *Response*

[17]line 200: "$T\,\textbf{.}\,n$"should be "$T\,\textbf{..}\,n$" according to Petters (2018). Presumably $T$ is a    [17] Referee
vector, but this differs from the notation conventions given in lines 87-88.

[18]*Yes, "$T\,\textbf{.}\,n$"should be "$T\,\textbf{..}\,n$" according to Petters (2018). (T is a vector). The original    [18] *Response*
version was developed on julia v0.6 and it allowed me to create the "$T\,\textbf{..}\,n$" construct which
was desirable to create a consistent treatment of vectors. Once julia updated to 1.x series, it
was not longer possible to use this notation and I dropped the extra dot. This difference is
now noted in the text.*

[19]**See updated section 2.3**                                                                  [19] **Revision**

[20] Eq. (10): Here, $T_{size}^{\Lambda,\delta}$ alone characterizes transfer through the DMA. Evidently the    [20] Referee
balance of this expression puts this into the required form for later matrix manipu-
lation. Some additional explanation of how this matrix is created from $T_{size}^{\Lambda,\delta}$ would
be useful here. And though perhaps only parentheses may be used in program-
ming, the readability of this equation would be greatly improved by alternating "(
)" with "[ ]" and "{ }".

[21]*The equation(s) have been updated for readability by alternating "( )" with "[ ]" and "{*    [21] *Response*

*}". The equation has also been rewritten for clarity by removing julia language specific constructs and giving much more details about the functions used in the text. As pointed out in Petters (2018) "It may not be immediately obvious why the expression ... evaluates to the convolution matrix (or that it evaluates to a matrix at all). A step-by-step explanation is in Notebook S2." The reference to the supplement of the preceding work is still valid. Nevertheless, some additional explanation has also been added to the text.*

[22] **Please see entire updated section 2.3 for further context and definitions. The specific changes referring to this comment are**

[22] Revision

**Petters [2018] also gives an expression that evaluates to the convolution matrix for passage through a single DMA.**

$$\mathbf{A} = \text{mapreduce}\{z^{\mathbf{s}} \to \Sigma[k \to T_{size}^{\Lambda,\delta}(k,z^s), m]^T, \text{vcat}, Z\} \tag{11}$$

**where, $m$ is the upper number of multiply charged particles, $^T$ is the transpose operator, and $Z$ is a vector of centroid mobilities scanned by the DMA. Eq. (11) evaluates to the same as Eq. (8) in Petters (2018), but the notation is revised to be more general by removing the julia language specific splatting construct and replacing it with the widely used higher-order function mapreduce defined earlier.**

**To help with parsing the expression, $T_{size}^{\Lambda,\delta}(k,z^s)$ evaluates to a vector of transmission for $k$ charges and set point centroid mobility $z^s$ as a function of the entire mobility grid (e.g. 120 bins discretized between mobility $z_1$ and $z_2$). The function $\Sigma[k \to T_{size}^{\Lambda,\delta}(k,z^s), m]$ superimposes the vectors for all charges. Mapping $z^{\mathbf{s}} \to \Sigma[k \to T_{size}^{\Lambda,\delta}(k,z^s), m]$ over the mobility grid $Z$ produces an array of vectors, each corresponding to the transmission for a single size bin. Transposing the vectors and reducing the collection through concatenation produces the design matrix that links the mobility size distribution to the response function, i.e.**

$$\mathbb{r} = \mathbf{A}\mathbb{n} + \epsilon \tag{12}$$

**where $\mathbb{r}$ is the response distribution, $\mathbb{n}$ is the true mobility size distribution, and $\epsilon$ is a vector denoting the random error that may be superimposed as a result of measurement uncertainties. Note that by design $\mathbb{n}$ and $\mathbb{r}$ are *SizeDistribution* objects, which represented the distribution as a histogram in both spectral density units (dN/dlnD) and concentration per bin units.**

[23] lines 213, 230: Though $z^s$ is defined in lines 223-224, what is $z_k^s$ in the indicated lines? Since $z^s$ is used in Eq. (10), it might be more conveniently defined in line 211 (rather than line 224) along with Z as "... Z is a vector of centroid mobilities, $z^s$, scanned by the DMA..."

[23] Referee

[24]*The term $z_k^s$ has been rewritten as $z^s/k$ for clarity*

[24] Response

[25] lines 216-217: "The size distribution after passage through the DMA is given by $\mathbb{r} = \mathbf{A}\mathbb{m} + \epsilon$, where $\mathbb{r}$ is the response function... ." The size distribution exiting the DMA and the response of the detector are not the same thing. The former is usually given as $dN/dlogDp$ while the latter, as in the case of a CPC, is given by $N_{CPC}$, a simple number concentration. Also, there is the matter of the detector efficiency as well as the transport efficiency between the DMA and the detector, unless the latter has been subsumed into the DMA transport efficiency. As **A** is to later serve as the operator corresponding to transfer through DMA1 in a tandem DMA setup, **A** must represent a size distribution, not a response function.

[25] Referee

[26]*It is both. Please see revisions to preceding comment which spells this out. Currently the detector efficiency is not treated separately, but it can be easily added by adding terms to $T_{size}^{\Lambda,\delta}(k, z^s)$.*

[26] *Response*

[27] Eq. (11) and following: The double character notation for growth factor as "gf" is atypical as far as normal mathematical notation is concerned. It is too easily interpreted as g times f, rather than as a single parameter. And in this draft of the manuscript there is actually extra space between the two letters, increasing the likelihood of the wrong interpretation. However, it is seen that this space is eliminated in Petters (2018) so presumably it can and will be eliminated in the final typeset form. If not for this preexisting work and a strong preference to remain consistent with that, it would be better to change this to a single character form such as simply "g". Also, the reason for the choice of the cn superscript on $\mathbb{n}^{cn}$ for the input distribution is quite obscure. Does that stand for something?

[27] Referee

[28]*The origin for the "cn" superscript was carryover from the previous publication where it was desirable to distinguish the distribution that is measured with a CN counter as detector from a distribution that is measured with a CCN as detector. This way the true activated fraction could be conveniently modeled as $\mathbb{n}^{ccn}$ ./ $\mathbb{n}^{cn}$. The "cn" is not needed here and has been dropped; gf has been changed to g for clarity as suggested.*

[28] *Response*

[29]line 230: Given the length and complexity of the expression for the operator **O**, it would be better placed on a line by itself and numbered.

[29] Referee

[30]*Done*

[30] *Response*

[31]**See updated section 2.3**

[31] **Revision**

[32] Discretization: As noted (lines 461-462), the forward model for the TDMA rep-

[32] Referee

resents a triple integral. The parameters of integration may be denoted as $D_i$ and $D_o$, the mobility diameters before and after growth, and $gf_0$, the size-independent growth factor. Though the discretization of these parameters is automated in the software, some discussion of the constraints on this discretization should be included here. For instance, is there a restriction between the number of particle diameter bins and the number of measurement bins? Eq. (15) and the statement (line 248) "The size of $\mathbf{A}_2$ is $n^2$, ..." would imply that the number of $gf$ bins must be equal to the number of measurement bins. Is this a necessary condition and, if so, why?

[33] *Thank you for raising this. The under-the-hood binning procedure is now explained. The text reflected a choice I made when I wrote the code such that the measurement grid(s) can, but don't have to be interpolated onto any desired grid representation through interpolation (see below). It is worth pointing out here that the generic interface described in section 2.2 is designed such that the user can query the forward model at arbitrary points, which creates non-square matrices that link the grid of $P_g$ and that of DMA 2 measurement representation.*

[33] *Response*

[34]**Using the notation in section 2.2,**

[34] **Revision**

$$F(\mathbf{x}, \mathbf{c}) = \int_0^\infty P_g \left[ \sum_{k=1}^m \left( \mathbf{O}_k * \mathbb{M}_k^{\delta_1} \right) \right] dg_0 \qquad (17)$$

**where $\mathbf{x}$ is the true $P_g$ and the vector $\mathbf{c}$ of constraining parameters comprises the DMA setup $\Lambda_1, \Lambda_2, \delta_1, \delta_2$ and upstream size distribution $\mathbb{n}$. Computer code that creates a forward model for tandem DMAs has been added to the *DifferentialMobiltyAnalyzers.jl* package and is annotated in the documentation of the package. For purposes of the forward model, the mobility grid for DMA 1 is discretized at a resolution of $i$ bins. Transmission through DMA is computed for a specified $z^s$ (the dry mobility), $g_0$ (the growth factor), and an input size distribution, which results in a vector $i$ concentrations along this grid. If the input size distribution does not match the mobility grid the grids are merged through interpolation. The mobility grid for DMA 2 is discretized at a resolution of $j$ bins. The transmitted and grown distribution from DMA 1 ($i$ bins along the mobility axis of DMA 1) is interpolated onto the mobility grid of DMA 2. The outer integral in Eq. (17) is discretized into $n$ bins that models $P_g$. If the output mobility of grid of DMA 2 does not match, the grids are merged through interpolation. The choice of $i$, $j$, $n$, the ranges of mobility grids for DMA 1, DMA 2, and the range of $P_g$ is only constrained by computing resources and a physically reasonable representation of the problem domain. Reasonable choices are $i = 120$, $j = n = 30$. The forward model is used to cast Eq. (17) into matrix form such that the humidified mobility distribution function is given by**

$$\mathbb{m}_t^{\delta_2} = \mathbf{A}_2 P_g + \epsilon \qquad (18)$$

**where the subscript 2 specifies transmission through DMA 2, the matrix $\mathbf{A}_2$ is understood to be computed for a specific input aerosol size distribution, and $\epsilon$ is**

**a vector that denotes the random error that may be superimposed as a result of measurement uncertainties. The size of $A_2$ is $j \times n$.**

[35]Figs. 1-4: Frequency vs. Growth Factor: Growth factor $gf$ and its frequency distribution $P_{gf}$ are naturally continuous functions, though the former is (artificially) discretized for the purposes of inversion. Just as the size distribution, $\mathbf{n}^{cn}$, is explicitly written as $dN/dDp$ or $dN/dlnDp$ with total integral $N$, the growth factor frequency distribution is also a derivative, $dF/dg$ for $dF/dlngf$, with total integral $F = 1$. However, in the indicated plots, the frequency is plotted as for a parameter with truly discrete values such that the sum of the heights, rather than the areas, of the bars is equal to 1. That is, the height of each bar is given by $(dF/dlngf) \cdot lngf$ where $lngf$ is the width of the bar. If the growth factor is discretized such that lngf is constant, then what is plotted is simply a uniformly scaled version of the more traditional $dF/dlngf$ plot, though this would normally be versus $lngf$. As plotted, the area under these curves is not equal to 1.

[36]Number Concentration vs. Apparent Growth Factor: In these plots, the Apparent Growth Factor is evidently given by
$gf_{app} = D_1(z_2^s)/D_1(z_1^s)$.
The "Concentration" parameter is apparently the first-order inverted number distribution function given by
$dN_{app}/dlnD_{p2} = (dN_{app}/dlnZ_{p2})(dlnZ_{p2}/dlnD_{p2}) = (NCPC/\beta_2)(dlnZ_p/dlnD_{p2})$
where $\beta_2 = Q_{aerosol}/Q_{sheath}$ for DMA 2. This is also seen to be a scaled version of the apparent growth factor frequency distribution as
$dN_{app}/dlnD_{p2} = N_{t,2} \cdot (dF_{app}/dlngf_{app})$
where $N_{t,2}$ is the total concentration exiting DMA 2. If this is to be compared to the Frequency vs. Growth Factor plot, this would need to be multiplied by $lngf_{app} = \Delta lnD_1(z_2^s)$. For the two plots to be directly comparable, $\Delta lnD_1(z_2^s)$ would have to be a constant.

[37]*Combined response to 35 and 36. In the submitted draft, the concentration is the raw number concentration the detector would measure for that bin. Neither the concentration nor the frequency histograms were normalized by the bin width. I changed the revised version to show the probability density functions such that the area under the curve equals to 1. I want to retain the number concentration vs. apparent growth factor plots since the values represent the measurement. I clarify the representation in the text.*

[38]**Please see revised manuscript.**

[39]line 315-316:"...the residual is high  if the true input is a broad growth factor frequency distribution..."

[40]*Fixed*

[35] Referee

[36] Referee

[37] Response

[38] **Revision**

[39] Referee

[40] Response

[41] lines 332-333: "Errors from scans with low non-zero concentration at the edge of the size distribution propagate back into the inversion at other dry sizes."

[41] Referee

[42] *Fixed*

[42] *Response*

[43] line 345: "...a cylindrical DMA column (TSI 3080)."Model "3080"does not specify the actual DMA column. Assuming it is the TSI long DMA, this should be specified as either "TSI 3080L"for the whole system or "TSI 3081"for just the column.

[43] Referee

[44] *Thank you, we have just the column. Corrected.*

[44] *Response*

[45] lines 386-387: "...with the timestamp closest to the a̶ scan..."Eliminate "a"

[45] Referee

[46] *Corrected.*

[46] *Response*

[47] line 419: "...a marine inflow event on March 27–28 2015."Use a date format consistent with the other dates, i.e. 27-28 March 2015. However, this date is beyond the limits of the plot in Fig. 6.

[47] Referee

[48] *The date formats are now consistent. Also, the text should have been 27-28 **February** 2015, which is on the plot.*

[48] *Response*

[49] line 422: "...9 February 2015,...". Shouldn't this be 11 February 2015?

[49] Referee

[50] *Thank you. Corrected.*

[50] *Response*

[51] Lines 505-513: "The inverted dataset ...closure (Mahish et al., 2018)." This is a very long run-on sentence. It needs to be broken up into several sentences.

[51] Referee

[52] *Done.*

[52] *Response*

[53] "Best fit"vs "good fit": Though regularization produces what might be considered a best fit solution to the inversion problem, this does not necessarily imply it is a good fit. It would be best to calculate a fit parameter such as the chi square of the normalized residuals over the degrees of freedom. For a good fit, this should be near 1. That is, the residuals are on the order of what is predicted by Poisson

[53] Referee

statistics. Values an order of magnitude or more greater than that would suggest some sort of problem either with the dataset or the model.

[54]*It is correct that the best-fit is not necessarily a good fit. Worse, even a good fit may be a poor model. Unregularized regression can almost eliminate the residual, but produce estimation parameters that are extremely poor, even if the regression looks good. For example:* [54] Response

[Figure]

*The left panel shows a true input vector x. The middle panel on the left shows a response vector b (red) computed as $b = Ax + \epsilon$, where $\epsilon$ the some random error. The right panel shows the estimate $\hat{x} = A^{-1}b$, which is extremely poor. Computing the model response from the estimate, $A\hat{x}$, shows the best fit solution. If we only have the observations, we can only compute error metrics based on some residual between $A\hat{x}$ and b. The comment in question in refers to some form of Figure 4*

[Figure]

*where the left panel corresponds to $A\hat{x}$ (colors) and b (Input) in the example above, and the right panel to the comparison between x (Truth) and $\hat{x}$ (colors) in the example above. The question raised by this referee (and referee #1) is what the best error metric might be to determine the goodness of the fit and by extension and goodness of model reconstruction. I selected the root mean square error (RMSE), $RMSE = \sqrt{\sum \frac{(O_i - E_i)^2}{n}}$, where $O_i$ are the observed and $E_i$ the expected values. The RMSE is zero for a perfect fit and greater than zero for a less-than optimal fit.*

*Many goodness-of-fit statistical tests involve some form of the chi-square statistic. Computing chi-square as $\chi^2 = \sum \frac{(O_i - E_i)^2}{E_i}$, where $O_i$ are the observed and $E_i$ the expected values is not valid for many of the cases here, because the expected values can be zero. Any residual in a bin with zero expected value would immediately raise $\chi^2$ to infinity. Excluding bins with zero expected value would be incorrect, as it would not capture "bad" models that*

*predict output for zero bins. The referee's language "chi square of the normalized residuals over the degrees of freedom" seems to refer to the adjusted goodness-of-fit index, commonly abbreviated as AGFI [e.g. Sun, 2005], which for the same reason is not valid here.*

*The main desired property of a goodness-of-fit index is that it might provide a clean quantitative measure on when to reject a solution. The AGFI seemingly provides this information, with values near 1 indicating a good fit. Although the AGFI is not applicable here, the RMSE can be used in a similar manner. Values below a certain threshold indicate a good (or good enough) fit. The only difference is that the lower threshold value is not immediately clear.*

*The way RMSE is used here is in a relative comparison between $L_0 D_{1e-3} B_{[0,1]}$, $LSQ_1$, and $LSQ_2$. $LSQ_1$, and $LSQ_2$ are well-behaved and do not have oscillatory solutions. However, they will fail when true growth factor frequency distribution is broader than can be explained by one or two compounds. Conversely $L_0 D_{1e-3} B_{[0,1]}$ will have poor solution (oscillatory solution) when the true input distribution is narrow. Truncation of the negative values is what amplifies the RMSE in this case. Thus RMSE is not quite used to declare that the fit is good or that a model is valid. It is used to determine whether the input distribution is narrow enough to warrant fit to a single component, two component, or multicomponent model.*

*This still leaves the ultimate question unaddressed. How well can we trust the proposed (regularized) solution? As I argue in the manuscript, the simulations address this point.*

> Since the true noise-free input growth factor frequency distribution is known, the fidelity of the inversion can be evaluated by computing the root mean square error between the noise-free solution and the regularized solution. The figure shows that both inversion methods produce a root mean square error between 0.02 and 0.03. These values are typical for the of reconstruction (see supporting information). Visual evaluation of the agreement between the reconstruction and the input suggest that either method is suitable for inversion.

*Whether this is acceptable remains ultimately up to the user. I am skeptical that a statistical procedure such as AGFI (if it were applicable) would really help here. Tests should be performed to validate the physical plausibility of the solution. For examples, the mode of the apparent growth factor and the mode of the inverted growth factor should be similar. The retrieved growth factors should be physically plausible. The distribution of RMSE can be plotted for a large data set. Visual inspection of fits for large RMSE can be used to derive a threshold above which fits are automatically rejected. The text now mentions these quality assurance examples.*

[55]**Note, however, that the low residuals between the apparent growth factor distribution and the model do not automatically ensure that the algorithm a good or adequate solution. Additional tests should be performed to validate the physical plausibility of the solution. For example, the retrieved growth factors should be physically plausible at the applied relative humidity. The mode of the apparent growth factor distribution and the mode of the inverted growth factor distribution should be similar. A histogram of the root mean square error between**

can be plotted for a large data set. Visual inspection of fits for large root mean square error can be used to derive a threshold above which reconstructions are automatically rejected. The integrated probability density function of the reconstructions should be near unity. Deviations from unity may occur due to concentration errors between the size distribution measurement and the growth factor distribution measurement, unaccounted transmission losses, and errors from the inversion. Reconstructions deviating significantly from unity should be flagged and rejected.

*Revised Section 2.3*

**Design Matrices For Differential Mobility Analyzers**

Differential mobility analyzers consist of two electrodes held at a constant- or time-varying electric potential. Cylindrical [Knutson and Whitby, 1975] and radial [Zhang et al., 1995, Russell et al., 1996] electrode geometries are the most common. Charged particles in a flow between the electrodes are deflected to an exit slit and measured by a suitable detector, usually a condensation particle counter. The fraction of particles carrying $k$ charges is described by a statistical distribution that is created by the charge conditioner used upstream of the DMA. The functions governing the transfer through bipolar charge conditioners, single DMAs, and tandem DMAs are well understood [Knutson and Whitby, 1975, Rader and McMurry, 1986, Reineking and Porstendörfer, 1986, Wang and Flagan, 1990, Stolzenburg and McMurry, 2008, Jiang et al., 2014].

The traditional mathematical formulation of transfer through the DMA is summarized in Stolzenburg and McMurry [2008] and references therein. Briefly, the integrated response downstream of the DMA operated at voltage $V_1$ is given by a single integral that includes a summation over all selected charges. The size distribution is measured by varying voltage $V_1$, which produces the raw response function defined as integrated response downstream of the DMA as a function of upstream voltage. The size distribution is found by inversion. The basic mathematical problem associated with inverting the response function to find the size distribution is summarized by Kandlikar and Ramachandran [1999]. The integral is discretized by quadrature to find the design matrix that maps the size distribution to the response function. $L_2$ regularization is one of several methods to reconstruct the size distribution from the response function [Voutilainen et al., 2001, Kandlikar and Ramachandran, 1999].

The integrated response downstream of a tandem DMA that is operated at voltages $V_1$ and $V_2$ is given by a double integral and the summation of all selected charges. The integrals are over the upstream size distribution and the aerosol conditioner function, which here is the growth factor frequency distribution. 
[revised manuscript text omitted]
), $g_0$ (the growth factor), and an input size distribution, which results in a vector $i$ concentrations along this grid. If the input size distribution does not match the mobility grid the grids are merged through interpolation. The mobility grid for DMA 2 is discretized at a resolution of $j$ bins. The transmitted and grown distribution from DMA 1 ($i$ bins along the mobility axis of DMA 1) is interpolated onto the mobility grid of DMA 2. The outer integral in Eq. (17) is discretized into $n$ bins that models $P_g$. If the output mobility of grid of DMA 2 does not match, the grids are merged through interpolation. The choice of $i$, $j$, $n$, the ranges of mobility grids for DMA 1, DMA 2, and the range of $P_g$ is only constrained by computing resources and a physically

reasonable representation of the problem domain. Reasonable choices are $i = 120$, $j = n = 30$. The forward model is used to cast Eq. (17) into matrix form such that the humidified mobility distribution function is given by

$$\mathrm{m}_t^{\delta_2} = \mathbf{A}_2 P_g + \epsilon \tag{18}$$

where the subscript 2 specifies transmission through DMA 2, the matrix $\mathbf{A}_2$ is understood to be computed for a specific input aerosol size distribution, and $\epsilon$ is a vector that denotes the random error that may be superimposed as a result of measurement uncertainties. The size of $\mathbf{A}_2$ is $j \times n$. Uncertainties in the size distribution propagate into $\mathbf{A}_2$. The main influence of the error will be the relative fraction of +1, +2, and +3 charged particles. Assuming a random error of $\pm 20\%$ in concentration, the overall effect on the $\mathrm{m}_t^{\delta_2}$ is expected to small.

---

## Author Comment (AC3)

*Response to Reviewer Comments*

*August 3, 2021*

*Author Statement*

Dr. Stolzenburg contacted me via email and offered to send additional comments regarding Section 2.3 as written in my response to the original comments. He did so on July 10, 2021. I thank Dr. Stolzenburg for his additional effort (and very appreciated comments) to help me improve the clarity of the manuscript. Since I had not uploaded a revised version of the manuscript prior to receiving these comments, I respond to them here. The revised version of the manuscript that will be submitted to AMT will have taken these comments into account. A new version of section 2.3 is included at the end of this comment. It supersedes the version in AC1 and AC2.

*Response to Reviewer #2 (Mark Stolzenburg)*

The comments were sent in the form of an annotated pdf document. Below I transcribe the comments in the following form (1) Text passage that the comment refers to, (2) Verbatim annotated comment, (3) response to comment, (4) revisions. If the context is unclear from the text fragment, please see the full text on starting on pg. 13 of AC2 on the discussion site.

**Overview**

[1] The integrated response downstream of a tandem DMA that is operated at voltages $V_1$ and $V_2$ is given by a double integral and the summation of all selected charges. The integrals are over the upstream size distribution and the aerosol conditioner function, which here is the growth factor frequency distribution.

[1] Text

[2] The proper verbiage used with integrals is as follows: integral( f(x)dx ) is the integral of f(x) over x. f(x) is called the integrand. The integrals here are actually over the upstream particle size, or some substitute for it, and the grown or downstream particle size, or some substitute for that. The functions you specify represent only part of the integrand in each case given here. Even if you change the "over"s to "of"s, the statement is still very misleading given the many unaccounted for factors missing in the integrands. An integral over "the aerosol conditioner function" makes little sense since it is a function of both upstream and downstream sizes.

[2] Referee

[3] *Thank you.*

[3] *Response*

[4] **The integrated response downstream of a tandem DMA that is operated at voltages $V_1$ and $V_2$ requires solving integrals of the upstream particle size distribution over size and the grown particle size distribution over size. The integration must be repeated for each charge state.**

[4] Revision

[5]The objective is to find find a design matrix that maps the growth factor frequency distribution to the raw TDMA response function.

[5] Text

[6] Perhaps it would be clearer to say "For the forward calculation, the objective is ...". Correct "find find".

[6] Referee

[7] *Thank you.*

[7] *Response*

[8] **For the forward calculation, the objective is to find a design matrix that maps the growth factor frequency distribution to the raw TDMA response function.**

[8] **Revision**

[9]The resulting expressions are concise. They are easily identified within actual source code.

[9] Text

[10] For anyone familiar with the specific language used.

[10] Referee

[11]**The resulting expressions are concise. They are easily identified within actual source code when working through the examples provided with the package documentation.**

[11] **Revision**

[12]Size distributions encoded as a *SizeDistribution* composite data type.

[12] Text

[13] This is not a sentence, there is no verb. "Size distribution are encoded .."?

[13] Referee

[14]**Size distributions are represented as a histogram and internally stored in the form of the *SizeDistribution* composite data type.**

[14] **Revision**

[15]Composite data types combine multiple arrays into a single symbol for ease of use, facilitating faster experimental design and analysis. *SizeDistribution* consists of vectors of bin edges, bin midpoints, number concentration, log-normalized spectral density, and logarithmic bin widths.

[15] Text

[16]"+1 mobility diameter bin edges, bin midpoints". Since later many different "size" parameters are introduced and used, it is best to be specific here. As with the DMA transfer function, are corresponding +1 mobilities also part of this data type?

[16] Referee

[17]*I agree with the suggestion. Yes, the mobility grid is also included. In practice arrays of centroid mobilities and mobility bin edges are created, from which the +1 mobility diameter is computed.*

[17] *Response*

[18] **Composite data types combine multiple arrays into a single symbol for ease of use, facilitating faster experimental design and analysis. The size distribution data type *SizeDistribution* includes vectors of the selected mobility bins considered by the DMA, +1 mobility diameter bin edges and +1 mobility diameter bin midpoints computed from the mobility grid, number concentration, log-normalized spectral density, and logarithmic bin widths.**

[18] Revision

[19] (Note that  Petters (2018) used $T \cdot \dot{n}$  the elementwise scaling. The extra dot has  been dropped to stay consistent with the current software implementation).

[19] Text

[20] [The referee highlighted multiple places of dangling wording introduced during editing and made suggestions for improvements.]

[20] Referee

[21] **(Note that Petters (2018) used $T \cdot \dot{n}$ as the elementwise scaling. The extra dot has been dropped to stay consistent with the current software implementation).**

[21] Revision

[22] Functions are used to  expressions.

[22] Text

[23] Not all of the following reduce the dimension of the expression. Or are you talking about a reduction or compactness in notation? If so, use a different word to avoid confusion with later usage.

[23] Referee

[24] **Functions are used to evaluate expressions.**

[24] Revision

[25] If $f(X)$ evaluates to a vector, the sum is the sum of the vectors.

[25] Text

[26] Should this be $f(x)$ or are you intentionally using $X$ in indicate a vector?

[26] Referee

[27] **If $f(x)$ evaluates to a vector, the sum is the sum of the vectors.**

[27] Revision

[28] The function $map(f, x)$ applies f(x) to each element of vector x and returns a vector of results in the same order.

[28] Text

[29] Perhaps this should be $(f, X)$?

[29] Referee

[30] *Based on the above, no change.*

[31] The function *reduce*$(f, x)$ applies the bivariate function $f(x, y)$ to each element of $x$ and accumulates the result.

[32] From this and what I have read online, my understanding is that f(x,y) uses the result of its previous application and combines that with the next x value. Nearly every online example I can find only uses functions that treat x and y the same, e.g. f is given as simply "+" or "*" such that interchanging x and y has no effect. Since f can be a user-defined function, it need not be symmetric in x and y, e.g. f(x,y)=x2-y. The documentation does not make it clear whether such usage is allowed. However, if it is, the order of arguments in f(x,y) matters, that is, which is the result of the previous operation and which is the new x value. From online examples, I have gotten the impression that x is the previous result and y is the new x value. This makes the limited documentation such as given here quite misleading. Given no additional information, the most natural assumption would be that the new x value is associated with the x in f(x,y) and the previous result is associated with y. But this would be just the opposite of what is needed. At minimum, I would suggest writing this as "f(y,x) where y is the result of the preceding operation". Also according to an online example at jhub.com/julia, note that the for the first operation, x1 is the previous result and x2 is the new x.

[33] *The referee is correct that reduce is not associative. The order of operation matters. The more general version of reduce(f,x) is foldl(f,x) and foldr(f,x), which guarantee left or right associativity. In regular Julia programs, reduce(f,x) = foldl(f,x). However, there is no guarantee made by the language and any applied parallelism could break the expression. Using strictly associative folds is more precise. The change has been made throughout. The definition has been clarified as suggested by the referee.*

[34] **The function** $\text{foldl}(f, x)$ **applies the bivariate function** $f(a, x)$ **to each element of** $x$ **and accumulates the result, where** $a$ **represents the accumulated value. For the first element in** $x$**,** $a$ **is the neutral value. For example** $\text{foldl}(-, [1, 2, 3])$ **evaluates the function** $-(a, x)$ **and yields** $1 - 2 - 3 = -4$**. The function** $\text{mapfoldl}(f, g, x)$ **combines** map **and** foldl**.**

[35]

$$\mathbf{A} = \text{mapfoldl}\{z^s \rightarrow \Sigma[k \rightarrow T_{size}^{\Lambda, \delta}(k, z^s), m]^T, \text{vcat}, Z_s\} \tag{11}$$

[36] It applies function $f$ to each element in $x$, and then reduces the result using the bivariate function function $g(x, y)$.

[37] $g(y, x)$ or even $g(a, x)$ where a represents the accumulated value so far. See preceding note.

[37] Referee

[38] *Thanks for the suggestion.*

[38] Response

[39] **It applies function $f$ to each element in $x$ such that $y = f(x)$ and then reduces the result using the bivariate function function $g(a, y)$ where $a$ represents the accumulated value. For the first element $f(x)$, $a$ is the neutral value. For example, $\text{mapfoldl}(\text{sqrt}, -, [4, 16, 64])$ evaluates to $\text{foldl}(-, [2, 4, 8]) = 2 - 4 - 8 = -10$.**

[39] Revision

[40] The function $vcat(x, y)$ concatenates arrays x and y along one dimension.

[40] Text

[41] For clarity, shouldn't this be "first dimension" according to online documentation?

[41] Referee

[42] **The function $vcat(x, y)$ concatenates arrays x and y along the first dimension in Julia. However, other programming languages may concatenate along a different dimension as definition of horizontal and vertical is arbitrary.**

[42] Revision

[43] Petters (2018) gives a simple expressions that model transfer through the DMA.

[43] Text

[44] Either "a simple expression that models" or "simple expressions that model" with no preceding "a".

[44] Referee

[45] **Petters (2018) gives a simple expression that models transfer through the DMA.**

[45] Revision

[46]

$$T_{size}^{\Lambda, \delta}(k, z^s) = \Omega(Z, z^s/k, k). * T_c(k, D_{p,1}). * T_l(Z, k) \tag{10}$$

[46] Text

[47] The given dependence of omega does not make sense to me. Let z be an element of Z. For the mobility passed in a basic transfer calculation, it is the value of z/zs that matters. For the diffusion calculation, it is z/k that matters. zs/k does not get used directly. Though what is shown may not be technically wrong, it obscures the true dependencies. Simply (Z, zs, k) would be better, as in Eq. (15) for DMA 2.

[47] Referee

[48] *Note that revisions will be indicated after responding to the next few comment about this equation. The referee is correct that implementation of the function is $\Omega(Z, z^s, k)$. The charge state is needed to compute the diffusion coefficient. However, for the $T_{size}^{\Lambda, \delta}(k, z^s)$*

[48] Response

*function $z^s/k$ is the correct argument to produce the actually transmitted mobility size distribution. One way to think about this is to ask the question: what centroid mobility $z^s$ would I have to set the DMA to transmit particles carrying $k$ charges if they had only a single charge? Eq. (10) therefore represents the mobility size distribution transmitted. In Eq. (15) is $\Omega(Z, z^s, k)$, as all particles have the same mobility. This version allows using the same omega function in both cases.*

49

$$T_{size}^{\Lambda,\delta}(k, z^s) = \Omega(Z, z^s/k, k). * T_c(k, D_{p,1}). * T_l(Z, k) \tag{10}$$

49 Text

[50]What is Dp,1? Apparently, it is the true particle mobility (not electrical mobility) diameter, Dp,1=Dp(z,k=1), where z is an element of Z. If it is to be used, has it been defined elsewhere? Perhaps it is part of the definition of the size distribution composite data type. If so, it should be defined there. Tl (penetration efficiency, not loss) also depends on this diameter, Dp,1. Why is it not written that way? Otherwise, simply Tc(Z,k).

50 Referee

[51]Are the dots necessary here in ".*" or should they be dropped as in Eq. 13?

51 Referee

[52]In Eq. (10), $Z$ is a vector of mobilities -> "a vector of particle mobilities". As Z is apparently used for both particle mobilities as well as DMA centroid mobilities (see note below), it would be best to be clear to which it applies each time it is used.

52 Referee

[53] *Loss has been changed to "penetration efficiency". Yes, the dots are still required for the \* operator. They were only dropped for the · operator. The interpretation of $D_{p,1} = D_p(z, k = 1)$, where z is an element of Z is correct. Tl was written this way in the original draft and in Petters (2018), but changed during the first round of revisions. The change is reverted here to stay consistent with previous notation. I revised the text as follows to better explain how this equation works. The meaning of Z is also further clarified when discussing the discretization scheme near the end of the document.*

53 *Response*

[54] **Petters [2018] gives a simple expression that model transfer through the DMA. The function $T_{size}^{\Lambda,\delta}(k, z^s)$ evaluates to a vector representing the fraction of particles carrying $k$ charges that exit DMA$^{\Lambda,\delta}$ as a function of mobility**

54 **Revision**

$$T_{size}^{\Lambda,\delta}(k, z^s) = \Omega(Z, z^s/k, k). * T_c(k, D_{p,1}). * T_l(D_{p,1}) \tag{10}$$

**where $z^s$ is the centroid mobility selected by the DMA (determined by the voltage and DMA geometry), $Z$ is a vector of particle mobilities, $\Omega$ is the diffusing DMA transfer function [Stolzenburg and McMurry, 2008], $T_c$ is the charge frequency distribution [Wiedensohler, 1988], and $T_l$ is the diameter-dependent penetration efficiency [Reineking and Porstendörfer, 1986]. The diameter $D_{p,1} = D_p(z, k = 1)$, where $z$ is an element of $Z$. The function $\Omega$ has been updated from Petters (2018). The version in Petters (2018) computed the shape of the transfer function corresponding to singly charged particles and then applied the same shape of**

**the transfer function and diffusional loss to the multiply charged particles. The functional $\Omega$ depends on three arguments $\Omega(Z, z^s, k)$ [Stolzenburg and McMurry, 2008]. The charge state is used to compute the diffusion coefficient and thus account for diffusional losses and broadening of the transfer function for multiply charged particles.**

**The output of $T_{size}^{\Lambda,\delta}(k, z^s)$ is the transmission of particles through the DMA in terms of the true particle mobility diameter. This is achieved by passing $z^s/k$ as argument to $\Omega$, which corresponds to the centroid mobility setting for the DMA to transmit particles with the size of particles with $k$ charges under the assumption that they carry only a single charge. The net result is that $D_{p,1} = D_p(z, k = 1)$, where $z$ is an element of $Z$ becomes equal to the true mobility diameter axis. As a consequence the charge fraction $T_c(k, D_{p,1})$ and penetration efficiency $T_l(D_{p,1})$ are evaluated at the correct diameter. The function $T_{size}^{\Lambda,\delta}(1, z^s)$ evaluates to a vector of the same length as $Z$. Performing an elementwise sum over all $T_{size}^{\Lambda,\delta}(k, z^s)$ produces the net mobility distribution transmitted by the DMA. Examples for $T_{size}^{\Lambda,\delta}(1, z^s)$, $T_{size}^{\Lambda,\delta}(2, z^s)$, and $T_{size}^{\Lambda,\delta}(3, z^s)$ is shown in Figure 2, right panel in Petters (2018). Note that Eq. (10) can be evaluated using arbitrarily discretized $Z$ vectors.**

[55] corresponding to singly charged particles and then apply the same shape of the transfer function

[56] "applied"

[57] **fixed in the revised paragraph above.**

[58] Petters (2018) also gives an expression that evaluates to the convolution matrix for passage through a single DMA.

[59] Summation over k means the information on particle physical diameter of multiply charged particles is
lost. Eq. (11) will not work for the first DMA in a TDMA setup using diffusing transfer functions or other diffusion effects in the second DMA. Something should be included here to indicate that, or that this expression is only for something like an SMPS system. I feel that "single DMA" is just not sufficient. The reference to "apparent +1 mobility diameter" at the end of this topic is useful in making this point but does not really make the point of "only apparent diameter, not physical" and comes far too late to make the point in question.

[60] *Correct. The information is now explicitly included*

[61] **Petters (2018) also gives an expression that evaluates to the convolution matrix for passage through a single DMA that is valid in the context of size distribution measurement system, e.g. SMPS. Since the expression includes a summation over all charges, the information on particle physical diameter of multiply charged particles is lost.**

[61] Revision

[62]

$$\mathbf{A} = \text{mapreduce}\{z^s \rightarrow \Sigma[k \rightarrow T_{size}^{\Lambda,\delta}(k, z^s), m]^T, \text{vcat}, Z\} \tag{11}$$

[62] Text

[63]I feel it would be cleaner to first define a convolution matrix for transport through a DMA, with no summation over k, as in Eq. 15, perhaps Okv or better yet Ov(k) where v (actually nu) is the DMA. This could then be used for DMA 2 as well below, using an analogous definition of Tsize. Here, for use as the convolution matrix through a single DMA, Av = sumk( Ov(k) ). Otherwise, in the current development of the matrices, the similarity of treatment for DMA 1 and DMA 2 is buried. At the very least, though hardly preferable, Eq. 11 should be rearranged to put the summation as the outermost operation on the right side. This would at least provide a little more symmetry of treatment of the two DMAs.

[63] Referee

[64] *I do not fully disagree with the referee about the potential elegance of making the expressions more symmetric. However, the matrices* **A** *and* **O** *serve two separate purposes. The former is valid for passage through a single DMA that is valid in the context of size distribution measurement system (which is now clarified), the latter is valid for evaluating the response after passage through the tandem DMA, starting with the transmitted distribution. The expressions for these are well-defined. It should be clear by now that the approach is in principle highly expressive. The version here, or the version in Petters (2018) are two examples how to write* **A**. *More ways certainly exist, perhaps even more elegant ways. In the context of this work, I believe that this approach will lead to further distance the expressions from the original work and in the end result in more confusion rather than additional clarity.*

[64] *Response*

[65] ... and Z is a vector of centroid mobilities scanned by the DMA.

[65] Text

[66]Z=Zs (see following note) implies **A** is a square matrix. Do you really want to introduce that restriction at this point?

[66] Referee

[67]Eq. 10 used Z as a vector of particle mobilities. Are these two vectors identical, that is, the DMA centroid mobilities and input particle mobility bin midpoints? If so, that should be explicitly noted to avoid confusion. Better yet would be to change Z here to Zs noting somewhere previously that Z=Zs.

[67] Referee

[68] *Although I have been using the same Z vectors in Eq. (10) and (11) - and thus square matrices - this is not a necessary restriction. This is also further clarified when discussing the discretization scheme.*

[68] *Response*

[69] **... and $Z_s$ is a vector of centroid mobilities scanned by the DMA. The matrix is square if $Z_s = Z$ in Eq. 10. However, this is not a necessary restriction.**

[69] **Revision**

[70] ...removing the julia specific splatting...

[70] Text

[71] Julia documentation uses this as capitalized, "Julia". Also, this should be hyphenated as "Julia-specific".

[71] Referee

[72] *Changed capitalization here and throughout the work.*

[72] *Response*

[73] **...removing the Julia-specific splatting...**

[73] **Revision**

[74] To help with parsing the expression, $T_{size}^{\Lambda,\delta}(k,z^s)$ evaluates to a vector of transmission for $k$ charges and set point centroid mobility $z^s$ as a function of the entire mobility grid (e.g. 120 bins discretized between mobility $z_1$ and $z_2$).

[74] Text

[75] So Tsize is actually a matrix of size [n x m] where n=120 is the number of particle mobility bins. As the above equations are defined it would be useful to clearly indicate the dimensions of the operands and the result.

[75] Referee

[76] "entire particle mobility grid". Though Z serves a dual purpose, in this context it is particle mobility.

[76] Referee

[77] *$T_{size}^{\Lambda,\delta}(1,z^s)$ evaluates to a vector of length Z; $T_{size}^{\Lambda,\delta}(2,z^s)$ evaluates to a vector of length Z. The text prior to this paragraph is updated to better explain the origin and dimensionality $T_{size}^{\Lambda,\delta}(k,z^s)$.*

[77] *Response*

[78] **See revisions to previous comments.**

[78] **Revision**

[79] Note that by design n and r are *SizeDistribution* objects, which represented the distribution as a histogram in both spectral density units (dN/dlnD) and concentration per bin units.

[79] Text

[80] The input distribution, (blackboard "n"), is readily defined in terms of the true mobility (k=1) diameter, Dp1. However, information on the original charge state exciting the charger is lost from the response distribution (blackboard "r") leaving

[80] Referee

only electrical mobility, z, as the size parameter. Thus, the response distribution cannot accurately be converted to dN/dlnD. Any such conversion must neglect the true charge distribution.

[81] *That is correct. Nonetheless the mathematical/computational representation is that of a SizeDistribution object. The limitation is now noted.*

[82] **By design `n` and `r` are *SizeDistribution* objects, which represent the distribution as a histogram in both spectral density units (dN/dlnD) and concentration per bin units. The latter is the raw response function defined as integrated response downstream of the DMA as a function of upstream voltage (or corresponding $z^s$ or apparent +1 mobility diameter but not true physical diameter for multiply charged particles). Note, however, that the response function is not a true particle size distribution in the scientific sense since information about multiply charged particles is lost. The representation of `r` as *SizeDistribution* object is to allow response functions to used in the expression-based framework used here.**

[83]The latter is the raw response function defined as integrated response downstream of the DMA as a function of upstream voltage (or corresponding $z^s$ or corresponding apparent +1 mobility diameter).

[84]The first "corresponding" is sufficient for both, delete second "corresponding". Perhaps one could add here ", but not true physical diameter for multiply charged particles". Just a thought as to how to make this point clear.

[85] *Thank you for the suggestion.*

[86] **The latter is the raw response function defined as integrated response downstream of the DMA as a function of upstream voltage (or corresponding $z^s$ or apparent +1 mobility diameter but not true physical diameter for multiply charged particles).**

[87]

$$\mathbb{M}_k^{\delta_1} = \Pi_k \cdot \left\{ g_0 \cdot \left[ T_{size}^{\Lambda,\delta}(k, z^s) * \mathbb{n} \right] \right\} \tag{13}$$

In Eq. (13), $\mathbb{M}_k^{\delta_1}$ evaluates to the apparent +1 mobility distribution particles that exit the DMA$^{\Lambda,\delta}$ at the nominal setpoint-diameter defined by mobility $z^s$ (or z-star) in DMA 1 and particle charge $k$.

[88]$\Pi_k$: Given the appending of superscripts to this function in the following text, it

should appear the same here with (capital lambda)1

[89] $T_{size}^{\Lambda,\delta}(k, z^s)$: The superscripts should have subscript 1 corresponding to the subscript 1 of the left side of the equation.

[89] Referee

[90] You say that this is for DMA 1 but there should be an index/subscript of 1 on the superscripts of the first "DMA" in this sentence and of Tsize in the above equation to indicate this. Otherwise, there is no clear relationship between (delta)1 on the left side of the equation and (delta) (no subscript) on the right side. And (capital lambda)1 should also have a subscript of 1.

[90] Referee

[91] DMA 1 and 2 which possibly have different geometries, flow rates, and grids, e.g. $\Lambda_1$, $\Lambda_2$ and $\delta_1$, $\delta_2$.

[91] Text

[92] You introduce this notation here but then fail to properly apply it in Eq. (13) and following.

[92] Referee

[93] *Thank you for pointing this out. Superscripts have been added in the appropriate places.*

[93] *Response*

[94] $\Pi_k^{\Lambda,\delta}$ is the projection of particles having physical diameter $D$ and carrying $k$ charges onto the apparent +1 mobility grid.

[94] Text

[95] This function needs to interpolate the diameters of the grown particles onto the +1 mobility diameter grid corresponding to Z. From your description, it sounds like multiply charged particles are mis-sized to smaller diameters. If this function is used with storing the apparent diameter, rather than true diameter, in the distribution, then there would have to be a later function just to undo that before use in Eq. 14.

[95] Referee

[96] This function involves both calculation of apparent mobility but also interpolation of the grown bin sizes from Z back onto Z. This interpolation step is important as it affects the propagation of the random error of the input distribution. As such, it should be explicit noted here. In general, any interpolation of a noisy distribution or decedents there of will tend to reduce the overall noise level, but not, I believe, in a very predictable way.

[96] Referee

[97] $\Pi_k^{\Lambda,\delta}$ *calculates the apparent +1 diameter of a particle that carries multiple charges. The implementation is quite simple. Start with a physical size e.g. 100 nm. Next compute the mobility of that particle given it's charge state. Next, reinterpret that mobility as if the particle were to carry only a single charge. Finally divide this by the initial diameter to get the projection. Obviously* $\Pi_1^{\Lambda,\delta} = 1$. *Less obviously* $\Pi_2^{\Lambda,\delta} < 1$. $\Pi_k^{\Lambda,\delta}$ *does not do any interpolation. In the expression,* $\left\{ g_0 \cdot \left[ T_{size}^{\Lambda,\delta}(k, z^s) * \mathbb{n} \right] \right\}$ *evaluates to the grown transmitted size distribution.* $\Pi_k \cdot \left\{ g_0 \cdot \left[ T_{size}^{\Lambda,\delta}(k, z^s) * \mathbb{n} \right] \right\}$ *evaluates to the apparent size*

[97] *Response*

*distribution. The · operator is what shifts the sizes. If a misfit occurs during the shift (which invariably happens), the result in interpolated onto the original size grid of* m. *The role of interpolation is now discussed together with the discretization scheme (see further below).*

[98] Also, this function does NOT depend on the DMA configuration (capital lambda)1. It does depend on (delta)1, including the subscript 1.

[98] Referee

[99] *It does depend on the DMA configuration, which includes temperature and pressure, because the conversion from mobility to diameter, and hence the projection is temperature and pressure dependent.*

[99] *Response*

[100] ...humidifier, $T_{size}^{\Lambda,\delta}(k,z^s)$ is as in Eq. (10), and m is the mobility size distribution upstream of DMA 1.
To help parse Eq. (13), the product $T_{size}^{\Lambda,\delta}(k,z^s) * $ m evaluates to...

[100] Text

[101] Superscripts should have subscripts 1.

[101] Referee

[102] *Fixed*

[102] *Response*

[103] Equation (13) differs from that in Petters [2018] where it was assumed that particles of all charges grow by the same amount. This is incorrect. Particles carrying more than a single charge alias at a smaller particle size [Gysel et al., 2009, Shen et al., 2021]. The effect is due to the size dependence of the slip-flow correction factor and captured through the function $\Pi_k^{\Lambda,\delta}$. Equation (13) assumes that $g_0$ applies to all particle sizes.

[103] Text

[104] This does not make sense unless g0 is a function of particle size, or of zs and k. If this is the case, it has not been made clear. Otherwise, it seems that both here and in Petters it is assumed that particles of all charges grow by the same amount, g0, a constant.

[104] Referee

[105] *Yes, here and in Petters it is assumed that particles of all charges grow by the same amount, g0, a constant. It is said explicitly in " Equation (13) assumes that $g_0$ applies to all particle sizes." The initial wording was rather poor and the text is revised.*

[105] *Response*

[106] **Equation (13) differs from that in Petters [2018] where it was assumed that the apparent growth factor for particles carrying multiple charges is the same as for single charged particles. This is incorrect. Particles carrying more than a single charge alias at a smaller particle size [Gysel et al., 2009, Shen et al., 2021]. The effect is due to the size dependence of the slip-flow correction factor and captured**

[106] **Revision**

**through the function $\Pi_k^{\Lambda,\delta}$. Equation (13) assumes that $g_0$ applies to all particle sizes.**

[107]

$$\mathbf{O}_k = \text{mapreduce}\{z^s \rightarrow [\Omega^{\Lambda_2,\delta_2}(Z,z^s,k) .* T_l^{\Lambda_2,\delta_2}(Z,k)]^T, \text{vcat}, Z\} \qquad (15)$$

[107] Text

[108]zs has been used as the centroid mobility for both DMAs. These need to be distinguished as two different parameters. Use subscripts "1" and "2" on these as appropriate to avoid confusion, e.g. here and Eq. 13. There are others in the text as well.

[108] Referee

[109]Is the range of the DMA 2 scan really the same as the range of the input distribution when the former uses 4 times fewer bins? The same Z is used for both.

[109] Referee

[110] *I clarified Z as Zs (same as your earlier comment). Subscript has been added as suggested. The range in Z is arbitrary. Clarification about how the various Z and Zs relate to the discretization has been added to the text (see further below).*

[110] *Response*

[111]**Please see revised section.**

[111] **Revision**

[112] Equations (14) and (15) modified from those in Petters (2018)

[112] Text

[113]"are modified" or "have been modified"

[113] Referee

[114] **Equations (14) and (15) have been modified from those in Petters (2018)**

[114] **Revision**

[115] ... matrix corresponding to singly charged particles and then apply the same matrix ...

[115] Text

[116]Referee suggested: "applied"

[116] Referee

[117] **... matrix corresponding to singly charged particles and then applied the same matrix ...**

[117] **Revision**

[118] If the aerosol is externally mixed, the humidified distribution function is given by ...

[118] Text

[119]"function exiting DMA 2". Otherwise, it sounds like the distribution entering

[119] Referee

DMA 2, right after humidification.

[120] **If the aerosol is externally mixed, the humidified distribution function exiting DMA 2 is given by ...**

[121] ... where $P_g$ is the growth factor probability density function ...

[122] This still doesn't say anything about the growth factor being a function of dry diameter. So is Pg the same for all dry particle sizes, including those of multiply charged particles?

[123] *Yes, it is. There is explicit discussion later in the text on how to potentially relax this assumption by using 2D inversions.*

*The following few comments are given without direct response. They all relate to the discretization of the grid and the underlying interpolation scheme. In response (see below) the paragraph was revised to clarify. The individual comments referenced here have been taken in to account.*

[124] For purposes of the forward model, the mobility grid for DMA 1 is discretized at a resolution of *i* bins. Transmission through DMA is computed for a specified $z_s$

[125] This is not clear - "DMA 2", "DMA1" or "both DMAs"? From the following text, it would appear that what is needed and being described is the "transmission through DMA1 and subsequent growth". That is, up to the point of entering DMA 2. Note, simply changing "DMA" to "DMA1" does not work as DMA 1 transmission does not depend on go.

[126] If the input size distribution does not match the mobility grid the grids are merged through interpolation.

[127] Presumably, the input size distribution bins are interpolated onto diameter bins corresponding to the Z bins. Saying they are "merged" is ambiguous as to which is interpolated onto the other.

[128] The transmitted and grown distribution from DMA 1 (*i* bins along the mobility axis of DMA 1) is interpolated onto the mobility grid of DMA 2.

[129] It seems to me the DMA 2 mobility grid must be dynamically set according to how much growth there is. This would be good to note here. Wouldn't it be simpler

to just have the DMA 2 grid be a subset of the DMA 1 grid? However, if the setups of the two DMAs are such that they do not have the same non-diffusing resolution, Qa/Qsh, then perhaps it would be better to use grids of different resolutions. As noted before, interpolations tend to smooth the data, thereby confounding the error analysis.

[130] *First, in this framework, the DMA 2 mobility grid is arbitrary and fixed. Interpolation is used throughout. The effect is factored into the framework through the size distribution operators, specifically the · operator. Potential smoothing effects are factored into the framework through the numerical tests.*

    *The reason for using interpolation throughout is to ensure generality of the approach. For example, the size distribution used in the forward model may come from a separate SMPS system (or even a model) that comes with binning that is not necessarily known ahead of time. The way we configured our TDMA is to set the voltage/size in DMA 1 denoted as Dd and then perform an SMPS scan over the range, for example, 0.7 \* Dd to 2.5 \* Dd over 60 s. The flow ratios in DMA 1 and DMA can differ, though we usually keep both 5:1. The bins are constructed as a geometrically stepped mobility grid between the lower and upper range. The only information about DMA is that of the nominal diameter.*

    *There might clever ways to select the bins in DMA 2 to be a subset of DMA 1, but this does not quite obviate the need for interpolation in the forward model, unless one also matches the allowable growth factor to the discrete bin values. Even then, the growth for particles that carry multiple charges the corresponding mobility will not match that of the discrete binning. The paragraph on discretization includes now text explaining where interpolation is necessary, and where it can be avoided.*

[130] Response

[131] Reasonable choices are $i = 120$, $j = n = 30$.

[131] Text

[132]The transitions between mobility and diameter with their different natural binning make it difficult to minimize the problems of unpredictable smoothing by interpolations. It seems little can be done about that unless you are willing to set up a universal scale throughout based on either mobility or mobility diameter. Then bin midpoints could be translated to the other parameter and from there to all other scales. Each scale should either match the universal resolution and e midpoints or use an integer multiple of the resolution (e.g. each midpoint in one scale matches every third midpoint in another scale). If this approach were used, it would seem best to use a universal scale with uniform increments in either ln(dp) or ln(z) as appropriate. I believe that if the universal scale matched that of Pg and with uniform increments in ln(g) or ln(dp), then grown particles from one bin would land exactly into another bin with no interpolation or fractional bin calculation required.

[132] Referee

[133] *Please see response to previous comment and text below. Interpolation is deeply interwoven in into this framework. Since the binning along all dimensions is arbitrary, it is possible to setup a universal (or near universal) grid in which interpolation is minimized, which is*

[133] Response

*now mentioned. The paragraph describing the discretization is revised as follows.*

[revised manuscript text omitted]
*, 40(2):134–151, February 2009. ISSN 0021-8502. DOI: 10.1016/j.jaerosci.2008.07.013.

Jingkun Jiang, Chungman Kim, Xiaoliang Wang, Mark R. Stolzenburg, Stanley L. Kaufman, Chaolong Qi, Gilmore J. Sem, Hiromu Sakurai, Naoya Hama, and Peter H. McMurry. Aerosol Charge Fractions Downstream of Six Bipolar Chargers: Effects of Ion Source, Source Activity, and Flowrate. *Aerosol Science and Technology*, 48(12):1207–1216, December 2014. ISSN 0278-6826. DOI: 10.1080/02786826.2014.976333.

Milind Kandlikar and Gurumurthy Ramachandran. Inverse Methods for Analysing Aerosol Spectrometer Measurements: A Critical Review. *Journal of Aerosol Science*, 30(4):413–437, 1999. ISSN 0021-8502. DOI: 10.1016/S0021-8502(98)00066-4.

E. O. Knutson and K. T. Whitby. Aerosol classification by electric mobility: Apparatus, theory, and applications. *Journal of Aerosol Science*, 6(6):443–451, 1975. ISSN 0021-8502. DOI: 10.1016/0021-8502(75)90060-9.

M. D. Petters. A language to simplify computation of differential mobility analyzer response functions. *Aerosol Science and Technology*, 52(12):1437–1451, December 2018. ISSN 0278-6826. DOI: 10.1080/02786826.2018.1530724.

D.J. Rader and P.H. McMurry. Application of the tandem differential mobility analyzer to studies of droplet growth or evaporation. *Journal of Aerosol Science*, 17(5):771–787, January 1986. ISSN 0021-8502. DOI: 10.1016/0021-8502(86)90031-5.

A. Reineking and J. Porstendörfer. Measurements of Particle Loss Functions in a Differential Mobility Analyzer (TSI, Model 3071) for Different Flow Rates. *Aerosol Science and Technology*, 5(4): 483–486, January 1986. ISSN 0278-6826. DOI: 10.1080/02786828608959112.

Lynn M. Russell, Shou-Hua Zhang, Richard C. Flagan, John H. Seinfeld, Mark R. Stolzenburg, and Robert Caldow. Radially Classified Aerosol Detector for Aircraft-Based Submicron Aerosol Measurements. *Journal of Atmospheric and Oceanic Technology*, 13(3):598–609, June 1996. ISSN 0739-0572. DOI: 10.1175/1520-0426(1996)013<0598:RCADFA>2.0.CO;2.

C. Shen, G. Zhao, and C. Zhao. Effects of multi-charge on aerosol hygroscopicity measurement by a HTDMA. *Atmospheric Measurement Techniques*, 14(2):1293–1301, 2021. DOI: 10.5194/amt-14-1293-2021.

Mark R. Stolzenburg and Peter H. McMurry. Equations Governing Single and Tandem DMA Configurations and a New Lognormal Approximation to the Transfer Function. *Aerosol Science and Technology*, 42(6):421–432, April 2008. ISSN 0278-6826. DOI: 10.1080/02786820802157823.

A. Voutilainen, V. Kolehmainen, and J. P. Kaipio. Statistical inversion of aerosol size measurement data. *Inverse Problems in Engineering*, 9(1):67–94, January 2001. ISSN 1068-2767. DOI: 10.1080/174159701088027753.

Shih Chen Wang and Richard C. Flagan. Scanning Electrical Mobility Spectrometer. *Aerosol Science and Technology*, 13(2):230–240, January 1990. ISSN 0278-6826. DOI: 10.1080/02786829008959441.

A. Wiedensohler. An approximation of the bipolar charge distribution for particles in the submicron size range. *Journal of Aerosol Science*, 19(3):387–389, June 1988. ISSN 0021-8502. DOI: 10.1016/0021-8502(88)90278-9.

Shou-Hua Zhang, Yoshiaki Akutsu, Lynn M. Russell, Richard C. Flagan, and John H. Seinfeld. Radial Differential Mobility Analyzer. *Aerosol Science and Technology*, 23(3):357–372, January 1995. ISSN 0278-6826. DOI: 10.1080/02786829508965320.

---

## Referee Report (RR1)

This paper first lays a brief theoretical framework around regularization, SMPS inversion, and TDMA inversion. The theory surrounding SMPS and TDMA inversion is framed within the developed Julia software environment and not within the traditional DMA/TDMA inversion. However, the author does reference the 2018 paper (first version of this software) which does translate the software framing into the traditional SMPS/TDMA framework. Readers new to this subject will likely need to use the 2018 paper to digest the results as mentioned at the end of the introduction. I was able to do so without issue.

During the theoretical explanation, the author documents the differences between the 2018 edition and this new edition. These changes in TDMA inversion are positive and highlight our current understanding of multicharged behavior. As such, this routine represents a full multi-charge inversion in TDMAs as we currently understand it. Additionally, different regularization techniques are now a part of this software package. These two changes are a marked improvement over existing inversion methods which restrict inversion to a single method or neglect multicharged particles.

The paper then proceeds to test different regularization methods on large data sets, which at this point, is of great value to the community and to me. Comparison of different regularization methods is of great interest, and I suspect I will read the final version several times to digest the results of this study. After reading, I have no major issue with the revision.

I have a few comments and questions as documented below.

Line 26: I understand "mixing state" to be internal, external, or a combination of the two. The mentioned variables do not fully define my understanding of mixing state. Maybe a sentence or two clarifying this statement is needed.

Line 61: I am not crazy about the use of the word "shape." I may misunderstand the inversion, but many of these routines do not assume a "parameterized function", however, they do assume a shape. (i.e., a series of rectangles or a series of trapezoids (lines))

Line 111: Is this suppose to begin a new paragraph???

Line 144: I do not understand where the 8 combinations come from. Is it omission or presence of D and B along with two algorithms?

Line 159: I may be a little confused. Does this mean that only the initial guess is bounded?

Line 169: Is the initial guess also the a-priori estimate? If so, the words "initial guess" seem inappropriate. When I see "initial guess", I assume these are the beginning values for x. However, equation 5 states that $x_0$ is not an "initial guess." Is $x_0$ both an initial guess and the estimate?

Line 276: there appears to be a Zs (or other variable) missing in the sentence.

Equation 11: I do not see how this yields an array. Based on the previous example for mapfoldl, I assume that the output is a sum (the variable a) as you have previously defined. Is the sum (or subtraction) only

an example and replacement of – with vcat changes the output from a progressing sum to a concatenating array? If so, can we change the writing in the example to say that the sum is an example.

Line 295: I think the word "exiting" should be "entering."

Lines 295 through 303: A gentle reminder to the reader regarding the meaning of the dot (·) between growth factor and the product of T and the inlet size distribution. For a long while, I thought you were converting the size distribution into another form. Only after some extended study did I realize that $g_0$ was applied only to diameter.

Line 337: you state that the mobility grid for DMA 2 is $Z_{s,2}$. Do you mean this vector is the array of centroid mobilities for DMA2? Line 315 states that it is such. Is this variable supposed to be Z?

Line 351: is n the number of growth factor bins? This sentence may need rearrangement.

Line 360: Equation 18? Do you mean 17?

Line 380: Our TDMA does not normally geometrically step (although it can). Will that impact the use of this routine?

Line 390: Please choose how to denote the subscript for *D*. Because there is no way to show the superscript for e, confusion can ensue. The example on Line 151 does not use exponential form. It would be good if one form or the other is used throughout. I spent a good amount of time trying to figure out what "e" was. I had a long laugh when I did figure it out.

Line 552: I expected a more direct statement that summarized which inversion method was better for the dataset. From this I assume I should choose $L_2B$. Is this correct?

Line 628: Would prefer using the word function instead of shape as the inversion uses rectangles which is also a shape.

---

## Referee Report (RR2)

**Referee Comments for Manuscript AMT-2021-51 Ver. 3 "Revisiting Matrix-Based Inversion of SMPS and HTDMA Data" Markus D Petters**

This manuscript addresses the important issue of automating the processing of SMPS and tandem DMA data. The idea of inverting data with regularization is sound. However, there is still a functional problem with the development of the matrix-based forward model of calculating system response from a known input distribution. Furthermore, some of the descriptions in that section could be much clearer. If these issues can be properly addressed, the resulting software package should prove of great utility.

**Major Comments**

As before, the main focus of this review is the development of the forward matrix model in the manuscript. The author has generally addressed my previous comments satisfactorily. In particular, the changes to properly account for multiply-charged particles in the second DMA of a tandem DMA setup are greatly appreciated. However, there appears to be a remaining problem in this situation with tracking the true mobility diameter of the particles after growth, $D_{wet}$, to the same diameter in DMA2.

When changing thermodynamic state, $\Lambda$ or $(T,p)$, true mobility diameter (*i.e.* physical diameter) of the particles is the only "size" parameter that remains constant regardless of charge state. For multiply-charged particles, apparent mobility diameter does not equal true mobility diameter and it depends on $(T,p)$. Therefore, for multiply-charged particles transitioning from DMA1 conditions to DMA2 conditions, the relationship between apparent and true mobility diameters changes. If apparent mobility diameter is held constant across the transition, then it is necessarily true that mobility diameter is not. The somewhat garbled explanation of the function of $\Pi_k^{\Lambda,\delta}$ (lines 299-301) seems to indicate that the particle size sent to the convolution matrix, $\mathbf{O}_k$, for transport through DMA2 is the apparent mobility diameter at DMA1 conditions. Though the actual end effect may be small, this is technically incorrect for multiply-charged particles.

In equation form, this may be seen as follows. Let the function $h$ represent the forward calculation of mobility from diameter as $z_1/k = h_1(D_m)$ where the calculation is done at DMA1 conditions. Then the inverse calculation is represented as $D_m = h_1^{-1}(z_1/k)$. Using this, the apparent diameter at DMA1 conditions is given as $D_{a1} = h_1^{-1}(z_1) = h_1^{-1}(k \cdot h_1(D_m))$. If $D_{a2}$ is equated to $D_{a1}$, then the corresponding true mobility diameter in DMA2 is given as $D_{m2} = h_2^{-1}(h_2(D_{a2})/k)$. For $|k|=1$, all these equations simply collapse such that $D_{m2}=D_{a2}=D_{a1}=D_m$. But for $|k|>1$, $D_{m2}\neq D_m$, meaning that the true mobility (physical) diameter is not preserved across the transition. Furthermore, if the output of $\Pi_k^{\Lambda,\delta}$ is apparent mobility diameter at

DMA1 conditions, then in order to correctly compute the diffusion effects associated with DMA2, this conversion to apparent mobility diameter would need to be directly reversed (at DMA1 conditions) just to get back to true mobility diameter, a computational waste. As far as I can see, the apparent mobility diameter is of no use in the context of these TDMA equations.

Eliminating that concept entirely from the development would also eliminate the need for such strange terminology as an "apparent +1 mobility diameter". At least to me, that equates to a "false true mobility diameter". The term "mobility diameter" alone is well-defined in terms of the particle dynamic mobility ($z/ke$ where $e$ is the unsigned electronic charge). The addition of "+1" is then, in fact, somewhat redundant. There is no need to retain the "+1" when referring to an "apparent mobility diameter".

Since I have not even looked at the code for this, I admittedly know little of the details of the programming of these equations for the calculation of the forward model. However, with the implementation of the corrections discussed above, it seems likely that there remains no compelling reason to treat the input parameters of DMA1, $\Omega^{\Lambda,\delta}(Z, z^s/k, k)$ (Eq. 10), and DMA2, $\Omega^{\Lambda,\delta}(Z, z^s, k)$ (Eq. 15), differently in these equations. Assuming that the $\Omega$ algorithm has no hidden switches for different input structures, these variations in input can be quite confusing and seemingly contradictory.

In addition, there is an interpolation step from the input distribution diameter grid to the DMA1 mobility grid and a very similar interpolation step from the grown particle diameter grid, $D_{wet}$, to the DMA2 mobility grid. Each of these interpolations takes as input the incoming diameter grid and the outgoing mobility grid with its associated DMA $\Lambda$ parameter. From this, a relatively sparse matrix is generated to operate on the incoming concentration vector resulting in an outgoing concentration vector interpolated to the new grid. In these equations, the two interpolations could be similarly codified as $\Pi^{\Lambda_1}(Z_{s,1}, D_{in})$ and $\Pi^{\Lambda_2}(Z_{s,2}, d_1^s \cdot G)$ where $D_{in}$ is the input diameter grid, $d_1^s = D_p(z_1^s)$ is the mobility diameter associated with $z_1^s$, $G$ is the vector of discretized $g_0$ values, and $d_1^s \cdot G$ is the grid of grown diameter bins.

Notation such as used above can greatly improve the clarity of these equations with significantly less reliance on explanations in the surrounding text. In general, there is much room to improve the consistency and logical arrangement of the notation. For instance, there is no obvious reason why the $Z$ dependence, which makes the result a vector, is omitted from the expression $T_{size}^{\Lambda,\delta}(k, z^s)$ (Eq. 10) and the subscript *size* seems to add no information at all. The use of a subscript 1 (to indicate singly-charged?) in the expression $D_{p,1}=D_p(z,k=1)$ (line 259) becomes confusing when used in Eq. (15) for DMA2 and especially so in $g=D_{p,1}/D_d$ (line 345) where it represents the diameter associated with an element of $Z_{s,2}$. It would be far better to reserve subscripts 1 and 2 for their most frequent usage in associating the parameter with a

particular DMA.  To distinguish particular values of diameter from the function that generated them, lowercase '$d$'s could be used as above.  And though the superscript $s$ of $z^s$ was apparently originally representative of "star" in "z-star" or $z*$, at this distant removal from that original notation and in this context, it simply looks out of place.  Even the author changed it to a subscript when the $Z_s$ parameter was introduced.  Given the number of flaws uncovered at this point in the formulation of the forward model in Petters (2018), there would seem to be little value in clinging to its confusing notation.  At least in terms of the forward model, the current work should be seen as superseding the previous work.

**Minor Comments and Corrections**

Line 31:  "For  example, …".  No matter how many examples are proffered in the following text, standard usage is "For example".

Lines 87-88:  "Capital bold-roman letters denote matrices (**A**), lower case roman letters denote vectors (x) and lowercase italic symbols denote scalars (*n*)."  However, this list is far from exhaustive.  For instance, uppercase italic $T$s are used as vectors in the later development of the forward model.

Line 88 or line 94:  Include the definition of the pseudo inverse, $\mathbf{A}^+ = (\mathbf{A}^T\mathbf{A})^{-1}\mathbf{A}^T$, as this is perhaps less well known to many.  It also helpful to have that available to compare to the form in Eq. (3).

Lines 100-101:  "Common choices are the first  or second derivative operator defined as the upper bidiagonal(-1; 1) and the  tridiagonal(1;-2; 1) matrix, respectively."  This same reversal is also found in line 145.  There is no such thing as an "upper tridiagonal"; the matrix is symmetrically populated about the main diagonal.

Line 106:  "… the derivative of the right hand side of Eq. (2) with respect to x, …"

Lines 108-109:  "The L-curve method involves a plot of $\log \left\| \mathbf{A}\, \mathrm{x}_\lambda - \mathrm{b} \right\|_2^2$ vs. $\log \left\| \mathbf{L}\left( \mathrm{x}_\lambda - \mathrm{x}_0 \right) \right\|_2^2$."

Note that $\mathrm{x}$ has been changed to $\mathrm{x}_\lambda$ in these expressions.

Line 109:  "The optimal $\lambda$ occurs at the corner of the resulting L-shaped curve, …"

Lines 111 and following:  The description up this point has been very helpful allowing for at least some intuitive grasp of what is going on in these equations and their optimization.  Then the GCV estimator is introduced (Eq. 4) with absolutely no physical context for what it means.  I realize that the full definition for this is somewhat involved and beyond the scope of this paper, but is there any sort of simplified view of what this parameter physically represents?  At least for me, the form of the equation provides no clue at all.  Perhaps that is in part due to my ignorance of the standard form and what that physically represents.

Line 113: $\mathbf{A}^T$ should be $\mathbf{A}^{\mathrm{T}}$ (no italics on the superscript), two places.

Lines150-151: Doesn't $D_{\hat{x}} = diag\left(\left|\hat{x}_1\right|,...,\left|\hat{x}_n\right|\right)$ imply $\left(D_{\hat{x}}\right)_{ii} = \left|\hat{x}_i\right|$? But then that together with $\left(D_{\hat{x}}\right)_{ii} = \varepsilon$ for all $\left|\hat{x}_i\right| < \varepsilon$ creates a contradiction if the set $\left|\hat{x}_i\right| < \varepsilon$ is not empty.

Line 179: "*RegularizationTools.jl* also provides an abstract generic interface …"

Line 196: "… the raw response function defined as the integrated response …"

Line 201: "… requires  evaluating integrals …"

Lines 203-206: "For the forward calculation, the objective is to find a design matrix that maps the growth factor frequency distribution to the raw TDMA response function."

Line 208: "… provide a domain  independent language …" This change would seem to make more sense in this context.

Line 223: "… +1 (singly charged) mobility diameter bin edges …" If you wish to continue to use the somewhat superfluous "+1" notation, it should be explained such as this on first use.

Line 226: "For  example, …"

Line 227: "…$f * \mathrm{n}$ is the uniform scaling of the concentration fields by factor $f$." Here and in the subsequent example of $f \cdot \mathrm{n}$, $f$ is treated as a simple scalar. But in the very next paragraph $f$ and $g$ are used as functions. Rather than tempt confusion, use some other letter such as $a$ for scalar multiplication, *e.g.* $a * \mathrm{n}$.

Line 233: "… and sums the results."

Lines 234 and following: The description of the $\mathrm{foldl}()$ function is clearer than that of the previous splatting function. However, you have stated the generic rule for the first element while giving examples directly from Julia documentation which use a different rule. The generic rule is that for the first element, $a$ is the neutral value. In the Julia implementation of $\mathrm{foldl}$, the first element acts as $a$ for the processing of the second element. For the example foldl($-$; [1; 2; 3]), the neutral value is 0 and the operation is $- (a, x) = a - x$. The generic implementation of this would be $0 - 1 - 2 - 3 = - 6$ while the Julia implementation is $1 - 2 - 3 = - 4$. The way $\mathrm{foldl}$ is used in the development of the forward model would appear to conform to the Julia implementation. So the description of the first element rule should be changed.

Lines 265-266: "… which corresponds to the centroid mobility setting for the DMA to transmit particles  with k charges under the assumption that they carry only a single charge." Even with this change, I find it very difficult to understand what is going on

in this section.  The only way I can make sense of this is that the $Z$ vector at this point contains values of $z/k$, not simply $z$.  If this is the case, it needs to be stated as so somewhere.  Also, if this is the case, then I believe the DMA transfer function dependence in Eq. (10) given as $\Omega\,(Z,\,z^{s}/k,\,k)$ should actually be $\Omega\,(Z,\,z^{s}/k,\,1)$, assuming $\Omega\,(z,\,z^{s},\,k)$ is the "normal" arrangement of inputs.

Line 268-270:  "The function $T_{size}^{\Lambda,\delta}(1,z^{s})$ evaluates to a vector of the same length as $Z$. Performing an elementwise sum over all $T_{size}^{\Lambda,\delta}(k,z^{s})$ produces the net mobility distribution transmitted by the DMA."  Perhaps it is clear enough from context, but noting that the summation is over all $k$ could be helpful.  More importantly, the result of the summation is only the net mobility distribution if $\sum_{k}T_{size}^{\Lambda,\delta}(k,z^{s})$ is multiplied by the input distribution, $\mathrm{n}$. Otherwise, the bare summation is simply the net transmission probability function.

Line 270:  "Examples for …  are shown in Figure 2, …"

Lines 276-277:  "… and $Z_s$ is a vector of centroid mobilities scanned by the DMA."  The vector $Z_s$ is missing from this description.

Line 277:  "The matrix is square if $Z_s = Z$ in Eq. 10."  I think you mean Eq. 11.  As noted earlier, this would be a lot clearer if the explicit $Z$ dependence were actually shown in $T_{size}^{\Lambda,\delta}(k,z^{s})$ as $T_{size}^{\Lambda,\delta}(Z,z^{s},k)$.

Lines 289-290:  "The latter is the raw response function defined as integrated response downstream of the DMA …"  The latter refers to $\mathrm{r}$ in Eq. 12.  $\mathrm{r}$ is similar to a size distribution, it is NOT integrated over all sizes as would be the case for the response of a CPC detector.

Line 295:  "The mobility distribution  entering DMA 2 …"

Line 297:  "… Eq. (13) … evaluates to … that exit the humidity conditioner  …"

Lines 298-299:  "Subscripts are used to differentiate DMA 1 and 2 which possibly have different geometries, flow rates, and grids, …"  This list should include thermodynamic states, that is, different temperature and/or pressure.  DMA2 is definitely at a lower pressure than DMA1 and this is reflected in the relationship between mobility and mobility diameter.

Line 311:  "The total humidified mobility distribution $\mathrm{m}_{t}^{\delta_{2}}$ exiting DMA 2 is given by (Eq. 14) …"  The total humified mobility distribution exiting DMA2 would be that observed at fixed $z_{s,2}$ without integrating over particle size.  $\mathrm{m}_{t}^{\delta_{2}}$ is the total integrated concentration versus $z_{s,2}$. It is not a true particle distribution.

Line 322:  "… the humidified distribution function exiting DMA 2 …"  Same as previous comment.  This might be considered to be a pseudo-distribution if what is meant by that is explained up front.

Line 324: In reference to Eq. 16, "…the diameters in $\mathbf{M}_k^{\delta_1}$ are normalized by $D_{dry}$." This does not seem correct. The diameters, or at least the associated mobilities, cannot be altered until after passage through DMA2. I can't figure out where in Eq. 16 you could safely normalize diameters.

Line 327: "… the DMA  setups $\Lambda_1$, $\Lambda_2$, $\delta_1$, $\delta_2$ …"

Lines 332-333: "Transmission through DMA 1 is computed for a specified $z^s$ (the dry mobility) and $g_0$ (the growth factor) via Eq. (13)." The humidity conditioner as represented by $g_0$ is NOT part of DMA1.

Line 339: "… the matrix is non square." This is only true if $j \neq i$. Perhaps $j$ is typically much smaller than $i$, but you haven't said anything about that.

Line 342: "The advantage of interpolation is that the  matrices $\mathbf{O}_k$ are smaller." Here again it is implied that $j$ is significantly smaller than $i$ but you have not explained why that is.

Line 347: "… the humidified mobility pseudo-distribution function …" or some such modification.

Lines 350-351: "If the grids for $P_g$ and that of DMA 2 do not align, interpolation is used to map the $P_g$ grid onto the DMA 2 grid." The $P_g$ grid would refer to the discretization of $g_0$, or $G$, using the notation from above. What is being interpolated onto the DMA2 grid is $d_1^s \cdot G$.

Lines 351-352: "The choice of $i$, $j$, $n$, the ranges of mobility grids for DMA 1, DMA 2, and the range of $P_g$, is only constrained …" Insert comma after $P_g$. The range of $P_g$ is simply [0,1]; it is the range of $g_0$ that is needed here or perhaps "the range of the growth grid for $P_g$".

Line 359: A new paragraph should start with "Figure 1 …".

Line 390: Concerning bounds for $P_g$, if $P_g = dF/dg$ from Figure 1 caption and as would be appropriate in Eq.16, then there is no upper bound for Pg. For if one introduced a perfectly monodisperse as the input aerosol, $P_g$ would range from zero to infinity. However, in discretizing the integral of Eq. 16 it is possible to assign the bin width factor, $\Delta g$, to the $dF/dg$ vector to get a $\Delta P_g$ vector. This would be properly bounded by [0,1]. This should be explained more clearly or the notation in the equations appropriately modified to show this.

Lines 396-397: "Visual evaluation …  suggests …"

The balance of the paper was not reviewed.

---

## Author Response (AR2)

**Referee Comments: Christopher Oxford**

This paper first lays a brief theoretical framework around regularization, SMPS inversion, and TDMA inversion. The theory surrounding SMPS and TDMA inversion is framed within the developed Julia software environment and not within the traditional DMA/TDMA inversion. However, the author does reference the 2018 paper (first version of this software) which does translate the software framing into the traditional SMPS/TDMA framework. Readers new to this subject will likely need to use the 2018 paper to digest the results as mentioned at the end of the introduction. I was able to do so without issue.

During the theoretical explanation, the author documents the differences between the 2018 edition and this new edition. These changes in TDMA inversion are positive and highlight our current understanding of multicharged behavior. As such, this routine represents a full multi-charge inversion in TDMAs as we currently understand it. Additionally, different regularization techniques are now a part of this software package. These two changes are a marked improvement over existing inversion methods which restrict inversion to a single method or neglect multicharged particles.

The paper then proceeds to test different regularization methods on large data sets, which at this point, is of great value to the community and to me. Comparison of different regularization methods is of great interest, and I suspect I will read the final version several times to digest the results of this study. After reading, I have no major issue with the revision. I have a few comments and questions as documented below.

Line 26: I understand "mixing state" to be internal, external, or a combination of the two. The mentioned variables do not fully define my understanding of mixing state. Maybe a sentence or two clarifying this statement is needed.

The paper is revised to be in line with the formal definition by Riemer et al.: "Aerosol mixing state is defined as the distribution of properties across a population of particles within an aerosol."

To predict the impact of aerosol on the Earth system, the distributions of particle size, chemical composition, hygroscopicity, and morphology must be known. The distribution of these properties across a population of particles formally defines the mixing state of the aerosol (Riemer et al., 2019). Accurate measurements of the mixing state are critical for formulating models that link aerosol, cloud, and climate properties.

Line 61: I am not crazy about the use of the word "shape." I may misunderstand the inversion, but many of these routines do not assume a "parameterized function", however, they do assume a shape. (i.e., a series of rectangles or a series of trapezoids (lines))

The paper has been changed to read "the functional form of the mobility growth factor frequency distribution".

Line 111: Is this supposed to begin a new paragraph???

Fixed.

Line 144: I do not understand where the 8 combinations come from. Is it omission or presence of D and B along with two algorithms?

Revised as follows.

There are eight combinations by which to compose methods via Eq. (5), L, $Lx_0$, $Lx_0B$, $Lx_0D$, $Lx_0BD$, LB, LBD, and LD. Combined with the three most common filter matrices $L_0 = \mathbf{I}$, $\mathbf{L}_1 =$ upper bidiagonal($-1$, 1) and, $\mathbf{L}_2 =$ upper tridiagonal(1, $-2$, 1) this results in 24 unique methods.

Line 159: I may be a little confused. Does this mean that only the initial guess is bounded?

The solution is bounded. If the initial guess produces results that are outside the bounds, passing this initial guess to the solver will result in an error. Hence the truncation. To be clear the text is revised as.

The net result is an optimized solution that is within the specified upper and lower bounds. The upper and lower bounds are vectors of the same size as x.

Line 169: Is the initial guess also the a-priori estimate? If so, the words "initial guess" seem inappropriate. When I see "initial guess", I assume these are the beginning values for x. However, equation 5 states that x0 is not an "initial guess." Is x0 both an initial guess and the estimate?

The text is now changed to read *a-priori* estimate.

Line 276: there appears to be a Zs (or other variable) missing in the sentence.

Fixed

Equation 11: I do not see how this yields an array. Based on the previous example for mapfoldl, I assume that the output is a sum (the variable a) as you have previously defined. Is the sum (or subtraction) only an example and replacement of – with vcat changes the output from a progressing sum to a concatenating array? If so, can we change the writing in the example to say that the sum is an example.

Yes, mapfoldl will just apply the given function to reduce the mapped collection. If the function is +, it is a sum. If it is vcat, it concatenates arrays.

However, other programming languages may concatenate along a different dimension as the definition of horizontal and vertical is arbitrary. Passing vcat to foldl (or mapfoldl) will result in a concatenated array.

Line 295: I think the word "exiting" should be "entering."

Thank you. Revised.

Lines 295 through 303: A gentle reminder to the reader regarding the meaning of the dot ($\cdot$) between growth factor and the product of T and the inlet size distribution. For a long while, I thought you were converting the size distribution into another form. Only after some extended study did I realize that g0 was applied only to diameter.

Caveat added a little but further below (line 305).

The size distribution is grown by the growth factor $g_0$, which is achieved by applying the $\cdot$ operator to ....

Line 337: you state that the mobility grid for DMA 2 is Z s,2 . Do you mean this vector is the array of centroid mobilities for DMA2? Line 315 states that it is such. Is this variable supposed to be Z?

The following has been added to line 315 to clarify.

Note that the choice of $Z$ inside $\Omega$ is up to the user. Sensible choices are $Z = Z_{s,1}$ or $Z = Z_{s,2}$ the implications of which are further discussed later.

Line 351: is $n$ the number of growth factor bins? This sentence may need rearrangement.

Yes. It is stated a few lines above. "Thus the growth factor probability distribution $P_g$ in Eq. (17) can be discretized into $n$ arbitrary growth factor bins."

Line 360: Equation 18? Do you mean 17?

The black line is from the Matrix form, i.e. Eq. (18). The colored lines are from Eq. (15) as stated in the caption.

Line 380: Our TDMA does not normally geometrically step (although it can). Will that impact the use of this routine?

It will work with any binning. Perhaps the sentence is too myopic because our HTDMA operates like that.

This is due to the evaluation of the humidified size distribution along a geometrically stepped mobility grid, which is typical in scanning DMA setups.

Line 390: Please choose how to denote the subscript for D. Because there is no way to show the superscript for e, confusion can ensue. The example on Line 151 does not use exponential form. It would be good if one form or the other is used throughout. I spent a good amount of time trying to figure out what "e" was. I had a long laugh when I did figure it out.

That is a good point. I fixed the notation for equation 6 and defined the use there.

The method $L_1 D_{1e\text{-}2}$ represents a filter matrix with a first-order derivative operator applied to Eq. (6) with $\epsilon = $ 1e-2. Exponential notation is used because subscripts are difficult to superscript.

Line 552: I expected a more direct statement that summarized which inversion method was better for the dataset. From this I assume I should choose $L_2 B$. Is this correct?

Perhaps. The answer for this dataset is yes. However, I don't want to overgeneralize. I could imagine cases where $L_2 B$ results in over-smoothing when two modes are close together.

Line 628: Would prefer using the word function instead of shape as the inversion uses rectangles which is also a shape.

Changed to

… functional form of the growth factor frequency distribution

**Referee Comments: Mark Stolzenburg**

Referee Comments for Manuscript AMT-2021-51 Ver. 3 "Revisiting Matrix-Based Inversion of SMPS and HTDMA Data" Markus D Petters

This manuscript addresses the important issue of automating the processing of SMPS and tandem DMA data. The idea of inverting data with regularization is sound. However, there is still a functional problem with the development of the matrix-based forward model of calculating system response from a known input distribution. Furthermore, some of the descriptions in that section could be much clearer. If these issues can be properly addressed, the resulting software package should prove of great utility.

**Major Comments**

As before, the main focus of this review is the development of the forward matrix model in the manuscript. The author has generally addressed my previous comments satisfactorily. In particular, the changes to properly account for multiply-charged particles in the second DMA of a tandem DMA setup are greatly appreciated. However, there appears to be a remaining problem in this situation with tracking the true mobility diameter of the particles after growth, $D_{\text{wet}}$, to the same diameter in DMA2.

When changing thermodynamic state, $\Lambda$ or $(T,p)$, true mobility diameter (*i.e.* physical diameter) of the particles is the only "size" parameter that remains constant regardless of charge state. For multiply-charged particles, apparent mobility diameter does not equal true mobility diameter and it depends on $(T,p)$. Therefore, for multiply-charged particles transitioning from DMA1 conditions to DMA2 conditions, the relationship between apparent and true mobility diameters changes. If apparent mobility diameter is held constant across the transition, then it is necessarily true that mobility diameter is not.

The somewhat garbled explanation of the function of $\Pi_k^{\Lambda, \delta}$ (lines 299-301) seems to indicate that the particle size sent to the convolution matrix, $\mathbf{O}_k$, for transport through DMA2 is the apparent mobility diameter at DMA1 conditions. Though the actual end effect may be small, this is technically incorrect for multiply-charged particles.

In equation form, this may be seen as follows. Let the function $h$ represent the forward calculation of mobility from diameter as $z_1/k = h_1(D_m)$ where the calculation is done at DMA1 conditions. Then the inverse calculation is represented as $D_m = h_1^{-1}(z_1/k)$ . Using this, the apparent diameter at DMA1 conditions is given as $D_{a1} = h_1^{-1}(z_1) = h_1^{-1}(k \cdot h_1(D_m))$ . If $D_{a2}$ is equated to $D_{a1}$, then the corresponding true mobility diameter in DMA2 is given as $D_{m2} = h_2^{-1}(h_2(D_{a2})/k)$. For $|k|=1$, all these equations simply collapse such that $D_{m2} = D_{a2} = D_{a1} = D_m$. But for $|k|>1$, $D_{m2} \neq D_m$ , meaning that the true mobility (physical) diameter is not preserved across the transition. Furthermore, if the output of $\Pi_k^{\Lambda, \delta}$, is apparent mobility diameter at DMA1 conditions, then in order to correctly compute the diffusion effects associated with DMA2, this conversion to apparent mobility diameter would need to be directly reversed (at DMA1

conditions) just to get back to true mobility diameter, a computational waste. As far as I can see, the apparent mobility diameter is of no use in the context of these TDMA equations.

Eliminating that concept entirely from the development would also eliminate the need for such strange terminology as an "apparent +1 mobility diameter". At least to me, that equates to a "false true mobility diameter". The term "mobility diameter" alone is well-defined in terms of the particle dynamic mobility ($z/ke$ where $e$ is the unsigned electronic charge). The addition of "+1" is then, in fact, somewhat redundant. There is no need to retain the "+1" when referring to an "apparent mobility diameter".

Since I have not even looked at the code for this, I admittedly know little of the details of the programming of these equations for the calculation of the forward model. However, with the implementation of the corrections discussed above, it seems likely that there remains no compelling reason to treat the input parameters of DMA1, $\Omega^{\delta,\Lambda}(Z, z^s/k, k)$ (Eq. 10), and DMA2, $\Omega^{\delta,\Lambda}(Z, z^s, k)$ (Eq. 15), differently in these equations. Assuming that the algorithm has no hidden switches for different input structures, these variations in input can be quite confusing and seemingly contradictory.

In addition, there is an interpolation step from the input distribution diameter grid to the DMA1 mobility grid and a very similar interpolation step from the grown particle diameter grid, D wet , to the DMA2 mobility grid. Each of these interpolations takes as input the incoming diameter grid and the outgoing mobility grid with its associated DMA $\Lambda$ parameter. From this, a relatively sparse matrix is generated to operate on the incoming concentration vector resulting in an outgoing concentration vector interpolated to the new grid. In these equations, the two interpolations could be similarly codified as $\Pi^{\Lambda 1}(Z_{s,1}, D_{in})$ and $\Pi^{\Lambda 2}(Z_{s,2}, d_1^s \cdot G)$ where D in is the input diameter grid, $d_1^s = D_p(z_1^s)$ is the mobility diameter associated with $z_1^s$ , G is the vector of discretized $g_0$ values, and $d_1^s$, G is the grid of grown diameter bins.

Notation such as used above can greatly improve the clarity of these equations with significantly less reliance on explanations in the surrounding text. In general, there is much room to improve the consistency and logical arrangement of the notation. For instance, there is no obvious reason why the Z dependence, which makes the result a vector, is omitted from $T_{size}^{\Lambda,\delta}(k, z^s)$ (Eq. 10) and the subscript size seems to add no information at all. The use of a subscript 1 (to indicate singly-charged?) in the expression $D_{p,1} = D_p(z,k=1)$ (line 259) becomes confusing when used in Eq. (15) for DMA2 and especially so in $g = D_{p,1}/D_d$ (line 345) where it represents the diameter associated with an element of $Z_{s,2}$. It would be far better to reserve subscripts 1 and 2 for their most frequent usage in associating the parameter with a particular DMA. To distinguish particular values of diameter from the function that generated them, lowercase 'd 's could be used as above. And though the superscript s of $z^s$ was apparently originally representative of "star" in "z-star" or z*, at this distant removal from that original notation and in this context, it simply looks out of place. Even the author changed it to a subscript when the $Z_s$ parameter was introduced. Given the number of flaws uncovered at this point in the formulation of the forward model in Petters (2018), there would seem to be little value in clinging to its confusing notation.

At least in terms of the forward model, the current work should be seen as superseding the previous work.

Overall response: The manuscript was revised to eliminate the need for $\Pi_k^{\Lambda, \delta}$. The transfer function $\Omega^{\delta, \Lambda}(Z, z^s/k, k)$ in (Eq. 10) and (Eq. 15) are now treated identically. This simplifies the notation and also addresses the concern about using the correct thermodynamic state for DMA 2 in the forward model. **Note that these changes did not affect the results in any meaningful way.**

In response to the comment to eliminate "apparent + 1 mobility diameter". To me, the concept of "apparent + 1 mobility diameter" and "apparent growth factor" are useful and I wish to retain them. I added a section upfront that defines these. This section also explains the utility of the concept as I see it.

The DMA selects particles by electrical mobility. The relationship between mobility and mobility diameter is well known and well defined. The relationship is given, for example, in Eq. (2) in Petters (2018). This work also makes use of the "apparent +1 mobility diameter". It is defined as the conversion from mobility to diameter assuming singly charged particles using the mobility grid scanned by either DMA 1 or DMA 2. The apparent +1 mobility diameter represents the natural diameter axis of a DMA response function, i.e. a plot of the raw detector response versus the nominal DMA setpoint diameter. It is an equivalent measure of mobility. The apparent +1 mobility diameter is ambiguous. Larger particles carrying more than one charge may have the same apparent +1 mobility diameter as smaller particles carrying fewer charges. The "apparent growth factor" is defined as the apparent +1 mobility diameter in scanned by DMA 2 divided by the nominal selected dry diameter in DMA.

In response to eliminating the $\Pi_k^{\Lambda, \delta}$ notation/concept: I followed the referee's suggestion but I do want to give a brief explanation. The $\Pi_k^{\Lambda, \delta}$ function was introduced to map from mobility diameter to apparent + 1 mobility diameter BEFORE entering DMA 2. The origin for this was a simple back fold for visualization between mobility and apparent +1 mobility as shown/calculated in Petters (2018). This is strictly correct when no size changes occur between DMA 1 and DMA 2. However, the mapping between mobility and apparent +1 mobility is non linear in the humidified tandem DMA. For this reason $\Pi_k^{\Lambda 1, \delta 1}$ was introduced. It was applied before entering DMA 2 to maintain conceptual symmetry to the prior work. As pointed out by the referee, the pressure in DMA 2 may be lower. In practice, a few 10s Pa of pressure drop have a negligible effect on $\Pi_k^{\Lambda 1, \delta 1}$. (In fact even 200 hPa is hardly noticeable). It is possible to apply $\Pi_k^{\Lambda 2, \delta 2}$ after passage through the DMA to address this concern. However, when doing this there is no utility for visualization. Furthermore, the $\Pi_k^{\Lambda, \delta}$ function can be eliminated while also addressing the referee's concern about the non-symmetrical treatment of the DMA transfer function in the **A** and **O** matrices. For this reason, the $\Pi_k^{\Lambda, \delta}$ function was removed from the treatment. However, there is one **BIG** disadvantage incurred by doing so. The size grid in DMA must be extended to much larger sizes to capture the humidified multiply charged particles. This results in extra computational cost. Furthermore, the revised figures will have a slightly different

binning due to the doubling of the range. The decision for removal of the Π function was motivated by the desire for conceptual clarity and precision over computational speed.

Changes

1. Removal of the PI function and associated text. The key change is that PI is removed and Omega in Eq. (15) is as in Eq. (10). The resulting Figure 1 is essentially unchanged (except for the change in the binning scheme explained below).

Please see changed text after Eq. (13) up to Eq. (16)

2. Bin numbers in all figures were doubled and the range was extended (resulting in approximately the same bin width as before. The reason for this is explained in the text. Due to the new calculation the root mean square error values changed slightly and are updated accordingly.

.. 120 bins between $0.8 < g < 5$. Note that the size grid (or apparent growth factor grid) must be extended to large sizes to capture the growth of multiply-charged particles computed via Eq. (13).

In response to the remaining comments about notation: I appreciate the input and suggestions by the referee. I have implemented most of the referee's suggestions in this and prior revisions. It is my opinion that the added clarifications are sufficient to make the current representation clear enough to follow. The current notation remains close to the implementation in the code and I prefer to keep it that way. Changing the formulation of T, renaming diameters, or introducing alternate constructs to express the interpolation steps will muddy the waters more than clearing it up. Specifically, the software package does retain full backward compatibility with Petters (2018), while also enabling the more precise treatment that is described here. This link will become harder to maintain, the more changes are introduced, and ultimately result in more confusion.

While one aspect of Petters (2018) contained an approximation, i.e. the mapping mobility and apparent +1 mobility diameter for humidified particles in the tandem DMA, it does not mean that it is flawed or superseded in general. The approximation was not that terrible, though it was an unnecessary oversight on my part. However, the remaining points discussed in Petters (2018) fully stand. For example, the excellent agreement between the TSI inversion and inversion using the **A** matrix, the method to compute moments aereas from size selected aerosol using the $T$ function, the application to fit of size-resolved CCN activation spectra while accounting for multiply charged particles, or the application to predict the output size distributions of the dual tandem DMA method. The **A** matrix may be formulated differently in this work, but it is still the **A** matrix. The larger point, a composable computational notation that can express transfer algebraically is also still true. One outcome of this approach is that multiple formulations exist to express the same thing, which is a feature of a language.

Almost all of the excellent suggestions made by the referee in the "Minor Comments and Corrections" were accepted as proposed. Exceptions are explained.

**Minor Comments and Corrections**

Line 31: "For  example, …". No matter how many examples are proffered in the following text, standard usage is "For example".

Corrected.

Lines 87-88: "Capital bold-roman letters denote matrices (**A**), lower case roman letters denote vectors (x) and lowercase italic symbols denote scalars ($n$)." However, this list is far from exhaustive. For instance, uppercase italic $T$s are used as vectors in the later development of the forward model.

Revised as follows.

Section 2.1 and 2.2 uses the following linear algebra notation. Capital bold-roman letters denote matrices (**A**) , lowercase roman letters denote vectors (x) and lowercase italic symbols denote scalars ($n$). $\mathbf{A}^\mathbf{T}$ denotes the matrix transpose, and $\mathbf{A}^+ = (\mathbf{A}^\mathbf{T}\mathbf{A})^{-1}\mathbf{A}^\mathbf{T}$ is the matrix pseudo-inverse. Section 2.3 uses additional notation described there.

Line 88 or line 94: Include the definition of the pseudo inverse, $\mathbf{A}^+ = (\mathbf{A}^\mathbf{T}\mathbf{A})^{-1}\mathbf{A}^\mathbf{T}$ , as this is perhaps less well known to many. It also helpful to have that available to compare to the form in Eq. (3).

Added.

Lines 100-101: "Common choices are the first and or second derivative operator defined as the upper bidiagonal(-1; 1) and the  tridiagonal(1;-2; 1) matrix, respectively." This same reversal is also found in line 145. There is no such thing as an "upper tridiagonal"; the matrix is symmetrically populated about the main diagonal.

Upper has been inserted for bidiagonal. The tridiagonal is defined as the upper tridiagonal matrix as in Huckle and Sedlacek (2012), which is also accordance with the definition given by others, e.g. Eq. 7 in Stout and Kalivas (2006), J. Chemometrics 2006; 20: 22–33, DOI: 10.1002/cem.975.

Line 106: "… the derivative of the right hand side of Eq. (2) with respect to x, …"

Corrected.

Lines 108-109: "The L-curve method involves a plot of $\log\left\|Ax_\lambda - b\right\|_2^2$ vs. $\log\left\|Lx_\lambda - x_0\right\|_2^2$. "Note that x has been changed to $x_\lambda$ in these expressions."

Corrected.

Line 109: "The optimal  occurs at the corner of the resulting L-shaped curve, …"

Corrected.

Lines 111 and following: The description up this point has been very helpful allowing for at least some intuitive grasp of what is going on in these equations and their optimization. Then the GCV estimator is introduced (Eq. 4) with absolutely no physical context for what it means. I realize that the full definition for this is somewhat involved and beyond the scope of this paper, but is there any sort of simplified view of what this parameter physically represents? At least for me, the form of the equation provides no clue at all. Perhaps that is in part due to my ignorance of the standard form and what that physically represents.

The red colored segments have been inserted.

The optimal regularization parameter can be obtained using a variety of techniques, including the L-curve method (Hansen 2000) and generalized cross-validation (GCV, Golub et al., 1979). Both methods use metrics that penalize solutions with large variance (amplified noise) or large bias.

The L-curve method involves …

The generalized cross-validation estimator presents a mathematical shortcut to compute the leave-one out cross-validation estimate, which removes one point from the data, creates a model, computes the error between the model and data point not included in the data, and then averages the result over all permutations. It is given by ...

Line 113: $\mathbf{A}^{\mathrm{T}}$ should be $\mathbf{A}^{\mathrm{T}}$ (no italics on the superscript), two places.

Corrected.

Lines 150-151: Doesn't $\mathbf{D}_{\widehat{\mathbf{x}}}^{-1} = diag\left(\left|\widehat{x}_1\right|, \ldots, \left|\widehat{x}_n\right|\right)$ imply $\left(\mathbf{D}_{\widehat{\mathbf{x}}}\right)_{ii} = \left|\widehat{x}_i\right|$? But then that together with $\left(\mathbf{D}_{\widehat{\mathbf{x}}}\right)_{ii} = \varepsilon$ for all $\left|\widehat{x}_i\right| < \varepsilon$ creates a contradiction if the set $\left|\widehat{x}_i\right| < \varepsilon$ is not empty.

Apologies to the poor typesetting, above which is due to limited support of mathtype in google docs. The sentence has been revised :

Elements that satisfy $|\hat{x}_i| < \varepsilon$ are set equal to $\varepsilon$, where $0 < \varepsilon \ll 1$.

Line 179: "RegularizationTools.jl also provides an abstract generic interface …"

Corrected.

Line 196: "… the raw response function defined as the integrated response …"

Corrected.

Line 201: "… requires  evaluating integrals …"

Corrected.

Lines 203-206: "For the forward calculation, the objective is to find a design matrix that maps the growth factor frequency distribution to the raw TDMA response function. "

Corrected.

Line 208: "… provide a domain  independent language …" This change would seem to make more sense in this context.

Domain specific language is used as a language that is specific to a domain, in this case to express the transformations occurring in the domain of differential mobility analyzer systems. No change was made.

Line 223: "… +1 (singly charged) mobility diameter bin edges …" If you wish to continue to use the somewhat superfluous "+1" notation, it should be explained such as this on first use.

Done.

Line 226: "For  example, …"

Corrected.

Line 227: "…$f * \mathtt{m}$ is the uniform scaling of the concentration fields by factor $f$." Here and in the subsequent example of $f \cdot \mathtt{m}$, $f$ is treated as a simple scalar. But in the very next paragraph $f$ and $g$ are used as functions. Rather than tempt confusion, use some other letter such as a for scalar multiplication, e.g. $a * \mathtt{m}$.

Corrected.

Line 233: "… and sums the results."

Corrected.

Lines 234 and following: The description of the foldl() function is clearer than that of the previous splatting function. However, you have stated the generic rule for the first element while giving examples directly from Julia documentation which use a different rule. The generic rule is that for the first element, a is the neutral value. In the Julia implementation of foldl, the first element acts as a for the processing of the second element. For the example foldl($-$; [1; 2; 3]), the neutral value is 0 and the operation is $-(a, x) = a - x$. The generic implementation of this would be $0 - 1 - 2 - 3 = -6$ while the Julia implementation is $1 - 2 - 3 = -4$. The way foldl is used in the development of the forward model would appear to conform to the Julia implementation. So the description of the first element rule should be changed.

The implementation is not Julia specific. Folds without specifying an initial value are used in other languages as well, e.g. Haskell's foldl1 which behaves identical to Julia foldl without initial value. The "neutral value" means that $-(a,x) = x$. Perhaps the identity element is a better description. In any case, the change below makes it explicit and has been added to the manuscript.

If no initial value is provided, as is the case in this manuscript, foldl applies the function to the first two elements of the list to compute the first $a$.

Lines 265-266: "… which corresponds to the centroid mobility setting for the DMA to transmit particles  with k charges under the assumption that they carry only a single charge."

Corrected.

Even with this change, I find it very difficult to understand what is going on in this section. The only way I can make sense of this is that the $Z$ vector at this point contains values of $z/k$, not simply $z$. If this is the case, it needs to be stated as so somewhere. Also, if this is the case, then I

believe the DMA transfer function dependence in Eq. (10) given as $\Omega$ ($Z$, $z^s$/$k$, $k$) should actually be $\Omega(Z, z^s /k, 1)$, assuming $\Omega(Z, z^s, k)$ is the "normal" arrangement of inputs.

This is not the case. It is just a mathematical trick to coax the generic function $\Omega(z, z^s, k)$ to map in a manner such that $D_{p,1}$ becomes equal to mobility diameter. It is what enables the use of $D_{p,1}$ in the subsequent terms. Below is a slightly revised explanation.

The functional $\Omega$ depends on three arguments $\Omega(Z, z^*, k)$ and implicitly on the DMA configuration $\Lambda$ (i.e., Eq. 13 in Stolzenburg and McMurry 2008). The output is a vector along the mobility grid $Z$. The maximum transmission occurs at $Z/z^* = 1$. The last argument denotes the number of charges carried by the particle. It is used to compute the mobility diameter from $z^*$ and in turn the diffusion coefficient which is required to account for diffusional broadening of the transfer function. The output of $T_{size}^{\Lambda,\delta}\left(k, z^s\right)$ is the transmission of particles through the DMA in terms of the true particle mobility diameter. This is achieved by passing $z^s/k$ as argument to $\Omega$, which corresponds to the centroid mobility setting for the DMA to transmit particles with $k$ charges under the assumption that they carry only a single charge. The net result is that $D_{p,1} = D_p(z,k=1)$, where $z$ is an element of $Z$, becomes equal to the true mobility diameter axis. As a consequence the charge fraction $T_c(k, D_{p,1})$ and penetration efficiency $T_l(D_{p,1})$ are evaluated at the correct diameter.

Line 268-270: "The function $T_{size}^{\Lambda,\delta}\left(1, z^s\right)$ evaluates to a vector of the same length as $Z$. Performing and elementwise sum over $T_{size}^{\Lambda,\delta}\left(k, z^s\right)$ produces the net mobility distribution transmitted by the DMA." Perhaps it is clear enough from context, but noting that the summation is over all k could be helpful. More importantly, the result of the summation is only the net mobility distribution if $\sum\limits_k T_{size}^{\Lambda,\delta}\left(k, z^s\right)$ is multiplied by the input distribution, $\mathbb{m}$. Otherwise, the bare summation is simply the net transmission probability function.

Revised as follows.

Performing an elementwise sum over all $T_{size}^{\Lambda,\delta}\left(k, z^s\right)$ (where the sum is over all charges $k$) produces the net transmission probability function. Multiplication of the transmission probability function with the input distribution results in the mobility distribution transmitted by the DMA. Examples for $T_{size}^{\Lambda,\delta}\left(1, z^s\right) * \mathbb{m}$, $T_{size}^{\Lambda,\delta}\left(2, z^s\right) * \mathbb{m}$, and $T_{size}^{\Lambda,\delta}\left(3, z^s\right) * \mathbb{m}$ are shown in Figure 2, right panel in Petters (2018).

Line 270: "Examples for …  are shown in Figure 2, …"

Corrected.

Lines 276-277: "… and $Z_s$ is a vector of centroid mobilities scanned by the DMA." The vector $Z_s$ is missing from this description.

Corrected.

Line 277: "The matrix is square if $Z_s = Z$ in Eq. 10." I think you mean Eq. 11.

Corrected.

As noted earlier, as this would be a lot clearer if the explicit Z dependence were actually shown in $T_{size}^{\Lambda,\delta}\left(k,\ z^{s}\right)$ as $T_{size}^{\Lambda,\delta}\left(Z,\ z^{s},k\right)$.

As discussed above I prefer to keep it the way it is. Once reason is that $T_{size}^{\Lambda,\delta}\left(k,\ z^{s}\right)$ function generally queries the $Z$ grid defined by the DMA (it doesn't have to). Thus, in code the Z vector becomes δ.Z and Dp,1 vector δ.Dp. Revising it as proposed would confuse the issue.

Lines 289-290: "The latter is the raw response function defined as integrated response downstream of the DMA …" The latter refers to ℝ in Eq. 12. ℝ is similar to a size distribution, it is NOT integrated over all sizes as would be the case for the response of a CPC detector.

Reworded as follows.

The latter is the raw response function, where each element corresponds to the integrated response downstream of DMA 1 for a set upstream voltage (or corresponding $z^s$ or apparent +1 mobility diameter but not true physical diameter for multiply charged particles).

Line 295: "The mobility distribution  entering DMA 2 …"

Corrected.

Line 297: "… Eq. (13) … evaluates to … that exit the humidity conditioner  …"

Corrected.

Lines 298-299: "Subscripts are used to differentiate DMA 1 and 2 which possibly have different geometries, flow rates, and grids, …" This list should include thermodynamic states, that is, different temperature and/or pressure. DMA2 is definitely at a lower pressure than DMA1 and this is reflected in the relationship between mobility and mobility diameter.

Added as suggested.

..,which possibly have different geometries, flow rates, thermodynamic state, and mobility grids, ...

Line 311: "The total humidified mobility distribution $m_t^{\delta 2}$ exiting DMA 2 is given by (Eq. 14) …" The total humified mobility distribution exiting DMA2 would be that observed at fixed $z_{s,2}$ without integrating over particle size. $m_t^{\delta 2}$ is the total integrated concentration versus $z_{s,2}$. It is not a true particle distribution.

The "apparent +1 caveat was stated directly after the equation. It is now moved upfront.

The total humidified apparent +1 mobility diameter distribution $m_t^{\delta 2}$ exiting DMA 2 is given by:

Line 322: "… the humidified distribution function exiting DMA 2 …" Same as previous comment. This might be considered to be a pseudo-distribution if what is meant by that is explained up front.

If the aerosol is externally mixed, the humidified apparent +1 mobility diameter distribution function exiting DMA 2

Line 324: In reference to Eq. 16, "…the diameters in $\mathbb{M}_t^{\delta 1}$ are normalized by $D_{dry}$." This does not seem correct. The diameters, or at least the associated mobilities, cannot be altered until after passage through DMA2. I can't figure out where in Eq. 16 you could safely normalize diameters.

Thank you. This is indeed a poor description on my part. (And the referee is of course correct). In fact the code evaluates

$$\sum_{k=1}^{m}\left(\mathbf{O}_k * \mathbb{M}_k^{\delta 1}\right)$$

which results in $m$ apparent + 1 mobility diameter distributions after passage through DMA 2. The resulting sizes are then normalized by $D_{dry}$. The text has been updated accordingly.

Line 327: "… the DMA  setups $\Lambda_1, \Lambda_2, \delta_1, \delta_2$ …"

Corrected.

Lines 332-333: "Transmission through DMA 1 is computed for a specified $z^s$ (the dry mobility) and $g_0$ (the growth factor) via Eq. (13)." The humidity conditioner as represented by $g_0$ is NOT part of DMA1.

Corrected.

Line 339: "… the matrix is non square." This is only true if j≠i. Perhaps j is typically much smaller than i, but you haven't said anything about that.

This was a "typo" (the sentence was meant to read):

If the vector $Z$ inside the square bracket of Eq. (15) equals that of DMA 1, the matrix is  square.

Line 342: "The advantage of interpolation is that the the matrices $\mathbf{O}_k$ are smaller."

Corrected.

Here again it is implied that j is significantly smaller than i but you have not explained why that is.

Corrected.

..the matrices $\mathbf{O}_k$ are square and of dimension $j \times j$. In that case, the transmitted and grown distribution from DMA 1 ( i bins along the mobility axis of DMA 1) is interpolated onto the mobility grid of DMA 2 prior to evaluating $\mathbf{O}_k * M_k^{\delta 1}$ The advantage of this approach is that for $j < i$ the matrices $\mathbf{O}_k$ are smaller and subsequent calculations are faster.

Line 347: "… the humidified mobility pseudo-distribution function …" or some such modification.

Corrected.

Equation (17) is cast into matrix form such that the humidified apparent +1 mobility diameter distribution function is given by

Lines 350-351: "If the grids for $P_g$ and that of DMA 2 do not align, interpolation is used to map the $P_g$ grid onto the DMA 2 grid." The $P_g$ grid would refer to the discretization of $g_0$, or G, using the notation from above. What is being interpolated onto the DMA2 grid is $d_1^s \cdot$ G.

Changed as follows.

If the grids used to represent the growth factor distribution and that of DMA 2 do not align, interpolation is used to map the growth factor bins from the growth factor distribution onto those corresponding to the DMA 2 grid.

Lines 351-352: "The choice of $i, j, n$, the ranges of mobility grids for DMA 1, DMA 2, and the range of $P_g$, is only constrained …" Insert comma after $P_g$. The range of $P_g$ is simply [0,1]; it is the range of $g_0$ that is needed here or perhaps "the range of the growth grid for $P_g$".

Changed as suggested.

The choice of $i$, $j$, $n$, the ranges of mobility grids for DMA 1, DMA 2, and the range of the growth grid for $P_g$, is only constrained by computing resources and a physically reasonable representation of the problem domain.

Line 359: A new paragraph should start with "Figure 1 …".

Corrected.

Line 390: Concerning bounds for $P_g$, if $P_g = dF/dg$ from Figure 1 caption and as would be appropriate in Eq.16, then there is no upper bound for $P_g$. For if one introduced a perfectly monodisperse as the input aerosol, $P_g$ would range from zero to infinity. However, in discretizing the integral of Eq. 16 it is possible to assign the bin width factor, $\Delta g$, to the $dF/dg$ vector to get a $\Delta P_g$ vector. This would be properly bounded by [0,1]. This should be explained more clearly or the notation in the equations appropriately modified to show this.

The regularization/fitting is performed in frequency space and the data are subsequently converted to $P_g = dF/dg$. The reasons for this are (1) the ability to specify $B_{[0,1]}$ and that the error metrics are better behaved in the frequency domain. This was hinted at in the caption of Figure 3, but should have been made more explicit in the text.

**Here $B_{[0,1]}$ is shorthand for setting all lower bounds equal to zero and all upper bounds equal to one. The a-priori estimate $x_0$ is taken to be the normalized apparent growth factor distribution derived from the measured response function, where the normalization ensures that the sum over all bins is unity. Note that the inversion is performed treating the growth factor distribution in units of frequency instead of frequency density. This choice enables the upper bound constraint of unity. Since the true noise-free input growth factor frequency distribution is known, the fidelity of the inversion can be evaluated by**

**computing the root mean square error between the noise-free solution and the regularized solution. Evaluating the root mean square error in frequency rather than frequency density space results in more comparable values when contrasting narrow and broad probability distribution functions.**

Lines 396-397: "Visual evaluation …  suggests …"

Corrected.

The balance of the paper was not reviewed.